# Striatum supports fast learning but not memory recall

Kimberly Reinhold[1], Marci Iadarola[1], Shi Tang[1], Annabel Chang[1], Whitney Kuwamoto[1], Madeline A. Albanese[1], Senmiao Sun[1], Richard Hakim[1], Joshua Zimmer[1], Wengang Wang[1] & Bernardo L. Sabatini[1✉]

Animals learn to carry out motor actions in specific sensory contexts to achieve goals. The striatum has been implicated in producing sensory–motor associations[1], yet its contributions to memory formation and recall are not clear. Here, to investigate the contribution of the striatum to these processes, mice were taught to associate a cue, consisting of optogenetic activation of striatum-projecting neurons in visual cortex, with the availability of a food pellet that could be retrieved by forelimb reaching. As necessary to direct learning, striatal neural activity encoded both the sensory context and the outcome of reaching. With training, the rate of cued reaching increased, but brief optogenetic inhibition of striatal activity arrested learning and prevented trial-to-trial improvements in performance. However, the same manipulation did not affect performance improvements already consolidated into short-term (less than 1 h) or long-term (days) memories. Hence, striatal activity is necessary for trial-to-trial improvements in performance, leading to plasticity in other brain areas that mediate memory recall.

Behavioural responses are reinforced if they lead to good outcomes and suppressed if they lead to bad outcomes. Such behavioural adaptation requires multiple cognitive processes including learning and memory recall. The striatum, the major input nucleus of the basal ganglia, is required for adaptive behaviour in humans and other animals[1–7], but whether the striatum contributes to forming a memory (that is, learning) or memory recall (short and long term) is not understood.

The part of the striatum that receives direct input from the visual cortex modulates behavioural responses to visual cues[2,8,9]. Lesioning this area, here referred to as the posterior dorsomedial striatum tail (pDMSt), in monkeys[10–14] and rodents[15–18] disrupts behaviours requiring a visual cue-to-action association. However, lesions[19] have irreversible, long-lasting consequences and therefore cannot be used to probe moment-by-moment contributions to behaviour, nor can lesions separately target learning and short-term memory recall.

Contributions of the pDMSt to learning and memory recall are unknown but can be addressed with a temporally precise, reversible loss-of-function optogenetic approach[20–26]. We developed such an approach and tested the hypothesis that the pathway from the visual cortex to the striatum stores the memory of a cue–action association acquired through practice and reinforcement.

In visually cued behaviours, the reinforced stimulus activates many parallel visual pathways, including subcortical pathways that bypass the visual cortex and its projection to the pDMSt. Therefore, to study the contribution of visual cortex-to-pDMSt in learning or memory recall, we designed a strategy in which the cue is optogenetic activation of pDMSt-projecting neurons in the visual cortex, ensuring that behaviour relies on these corticostriatal projection neurons. We combined this optogenetic cue with temporally precise optogenetic inhibition of striatal projection neurons (SPNs) to assess the contribution of the pDMSt to behaviours requiring an association with visual cortex activation.

We found that, in mice that learned an association between this optogenetic cue and a forelimb reach to obtain food, the pathway from the visual cortex through the striatum did not uniquely store the associative memory: loss of function of the pDMSt did not affect recall of the memory, indicating that non-striatum-projecting axon collaterals of the corticostriatal neurons probably triggered the cued action via another brain pathway. Indeed, inhibiting activity in the superior colliculus disrupted the initiation of cued actions, suggesting that this alternative pathway includes the superior colliculus.

Although inhibition of the pDMSt did not affect memory recall, it disrupted learning, including outcome-dependent, trial-to-trial incremental changes in reaching rates. Similarly, in an externally cued visual discrimination task, inhibiting the pDMSt disrupted learning but not memory recall, indicating that the optogenetic cue is learned by mechanisms analogous to those used in natural learning.

To reveal how the pDMSt supports learning, we studied dopamine signalling and the neural activity of putative SPNs[27,28] in the pDMSt during the behaviour. Dopamine release into the pDMSt represented the outcome of the reach. SPNs in the pDMSt encoded the combination of reach, outcome and the context of the reach (that is, whether it was cued or uncued). This combination predicted the behavioural change between trials during learning, consistent with a specific function of the pDMSt in trial-to-trial learning.

To study how the visual cue-recipient zone of the striatum[29,30] contributes to the trial-and-error acquisition and execution of a visual cortex-to-action association, we trained mice in a cued forelimb reaching task. Food-restricted, hungry and head-restrained mice first learned to reach forwards with the right forelimb[31] to retrieve food pellets

[1]Department of Neurobiology, Howard Hughes Medical Institute, Harvard Medical School, Boston, MA, USA. ✉e-mail: bernardo_sabatini@hms.harvard.edu

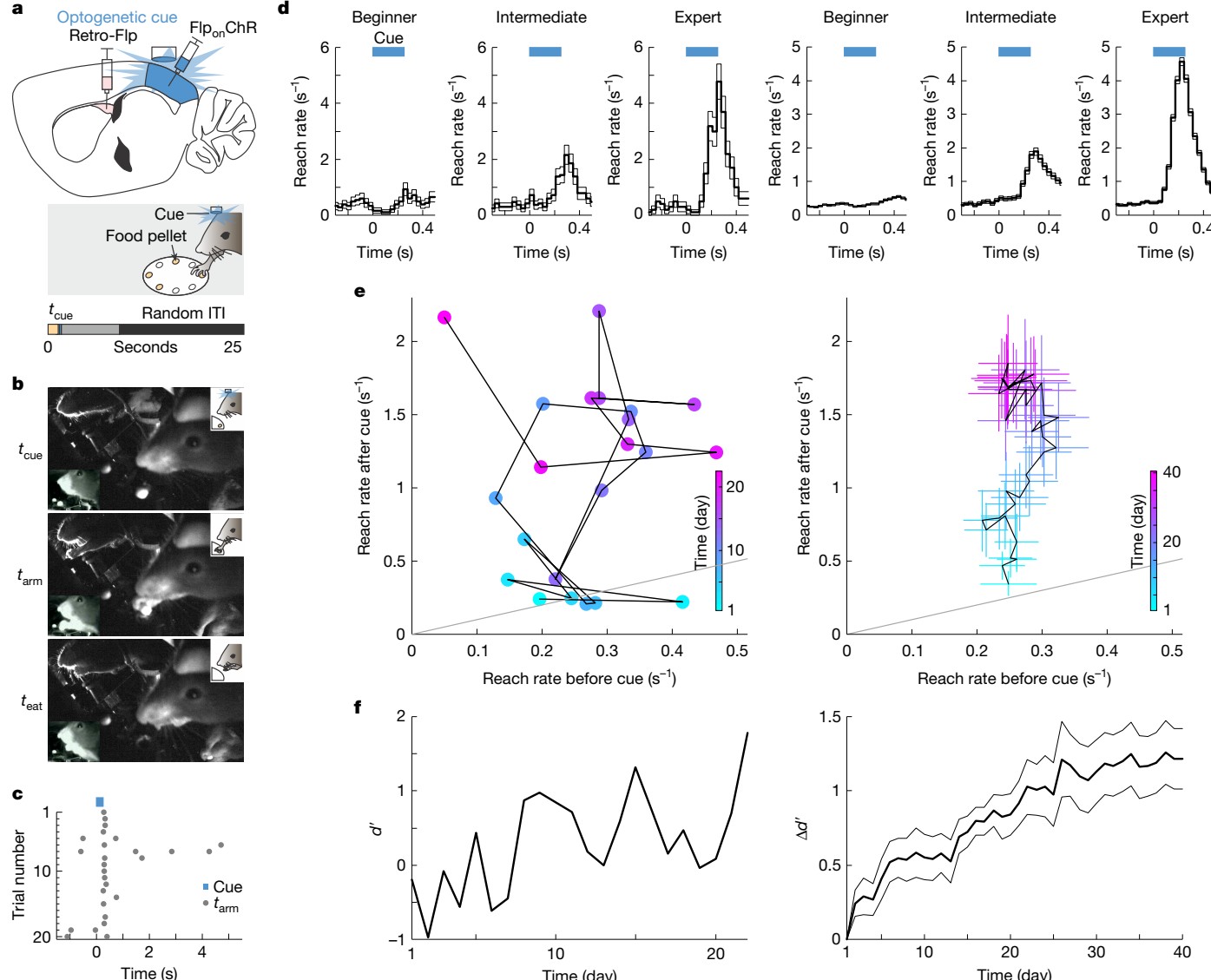

**Fig. 1 | Mice learned to associate the optogenetic activation of visual corticostriatal neurons with reaching to obtain food. a**, Schematic of a sagittal brain section showing injections of Flp-encoding retrograde AAV (retro-Flp) into the pDMSt (pink) and Flp-dependent ChR2-encoding AAV (FlpOn-ChR) into the visual cortex (blue; top). Blue light through a thinned skull activates ChR2-expressing visual cortex neurons projecting to the pDMSt, serving as the cue for food pellet availability. The behavioural apparatus with a disk delivering food pellets and the LED providing the cue is also shown (bottom). Each trial includes pellet presentation (tan), cue (blue) and random duration inter-trial interval (ITI; black). **b**, The optogenetic cue is paired with pellet presentation in 90% of trials. Infrared video frames show a mouse at cue onset ($t_{cue}$; top), reaching ($t_{arm}$; middle) and eating the pellet ($t_{eat}$; bottom). The insets show an alternative camera view (bottom left) and a task

schematic (top right). **c**, Example training session showing multiple trials (rows) aligned to $t_{cue}$ (blue) with reach timing ($t_{arm}$, grey dots). **d**, Reach rates across trials at different learning stages for an example mouse (left) and across 13 mice that learned (right; 9 male and 4 female) aligned to $t_{cue}$ (blue). The stages include: beginner ($d' < 0.25$; $n = 1,587$ and 14,822 trials for example and all mice, respectively), intermediate ($0.25 \leq d' < 0.75$; $n = 504$ and 4,110 trials) and expert ($d' \geq 0.75$; $n = 532$ and 8,268 trials). Data are mean ± s.e.m. **e**, Trial-averaged reach rates before versus after the cue over training days (colour code). Day 1 is the first day with 20 or more successful food grabs. **f**, $d'$ comparing reach rates before versus after the cue for an example mouse (left) and all mice (right; change in $d'$ relative to day 1) as a function of the training day. In **d**–**f**, example data are from the same example mouse (left), and summary data are from 13 mice (right).

presented at random intervals between 9.5 s and 26 s (Extended Data Fig. 1). The mice executed these forelimb reaches in a dark, light-tight box with masking stimuli that prevented sensory detection of the food pellet presentation, forcing the animals to perform reaches at random times to retrieve the food. Animal movements were recorded using multiple video cameras and analysed offline (Extended Data Fig. 1c–g). After 15 days of training, 97 out of 111 mice were able to retrieve and consume 20 or more pellets within a 1-h session.

After mice achieved this criterion, a food-predicting cue was introduced. This paradigm separates a first stage of motor learning (how

to physically retrieve the pellet) from a second stage that encourages, but does not require, learning about when to reach. Before introducing the cue, the baseline reach rate was low (approximately 0.25 Hz; Fig. 1), making any cue-evoked increase clear.

To limit the neurons that carry information predicting the presence of the food pellet, we used an internal, optogenetic cue that activates the visual cortex. We expressed blue-light-activated channelrhodopsin2 (ChR2) in corticostriatal neurons with cell bodies in the visual cortex that send axons to the pDMSt (injection of retrograde AAV-Flp into the pDMSt and Flp-dependent ChR2 into the visual cortex;

Extended Data Fig. 2a–c). We activated these neurons by unilaterally illuminating the visual cortex of the left hemisphere (that is, contralateral to the reaching arm) through a thinned skull (250-ms-long blue-light step pulse; Extended Data Fig. 2d,e). We refer to this optogenetic stimulus as the 'cue' (Fig. 1a). A distractor blue LED was positioned a few centimetres above the head and flashed at random times. The cue, delivered once per trial, predicted the availability of the pellet in 90% of trials (Fig. 1b). In these trials, the pellet became available shortly before cue onset (0.22 s before onset) and moved out of reach 8 s after cue onset. The delay until the next cue was random between 0 s and 16.5 s. In the remaining 10% of trials, unbeknownst to the mouse, the pellet was omitted.

Mice learned to use this internal, optogenetic cue to guide the timing of their reaches without altering reach kinematics (Fig. 1c,d and Extended Data Fig. 3; blue light, no opsin and other controls to ensure that mice attended to the optogenetic cue in Extended Data Fig. 4). The frequency of reaching immediately after the cue, compared with before the cue, increased across daily sessions pairing the cue with the pellet. After 20 days, the frequency of reaching was more than four times higher after than before the cue (Fig. 1d–f). We quantified the learning-related shift in reach timing as an increase in discriminability index ($d'$), which compares the probability of a reach occurring in the 400-ms time window immediately after the cue (cued window) to the probability of a reach occurring in the same-length window before the cue (uncued window; Fig. 1f).

Several controls indicated that the mice used the optogenetically driven activity of ChR2-expressing neurons as the cue to trigger forelimb reaches (Extended Data Fig. 4 and Supplementary Videos 1–6). First, in catch trials in which the food pellet was omitted, mice still reached immediately after the cue (Extended Data Fig. 4a). Conversely, on trials in which the cue was omitted but the pellet was presented, mice did not reach above chance levels (Extended Data Fig. 4b). Moreover, mice rarely reached in response to the distractor LED (Extended Data Fig. 4c). Furthermore, mice learned to respond to the optogenetic cue equally well when a red-light-sensitive optogenetic actuator, soma-targeted ChrimsonR, was used to activate the cue neurons, despite poor sensitivity of mouse retinas to red light (Extended Data Fig. 4d). By contrast, control mice that lacked any expression of an optogenetic actuator did not increase their reach rates around the light pulse (Extended Data Fig. 4e). These and other controls (Extended Data Fig. 4f) indicate that the increase in reaching frequency after the cue was triggered by a learned association with the optogenetic activation of visual corticostriatal neurons. We excluded sessions in which mice failed these controls (less than 15% of sessions).

The optogenetic cue targets the pellet-predicting information to visual corticostriatal neurons that innervate the pDMSt. However, these neurons also innervate other structures[32] and the cortex via collateral axons. To test whether neural activity in the pDMSt is required for mice to express the cue–reach association, we inhibited pDMSt SPNs using an optogenetic silencing approach (Extended Data Fig. 5). SPNs are the only output neurons of the striatum and send projections to downstream basal ganglia nuclei. Within the striatum, GABAergic interneurons, which synapse onto and powerfully suppress the activity of SPNs, selectively express NKX2.1. We exploited mice expressing Cre recombinase in NKX2.1+ cells to Cre-dependently express the red-light-sensitive optogenetic activator, ReaChR, in these striatal GABAergic interneurons (Extended Data Fig. 5a–c) within the region targeted by the ChR2-expressing cue neurons. To match the inhibited region of the striatum to the axonal target of cue neurons, we made ReaChR expression also contingent on the presence of Flp recombinase and injected AAV-Flp into the pDMSt. Thus, ReaChR expression (Flp and Cre dependent) and the retrograde labelling of the cue neurons (Flp dependent) were both controlled by the same Flp viral spread.

Optogenetically activating the interneurons (5 mW red-light step pulse) consistently suppressed more than 85% of the spiking activity of putative SPNs in the pDMSt in vivo, verified by high-density

multi-electrode array recordings in behaving mice across stages of learning (Fig. 2a–c and Extended Data Fig. 5d–f). Inhibition effectively suppressed the cue-evoked increase in pDMSt activity and was confined to areas within approximately 0.3 mm from the injection site (Extended Data Fig. 5f). This optogenetic loss-of-function approach was orthogonal to and combined with the blue-light-mediated optogenetic cue in the visual cortex (Extended Data Fig. 5g–k). Indeed, inhibition of SPNs using 5 mW of red light, when presented without the blue-light cue, did not elicit reaches in naive mice or in mice that had trained with the optogenetic cue (Extended Data Fig. 5g).

We used temporally precise, optogenetic inhibition of the pDMSt to determine what phases of task learning and execution require pDMSt activity. We first performed pDMSt inhibition in well-trained mice that consistently reached after the cue ($d' \geq 0.75$). Inhibition of pDMSt activity for 1 s beginning 5 ms before cue onset in a random subset of trials did not alter cue-evoked reach rates compared with interleaved control trials (Fig. 2d). Moreover, there were no effects of inhibiting the pDMSt on cue detection, reach initiation, the success rate of grabbing and consuming the pellet or other measures of motor kinematics (Extended Data Fig. 6a–f and Supplementary Videos 8–11). Hence, the cue–reach association can be fully expressed even during ongoing inhibition of the pDMSt, indicating that cue detection, action initiation and motor kinematics occur normally without neural activity in the pDMSt. Either the small amount of remaining activity in the pDMSt is sufficient to fully recapitulate the entire cued reaching behaviour, or long-term memory recall of the sensory–motor association is independent of the pDMSt and relies on signals sent via axon collaterals of the corticostriatal cue neurons to other brain regions[32].

To test the possible contribution of other brain regions to task performance in expert mice, we injected muscimol (Extended Data Fig. 6g–k) into the superior colliculus, which is downstream of the cue neurons via corticocortical synapses. Muscimol inhibition of neural activity in the superior colliculus disrupted the initiation of a reach in response to the cue after learning ($P = 0.00092$ comparing cued reaching in control versus muscimol, linear mixed-effects model; Methods) but did not affect spontaneous reaching ($P = 0.35$ comparing uncued reaching in control versus muscimol, linear mixed-effects model; Methods), supporting the interpretation that long-term memory recall after learning in this task is mediated by a pDMSt-independent pathway that includes the superior colliculus.

Independence of the learned behaviour from pDMSt activity enables a clear examination of its function during formation of the cue–action association. During learning, animals typically form short-term memories, which are later consolidated into long-term memories. Short-term memory, defined here as an improvement in task performance acquired during the daily approximately 1-h training session, might depend on pDMSt activity. To quantify the expression of short-term memory acquired during a session, we examined the change in $d'$ that occured from the beginning to the end of the session. On average, mice achieved a higher $d'$ by the end of each training session relative to the beginning ($d'$ of second half minus $d'$ of the first half of the session was 0.034 on average across 501 sessions from 24 mice; $P = 0.007$, Wilcoxon sign-rank test comparing difference to no change). For each mouse, we identified specific sessions, referred to as 'new learning days', in which the $d'$ achieved by the end of the day was higher than that achieved on any previous day (Fig. 2e–g). If pDMSt activity is required to express the improvement acquired within the day's session, inhibiting the pDMSt at the end of the session should reduce $d'$ to match its value at the beginning of the session. However, inhibiting the pDMSt at the end did not alter $d'$, indicating that short-term memory recall is also independent of pDMSt activity. Thus, improvements in performance acquired during a single training session can be recalled independently of pDMSt activity.

To test whether pDMSt activity is necessary for learning, we inhibited the pDMSt at every presentation of the cue, for 1 s beginning 5 ms before cue onset, over 20 consecutive days of training. This

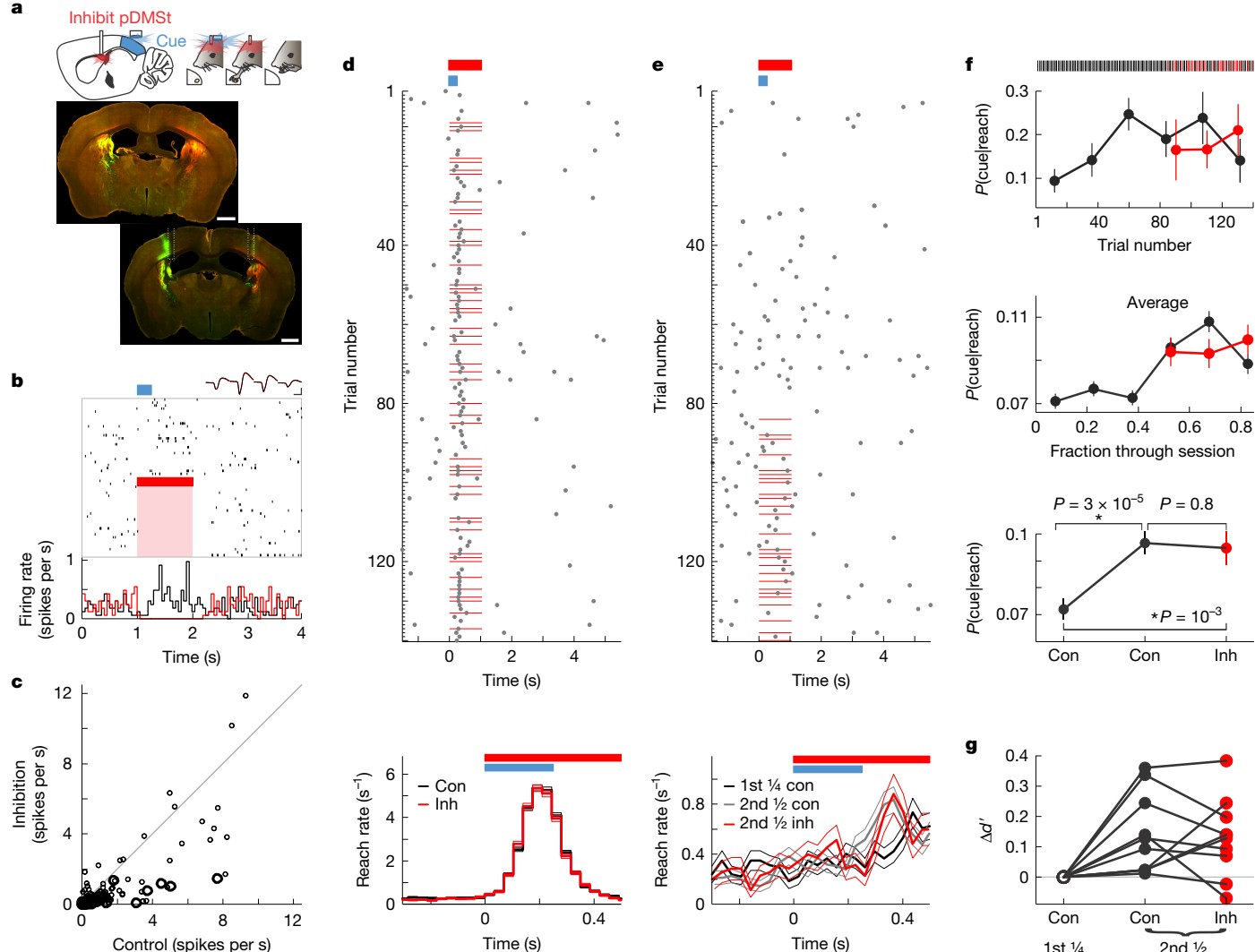

**Fig. 2 | Memory recall does not require activity in the pDMSt. a**, Schematic of pDMSt inhibition using red-laser illumination of ReaChR-expressing interneurons over 1 s starting 5 ms before the cue (top). Example mouse histology is also shown (bottom): retro-Flp (red), ReaChR–mCitrine (green) and fibre optic tracks. Scale bars, 1 mm. **b**, Example pDMSt SPN. Raster of action potentials (vertical lines) in a random subset of trials aligned to the cue (blue). The red bar and pink shading denotes laser inhibition. Spike waveforms (four channels) show no difference with the red laser on (red) or off (black). Scale bar, 0.5 ms by 20 μV. Trial-averaged peri-stimulus time histogram in control (black) versus laser trials (red) is also shown (bottom). **c**, Action potential rates of pDMSt SPNs comparing red laser versus control conditions for SPNs within 0.3 mm of peak ReaChR expression (thick line; $n = 40$ from 4 mice) and 0.3–0.5 mm away (thin line; $n = 184$ from 6 mice). **d**, Example expert session ($d' = 1.6$) displaying reach timing ($t_{arm}$) aligned to cue (blue) in control (con) and inhibition (inh)

trials (red lines; top). Reach rates across trials from expert mice ($d' \geq 0.75$; 11 mice, 105 sessions, $n = 7,577$ control trials in black and $n = 5,114$ inhibition trials in red) are also shown (bottom). Data are mean ± s.e.m. **e**, As in **d**, top, for the new learning day session ($d' = -0.09$) with the red laser interleaved in the second half (top). As in **d**, bottom, for 58 new learning day sessions ($d' < 0.75$; 10 mice including the red laser interleaved throughout session), separating the first fourth of the session ($n = 1,558$ control trials in black) and the second half ($n = 2,157$ control trials in grey, and $n = 1,004$ inhibition trials in red; bottom). **f**, Probability that reach was preceded by cue for datasets in **e** (example session (top), 58 new learning days (middle) and same 58 days binned into first quarter and second half of the session (bottom)) as mean ± s.d. of binomial across control (black) and inhibition (red) trials ($P$ values from two-proportion $Z$-tests). **g**, Change in $d'$ within a day's session ($n = 10$ mice). $n = 11$ males and 9 females (**a**–**g**).

dramatically impaired learning compared with a control cohort of mice that received that same light delivery pattern but did not express ReaChR (Fig. 3 and Extended Data Fig. 7). The improvement in $d'$ at days 15–20 of training was 0.77 ± 0.12 (mean ± s.e.m.) for the control cohort but only 0.12 ± 0.12 for the pDMSt inhibition cohort ($P = 6.2 \times 10^{-10}$, linear mixed-effects model; Methods). After these 20 days, pDMSt inhibition was stopped, and the previously inhibited cohort progressed in learning (0.54 ± 0.31 improvement in $d'$ by day 40; Fig. 3d), suggesting a temporary rather than a permanent deficit. These results demonstrate that pDMSt neural activity, in the period around the cued reach, is required for mice to learn that the cue indicates the presence of a food pellet. However, pDMSt neural

activity was not required for cue detection, reach initiation or any motor kinematics of the reach during and after learning, as described above (Extended Data Fig. 6a–f).

To examine the contribution of pDMSt activity to natural visual behaviours, we implemented a visual discrimination task (Extended Data Fig. 8). One of two visual stimuli was randomly presented: a reward-paired conditioned stimulus (500-ms ramp of light paired with the pellet) or an unpaired neutral stimulus (6-Hz flicker). These spatially identical but temporally distinct stimuli were emitted from the same LED. Control mice successfully learned to discriminate the stimuli, increasing reaching in response to the conditioned stimulus but suppressing reaching in response to the neutral stimulus (Extended Data

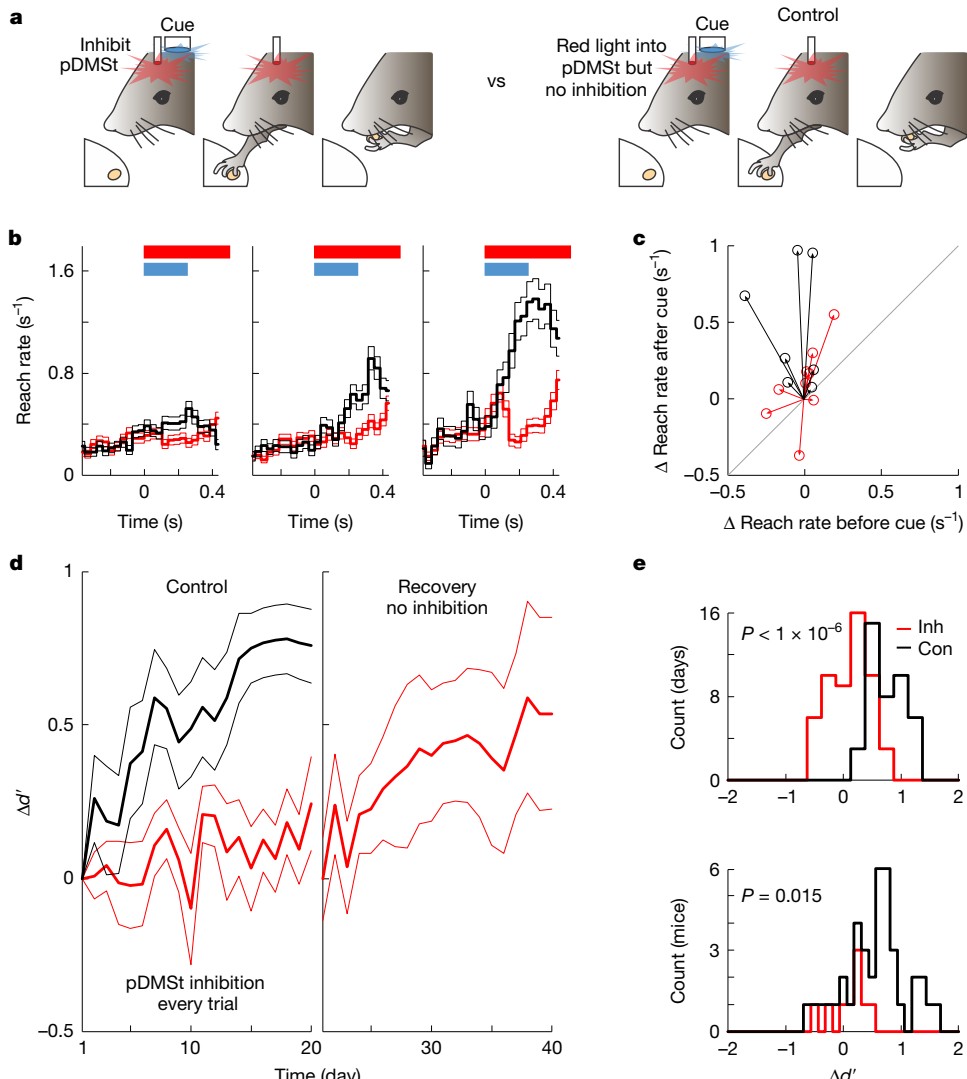

**Fig. 3 | Inhibiting the pDMSt disrupts learning. a**, One second of red light was delivered into the pDMSt at each cue onset. Separate cohorts did (left; *n* = 9 mice; red) or did not (right; *n* = 7; black) express ReaChR in striatal interneurons, serving as inhibition (left) and control (right) groups. Control mice received the same virus injections, fibre implants and red light to the pDMSt but lacked the recombinase-dependent ReaChR allele and, therefore, did not experience pDMSt inhibition. Experimenters were blinded to genotype. *n* = 9 pDMSt inhibition mice and *n* = 7 control mice. **b**, Reaching rate for inhibition (red) and control (black) mice (the blue bar denotes cue, and the red bar indicates the red light). Data are mean ± s.e.m. across trials for training days 1–7 (left), 8–14 (middle) and 15–20 (right). **c**, Change in cued and uncued reach rates from day 1 to days 15–20. Each line represents one control (black) or inhibition (red) mouse. Points above the grey line indicate more reaching after versus before the cue. **d**, Change in *d'* of reaching after versus before the cue

across mice for control (black) and inhibition (red) mice. Recovery refers to after day 20 when the red light was stopped in ReaChR-expressing mice (*n* = 8; 1 mouse died). Data are mean ± s.e.m. No significant difference in learning rate between recovery and control mice (*P* = 0.23, Wilcoxon rank-sum test comparing Δ*d'* on days 15–20 after normal activity in the pDMSt). **e**, Histograms of change in *d'* from days 1 to 15–20 across training sessions (top) and individual mice (bottom). The bottom panel includes all mice trained in this task: the pDMSt inhibition cohort (red; *n* = 9), control cohort (black; *n* = 7) and 32 more control mice that did not experience pDMSt inhibition consistently during learning (also black). *P* values are from a linear mixed-effects model (top; *P* = 6.2 × 10$^{-10}$; Methods) and two-sided Wilcoxon rank-sum test (bottom). There was no difference between the two groups of control mice, that is, 7 mice run as double-blinded controls and 32 other controls (*P* = 0.18 from two-sided Wilcoxon rank-sum test). *n* = 11 male and 5 female mice (**a**–**d**).

Fig. 8a,b and Supplementary Video 12). By contrast, when the pDMSt was inhibited during the 1-s window overlapping every presentation of both stimuli, mice failed to learn to discriminate between the stimuli, reaching equally in response to both (Extended Data Fig. 8b,c). After successful discrimination learning, inhibiting the pDMSt did not affect performance (Extended Data Fig. 8d). Therefore, learning and expression of a visual discriminative task rely on pDMSt activity in the same manner as the task using the optogenetic cue.

According to the theory of reinforcement learning, reinforcement of an association between the cue and the action depends on the outcome, such that only actions resulting in beneficial outcomes are reinforced.

In reinforcement learning, this outcome-dependent reinforcement leads to a behavioural update from one trial to the next. Exploiting the large dataset of trials acquired in the optogenetically cued behaviour, we examined whether successful reaches are reinforced in a manner consistent with reinforcement learning, as evidenced by a trial-to-trial change in behaviour. Furthermore, we examined whether any such reinforcement depends on neural activity in the pDMSt. We quantified the behavioural change from one trial to the next by considering sequences of three consecutive trials: trial *n* − 1, *n* and *n* + 1. We compared the behaviour on trial *n* − 1 to the behaviour on trial *n* + 1, contingent on the outcome of trial *n* (Fig. 4 and Extended Data Fig. 9).

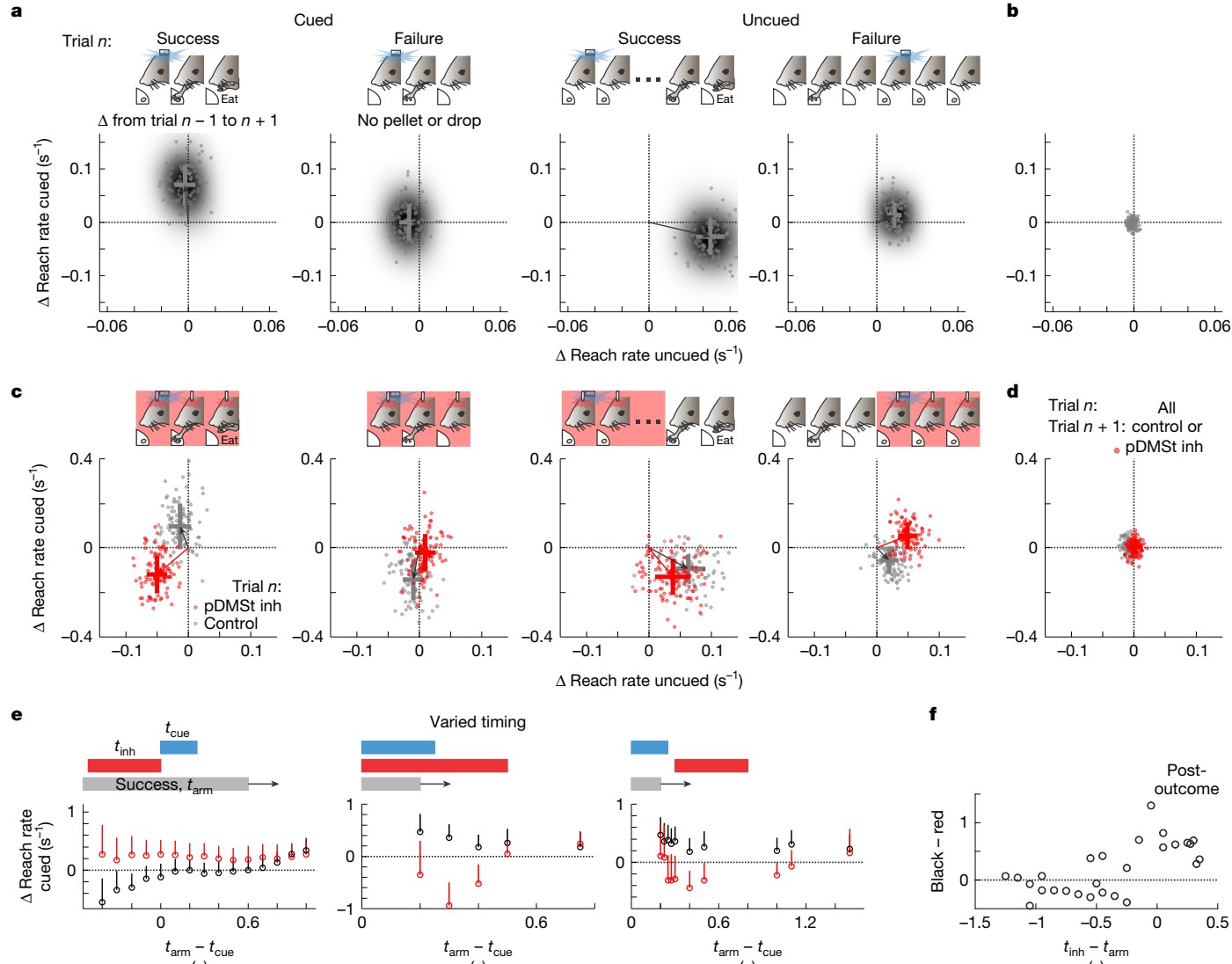

**Fig. 4 | Inhibiting the pDMSt disrupts outcome-dependent trial-to-trial reinforcement. a**, Changes in cued and uncued reach rates from trial $n - 1$ to trial $n + 1$ conditioned on context (cued versus uncued) and outcome (success versus failure) of the reach carried out in trial $n$. The $x$ axis indicates the change in reach rate in the uncued window (3–0.25 s before the cue). The $y$ axis denotes the change in reach rate in the cued window (cue onset to 400 ms after). Data from 37 mice are sequences with cued success ($n = 2,645$ trials), cued failure ($n = 3,280$), uncued success ($n = 1,703$) and uncued failure ($n = 6,264$) on trial $n$. The dots indicate 100 bootstrap runs (Methods) on the smoothed 2D histogram of change in cued and uncued rates from $n - 1$ to $n + 1$. The crosses denote mean ± s.e. across trials. **b**, As in **a**, but any reach type on trial $n$ ($n = 33,615$), showing that shifts depend on context–outcome conditioning. **c**, As in **a**, but with randomly interspersed pDMSt inhibition trials. The pDMSt was (red) or was not (grey) inhibited on trial $n$. Data from 16 mice include cued

success ($n = 464$ and 507 control and inhibition trials, respectively), cued failure ($n = 566$ and 580 control and inhibition trials, respectively), uncued success ($n = 278$ and 266 control and inhibition trials, respectively) and uncued failure ($n = 944$ and 925 control and inhibition trials, respectively) reaches on trial $n$. **d**, As in **c**, but inhibition is only on trial $n + 1$ (red; $n = 3,060$) versus control (grey; $n = 3,588$). **e**, Varied timing of pDMSt inhibition ($n = 15,727$ trials, 5 mice, 106 sessions). The $y$ axis is as in **a–d**, following a successful reach, as a function of timing of cue ($t_{cue}$, blue) and reach ($t_{arm}$, grey) in control (black circles) or inhibition (red circles, bar, $t_{inh}$). Reach windows in grey: 1.2 s (left) or 0.2 s (middle and right; Methods). Each point is the mean across trials. The vertical lines denote positive direction s.e. **f**, Black minus red data from **e** replotted as a function of relative timing of pDMSt inhibition ($t_{inh}$) and reach ($t_{arm}$). $n = 25$ male and 17 female mice (**a–f**).

We found that, if trial $n$ contained a cued reach resulting in a successful outcome, cued reaching was reinforced (that is, rate increased) on trial $n + 1$ relative to trial $n - 1$ (Fig. 4a,b). Furthermore, pDMSt inhibition that overlapped the cued reach on trial $n$ prevented this reinforcement (Fig. 4c,d; note that trial $n + 1$ does not experience pDMSt inhibition). To determine whether this effect was specific to the method used to inhibit the pDMSt, we also used an alternative method of inhibition by the inhibitory opsin GtACR2 expressed directly in SPNs. This alternative method also disrupted the reinforcement of the cued reach (Extended Data Fig. 9d). Therefore, mice demonstrate trial-to-trial

reinforcement of cue-triggered reaching that requires neural activity in the pDMSt.

Consistent with reinforcement learning, reinforcement of cue-evoked reaching was outcome dependent: if the mouse failed to grab the pellet on trial $n$, the rate of cued reaching was not increased on trial $n + 1$. Moreover, reinforcement depended on whether the reach was cued or uncued, such that cued reaching increased only if the reach in trial $n$ was cued, whereas uncued reaching increased only if the reach in trial $n$ was uncued (Fig. 4a). Finally, the effects of pDMSt inhibition depended on the timing of the reach relative to the inhibition. If the

mouse performed a successful uncued reach such that it did not overlap with pDMSt inhibition, then the reinforcement of uncued reaching occurred normally as in trials without inhibition (Fig. 4c). To measure the effect of pDMSt inhibition as a function of the timing of the action, we varied the timing of pDMSt inhibition with respect to the cue and reach (Fig. 4e,f and Extended Data Fig. 9c). pDMSt inhibition that overlapped the cue but preceded the reach did not disrupt reinforcement. By contrast, pDMSt inhibition that overlapped or immediately followed the reach disrupted reinforcement.

Our results indicate that neural activity in the pDMSt immediately after the reach is required for behavioural updates (Fig. 4) and learning (Fig. 3), but not expression of the memory (Fig. 2). To determine what features of pDMSt neural activity carry information about the cue, reach and action outcome, we measured both dopamine transients and neural spiking in the pDMSt in mice learning the task (Fig. 5; 65 sessions in beginner, 24 sessions in intermediate and 7 sessions in expert mice). We measured dopamine release within the pDMSt during behaviour by monitoring the fluorescence of the dopamine sensor dLight1.1 using fibre photometry (Fig. 5a). The cue did not evoke time-locked dopamine transients in the pDMSt (Fig. 5a). However, dopamine was modulated by action outcome: a successful outcome correlated with an increase in fluorescence, whereas a failure correlated with a dip in fluorescence, consistent with encoding of the reward. Hence, dopamine modulation in the pDMSt is outcome dependent.

To determine whether SPN activity is also outcome dependent, we measured the action potential firing of SPNs in the pDMSt using extracellular electrophysiological recordings with stereotactically targeted, high-density multi-electrode arrays. We limited our analysis to the activity of well-isolated single units that were putative SPNs (Fig. 5b), identified by established criteria[33]. Individual units responded to various sensory and behavioural events, including the cue, reach and outcome (Extended Data Fig. 10a). On average, unit activity increased around the reach and decreased after it (Fig. 5c).

If activity in the pDMSt drives trial-to-trial reinforcement of specific actions (for example, cued versus uncued reach), then pDMSt activity should encode the interaction between the action and its outcome, as needed to mediate the reinforcement of behaviour. Because the outcome only manifests after the mouse stretches out its arm and detects the presence or absence of the food pellet, we examined the 5-s 'post-outcome period' beginning when the arm is outstretched. This period does not include the cue nor the initiation of the motor action. On the basis of the neural response, as described by coefficients from a generalized linear model, we clustered the single-unit responses within this post-outcome period into two groups (Fig. 5d,e and Extended Data Fig. 10). One group was overall more active after a success than a failure (Fig. 5e,f). These same cells were also more active after a cued success than an uncued success (Fig. 5g,h). Therefore, this first group of cells preferred the cued context and a successful outcome. By contrast, the second group of cells was overall more active after a failure than a success (Fig. 5e,f) and tended to be suppressed by the cue (Fig. 5g,h). Therefore, this second group of cells preferred the uncued context and a failed outcome. Differences in behaviour (for example, chewing a pellet or not) during success and failure trials could give rise to different neural activity patterns. By contrast, cued successes and uncued successes could not be distinguished by metrics of behaviour (Extended Data Fig. 10p), suggesting that different neural activity patterns in these two trial types were shaped by the preceding cue and not by ongoing behavioural differences.

We examined whether the activity of these two cell groups in the post-outcome period (Fig. 5i) encoded the behavioural condition of trials sorted into four types: cued success, cued failure, uncued success and uncued failure. We divided the electrophysiology data equally into training and test sets and classified neurons as belonging to group 1 or group 2 using only trials in the training set. Using data from the test set, we attempted to decode the behavioural condition

of the trial. We found that a simple decoding scheme (that is, average firing rate of group 1 versus average firing rate of group 2; Fig. 5j) was sufficient to decode the behavioural condition (Fig. 5j–l; 76% accuracy for decoding cued success versus uncued success versus failure using 200 units) above chance levels (62 ± 2%, mean ± s.e.m.). Hence, the population neural activity in the pDMSt reflects both the context and the outcome of the reach. Moreover, the neural activity in the pDMSt after a success contains lingering information about the presence or absence of the cue up to 5 s after the cue disappears (Fig. 5j). By contrast, the behaviour of the animals, as opposed to neural activity, in this time window did not contain information about the past cue (Extended Data Fig. 10p).

Hence, pDMSt neural activity correlates with the combination of the reach context and outcome (Fig. 5), and this combination determines the direction of the trial-to-trial behavioural reinforcement (Fig. 4). Thus, the pDMSt neural activity is consistent with the specific reinforcement learning function of the pDMSt revealed by the optogenetic loss-of-function experiments.

## Conclusions

We found that activity in the pDMSt, the zone of the striatum that receives visual information, contributes to learning a sensory–motor association but not to recall of that association at either short (approximately 1 h) or long (days) timescales. Moreover, our study identifies a specific function of the pDMSt in the fast reinforcement of behaviour from one trial to the next during trial-and-error learning. Although it is not surprising that the striatum supports learning, it is striking that selective inhibition of this specific striatal subregion in a brief, 1-s window around the cue-evoked reach abolished learning over 20 days. By contrast, similar inhibition had no effect on carrying out the action, either spontaneously or as evoked by the cue, at any stage of learning. Thus, pDMSt activity only affected future actions in accordance with a function in behavioural reinforcement, that is, the pDMSt modulates the future likelihood of carrying out an action in a specific context depending on the outcome of the previous action. Indeed, we found that activity in putative SPNs of the pDMSt encodes this behavioural reinforcement.

### Striatum function after learning

A dominant theory is that the sensory cortex-to-striatum synapses are the storage site of learned cue–action associations, because corticostriatal plasticity correlates with learning[34], but see refs. 35,36. However, previous studies did not test the necessity of activity in the pathway through the striatum after learning. We found that associative memory recall was unperturbed by pDMSt inhibition, probably ruling out the possibility that corticostriatal synapses in this brain region are a necessary link between cue and action after an association has been learned.

Moreover, the absence of effect of pDMSt inhibition on the cue-evoked response precludes a direct contribution of pDMSt neural activity to detecting or attending to the cue. Our results contrast with previous work proposing a function of the pDMSt in visual attention[37]. However, our results are consistent with a recent study in mice showing that the projection from the visual cortex to the striatum is not required to respond to a visual cue after many weeks of training[9], although this lesion study could not probe the contribution of the pDMSt to the short-term memory recall of recently acquired associative memories. Here we found that these short-term memories are also independent of pDMSt activity.

The pDMSt-projecting cue neurons in the visual cortex have axonal branches forming synapses outside the pDMSt, for example, within the cortex. We hypothesized that synaptic connections outside the pDMSt might mediate recall of the cue–action associative memory after learning. For example, the visual cortex projects to the superior colliculus,

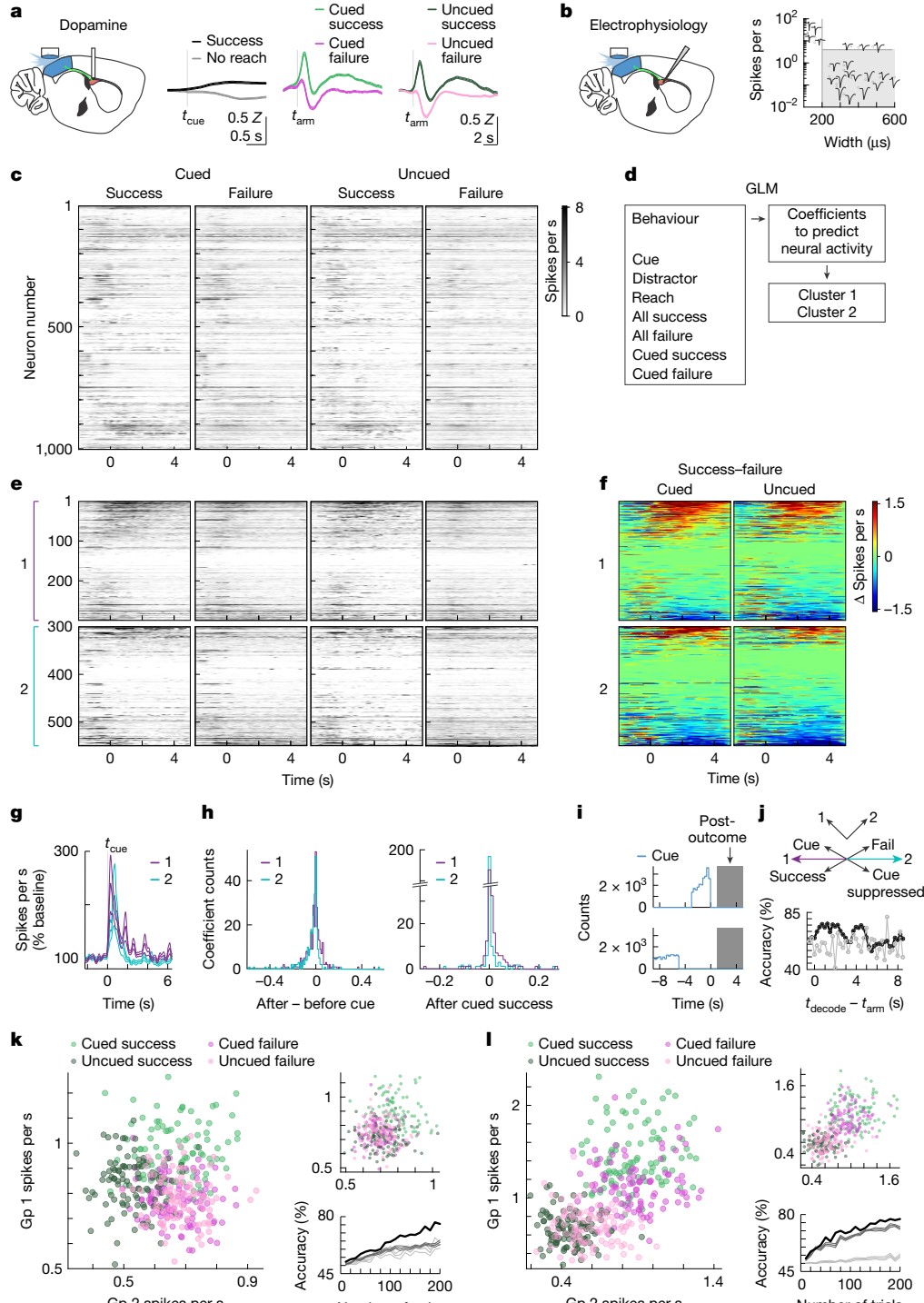

**Fig. 5 | Neural activity in the pDMSt correlates with reinforcement.**
**a**, pDMSt photometry of the fluorescent dopamine sensor dLight1.1. *Z*-scored
fluorescence (mean ± s.e.m. across *n* = 191 sessions, 12 mice) aligned to cue
onset ($t_{cue}$) for success (black) and no-reach (grey) trials, and to outstretched
arm timing ($t_{arm}$) for cued success (light green), cued failure (dark pink), uncued
success (dark green) and uncued failure (light pink) trials, baseline-subtracted
0.5 s before $t_{cue}$ or 2 s before $t_{arm}$. **b**, High-density neural recording in the pDMSt
(left), and single-unit waveforms from an example session (right). Putative
SPNs are in grey, and the rest of the figure shows only SPNs. **c**, Trial-averaged
spiking aligned to $t_{arm}$ (*t* = 0 s; *n* = 1,000 units, 16 mice). Greyscale from
0 to 8 spikes per second; black > 8. **d**, Generalized linear model (GLM)-based
identification of two SPN groups. **e**, As in **c**, but group 1 (top; purple) and group
2 (bottom; cyan) sorted by cued success minus cued failure (1–5 s after $t_{arm}$).
**f**, Success minus failure for cued (left) and uncued (right), sorted as in **e**. **g**, Spiking
normalized to the pre-cue baseline. Data are mean ± s.e.m. **h**, Histograms

indicate difference in GLM coefficient assigned to the cue for the period after
versus before the cue (left), and the GLM coefficient assigned to the period
after cued success (right). **i**, Histogram denotes $t_{cue}$ minus $t_{arm}$ for cued (top) and
uncued (bottom). Grey indicates the post-outcome period. **j**, Decoding scheme
contrasting cue-suppressed, failure-preferring group 2 versus cue-preferring,
success-preferring group 1 (top). The decoding accuracy (black) versus
trial-type shuffle (grey) using 0.5-s bins per 200 units is also shown (bottom).
**k**, Trial-type decoding (trials colour coded as in **a**) from post-outcome neural
activity (left). Each dot represents 1 iteration of bootstrap (200 units with
replacement). Also shown are shuffled group identities (top right) and
three-way decoding accuracy (combined cued-uncued failures) as a function
of unit count (black; bottom right). Shuffle group identity is in dark grey, and
shuffle trial type is in light grey. Data are mean ± s.e.m. across 10 shuffles.
**l**, As in **k**, but scatters from randomly sampling 90 individual trials. *n* = 13 males
and 11 females (**a**–**l**).

a known site of sensory–motor transformations, and polysynaptic activation of this brain structure might contribute to memory recall. Supporting this, pharmacological inhibition of the superior colliculus disrupted the cue–action association after learning.

## Striatum function during learning

Despite its lack of effect on task performance after learning, pDMSt inhibition profoundly disturbed learning. This aligns with a study showing impaired learning from dorsomedial striatum inhibition in mice using a brain–computer interface[21]. Reinforcement learning requires an animal to (1) use the outcome of an action to update the future behavioural plan, (2) store and recall the updated plan, and (3) execute it at the right time. pDMSt inhibition impaired neither action execution nor, after several tens of minutes, memory recall. However, pDMSt inhibition eliminated outcome-dependent performance improvements from one trial to the next. Hence, we propose that the pDMSt underlies outcome-dependent updates to the future behavioural plan enacted according to sensory context. This might explain why striatal activity is necessary for evidence accumulation tasks[20,22,38–40], in which animals continually update their future behavioural plans. This might also explain why ectopic striatal activations are sufficient to bias future behaviour[8,41–43].

The dependence of learning but not recall and performance on pDMSt activity was not limited to the optogenetically cued behaviour; inhibiting the pDMSt also impaired visual discrimination learning without impairing execution of the discrimination task once learned. However, visual detection was independent of pDMSt activity, because mice experiencing pDMSt inhibition during learning could still respond non-specifically to visual cues, although the mice failed to discriminate the conditioned from the neutral stimulus. Given direct projections from visual areas to the pDMSt[29,41], the pDMSt may be well placed to learn specific visual pairings.

## Striatal encoding of behaviour

In monkeys, neural activity in the visual cortex-recipient striatum encodes the visual cue, its value and signals related to value-guided saccades[10,27,44–48]. In rodents, pDMSt activity encodes the visual cue[49], but other features of the encoding scheme are unclear. We recorded approximately 1,000 putative SPNs and observed strong reach-related activity, as occurs in monkeys[50]. Furthermore, SPN activity encoded the combination of the action outcome and sensory context, and sensory context continued to be represented even after action completion. Changes in behaviour from one trial to the next depend on the combination of action outcome and sensory context; thus, the pDMSt contains the information necessary to drive learning-related behavioural changes. Indeed, striatum neural activity can drive behavioural changes[42]. Consistent with existing literature[43], dopamine transients in the pDMSt reflect the outcome of the action, providing a possible mechanism by which action outcome interacts with cue and action information in the striatum.

Here we identified a specific function of the pDMSt in learning as opposed to memory recall using a spatially and temporally precise loss-of-function approach. This same approach could be used to study other striatal subregions. Determining the function of the pDMSt brings us closer to understanding how the brain coordinates neural activity across functionally specialized brain systems to learn through trial and error.

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

## Methods

All procedures were carried out in accordance with the President and Fellows of Harvard College Institutional Animal Care and Use Committee protocol #IS00000571-6.

### Sex of mice

We used male and female mice in an approximately equal ratio ($n = 65$ males and $n = 62$ females). We did not observe any differences in the cued reaching behaviour between the sexes. All figures include both males and females.

### Housing of mice

Animals were housed on a reverse light cycle in groups (females) or singly housed (males). The room ambient temperature was 75 °F, and the relative humidity was 45%.

### Food restriction and habituation to head restraint

We weighed each head-plated and intracranially virally transduced mouse (see below) before beginning food restriction. During food restriction, we limited available chow to reduce the weight of each animal to approximately 85% of the pre-restriction weight of that animal. We switched the daily food from regular animal facility chow to Bio-Serv chocolate-flavoured, nutritionally complete food pellets (item number: F05301). We then began to handle the mice, as follows. On day 1, we habituated the mice to a gloved hand in the home cage and attempted to feed the mice peanut butter from the tip of the gloved finger. On days 2 and 3, we continued to feed the mice peanut butter and habituated mice to handling. On day 4, we began head restraint and fed the mice peanut butter while the head was restrained. On day 5, we fed the mice food pellets while the head was restrained. We presented the food pellets directly to the mouth by loosely attaching each food pellet to a wooden stick, using sticky peanut butter. The mouse could use its tongue and mouth to retrieve the food pellet and consume it. Once the mice comfortably ate food pellets while the head was restrained, we switched the mice to reach training (see the next section 'Training the forelimb reaching behaviour').

### Training the forelimb reaching behaviour

Forelimb reach training of mice (at least 2 months of age) was accomplished through manual interactions with the food-restricted and head-restrained mice over several days, according to the following stages. In stage 1, we taught each mouse to reach forwards with the right forelimb to touch a wooden stick. As a reward, we provided the mouse with a food pellet that was loosely attached to the stick with peanut butter, bringing the food pellet directly to the mouth of the mouse. In stage 2, we placed the food pellet at the end of the stick and required the mouse to push the food pellet off the stick and into its mouth. For stage 3, we gradually lowered the stick with the pellet until the mice reached forwards to the level of the food pellet presenter mechanism, located below and in front of the nose of the mouse. In stage 4, we removed the stick, requiring the mice to directly pick up the food pellets from the pellet presenter mechanism. During these manual interaction stages, we trained the mice on a behavioural rig that closely resembled the automated rig but with more space for the experimenter to interact with the mouse. We subsequently transitioned the mice to the automated behavioural rig, which included automated mechanisms for presenting pellets and was enclosed in a light-tight box (Extended Data Fig. 1a). On this rig, we trained mice to consistently and successfully pick up pellets in the dark[31]. Once the mice became proficient at reaching, we introduced a food-predicting cue (as described in the next section 'Training mice to associate a cue with presentation of the food pellet').

### Training mice to associate a cue with the food pellet

All cue training took place in an enclosed, dark, light-proof, sound-insulated behavioural box. Automated mechanisms, controlled by an Arduino, positioned the food pellets directly in front of and below the snout of the mouse (Extended Data Fig. 1a). After the mice became proficient at obtaining food pellets in the dark, we introduced the food-predicting cue. The trial structure was as follows (Extended Data Fig. 1b). The pellet moved into position in front of the mouse over 1.28 s. Following a 0.22-s delay, the cue turned on. The pellet remained stationary in front of the mouse for an additional 8 s before moving out of reach.

The 'pellet occupancy' is the likelihood that a pellet will be available in front of the mouse at any given time, unless the mouse has dislodged the pellet by reaching for it. The pellet occupancy was determined by the frequency of pellet loading. During the initial days on the automated rig, we trained the mice with a high pellet occupancy (80%) to provide them with ample practice in reaching for food pellets. Once the motor kinematics of the reaching movements stabilized, we reduced the pellet occupancy to 30%.

To prevent the mice from using the sound of the pellet presenter mechanism as a cue, (1) we continuously played an audio recording of the pellet presenter mechanism in motion, as a masking sound, and (2) the mechanism moved without presenting the pellet 70% of the time. This resulted in a 30% pellet occupancy. The sound of the pellet presenter mechanism was therefore not a reliable food-predicting cue.

To establish the inter-trial interval (ITI), we randomly selected a time interval from a uniform distribution between 0 and 3.5 s, as the first part of the ITI. Then, the automated behavioural rig entered one of two states. In state 1, occurring 30% of the time, the next trial began immediately. In state 2, occurring 70% of the time, the ITI continued for another 9.5–13 s, while the pellet presenter mechanism moved without presenting any pellet. Generally, mice did not reach before the cue (see 'Behavioural sessions included or excluded'), and mice appeared unable to time the ITI using an internal clock (Extended Data Fig. 4b,e,f).

### Catch trials

In a random 10% of trials when the cue turned on, the pellet was omitted. These catch trials were included to test whether the mouse paid attention to the cue or paid attention to the presence of the pellet.

### Preventing the mice from cheating

To encourage the mice to focus on the optogenetic cue and prevent them from using sensory systems to detect the presence of the food pellet through other means, we implemented the following strategies:
(1) We played a continuous, loud sound, which was pre-recorded audio of the pellet presenter mechanism, specifically, the stepper motor, through speakers positioned to the left and right of the mouse. This was done to mask the sound of the stepper motor.
(2) We placed fresh food pellets out of the reach of the mouse to mask the smell of the pellet that was directly in front of the mouse.
(3) A CPU fan was positioned to blow air continuously towards the nose of the mouse to prevent olfactory detection of the approaching food pellet.
(4) In a subset of mice, we trimmed their whiskers to test whether the mice used their whiskers to detect the food pellet. However, this did not have any effect on the cued reaching behaviour. Therefore, we did not trim the whiskers of all mice.
(5) The behavioural box was enclosed and completely dark to prevent the mouse from seeing the pellet.

We conducted numerous control experiments to determine whether each mouse responded to the optogenetic cue (Extended Data Fig. 4). In cases in which the mouse failed these controls, we excluded the entire behavioural session (see 'Behavioural sessions included or excluded').

### Video recording of the behaviour

We acquired video of the mice behaving using two infrared cameras (Extended Data Fig. 1a). The first infrared camera acquired the behaviour continuously at 30 frames per second (fps; Supplementary Videos 1–12). This camera sent the video to a DVR that logged the video onto a micro-SD card. The second infrared camera (Flea3 FLIR) acquired the behaviour at a higher frame rate: 255 fps. This high-speed camera acquired chunks of video beginning 1 s before each cue and continuing for 7.5 s after each cue with a gap in video acquisition between trials. This high-speed camera logged the video to a computer running the acquisition software FlyCapture2.

### Triangulating the paw position in 3D

To triangulate the paw position in 3D, we placed two mirrors around the mouse: one to the side of the mouse and one below the mouse (Fig. 1b and Extended Data Fig. 1a). These two mirrors gave orthogonal views, one from the side and the other bottom-up, of the paw during the reach (Fig. 1b). The high-speed infrared camera (Flea3 FLIR) was positioned so as to be able to see the paw from a top-down view and also, in the same frame, these two mirrors. We used DeepLabCut[51] to track the 2D position of the paw in each mirror. We then combined data from these orthogonal views to determine the paw position in 3D.

### Optogenetic cue

We used an optogenetic activation of corticostriatal neurons in the visual cortex as the food-predicting cue (Extended Data Fig. 2). To activate these corticostriatal neurons, we positioned the output of a fibre-coupled LED just above the thinned skull above the visual cortex of the left hemisphere. We placed a small U-shaped loop of clay around the fibre tip to confine the LED-emitted light to the area just above the skull. The fibre diameter was 1 mm. The fibre emitted 40 mW of blue light (473 nm). We controlled the LED with signals from the Arduino. The duration of the cue was 250 ms (step pulse). In some of the mice, we used the red light-activated opsin ChrimsonR[52] instead of ChR2 (ref. 53). Stimulation conditions were identical other than the use of 35 mW of 650-nm light for optogenetic activation. We did not observe any differences in the cued reaching between mice with ChrimsonR or ChR2 as the optogenetic activator in the visual cortex (compare Extended Data Fig. 4a with Extended Data Fig. 4d), and hence we combined these two groups of mice, unless otherwise specified.

### LED distractor

A distractor LED was positioned a few centimetres above the head of the mouse (Extended Data Fig. 1a). This LED flashed randomly with the same duration as the cue. The distractor LED was the same blue colour as the cue (473 nm). The distractor LED was too far away from the skull to optogenetically activate any neurons in the visual cortex. We controlled the distractor LED by signals from the Arduino. The duration of the distractor was 250 ms (step pulse).

### Blocked skull control

To investigate whether the reach is cued by the optogenetic activation of the visual cortex, we performed the following control. In expert mice that reliably reached to the optogenetic cue, we blocked the tip of the optical fibre conveying blue light from the LED to the thinned skull over over visual cortex, centered on primary visual cortex (V1). We inserted a small, thin piece of clay between the tip of the optical fibre and the skull. Blue light was still able to exit the fibre tip, but this blue light did not penetrate the skull. The optogenetic cue-triggered reaches were abolished by this procedure (Extended Data Fig. 4f), indicating that blue light must penetrate the brain to trigger the cued reach.

### Synchronizing the video with Arduino events

To synchronize the video of the mouse behaviour with Arduino events, we taped two small infrared LEDs to the front face of each camera. These infrared LEDs emitted light that was invisible to the mouse but detected by the infrared camera. One infrared LED turned on when the cue turned on. The other infrared LED turned on when the distractor LED turned on. Other behavioural events, for example, food pellet presentation, were directly recorded by the camera. Therefore, all relevant behavioural events were acquired along with the mouse behaviour and in the same frames as the mouse behaviour. Moreover, because the distractor LED flashed at random intervals, the pattern of this signal provided a unique sequence during each hour-long training session that enabled the alignment of all systems receiving a copy of the distractor LED signal.

### Processing the 30-fps video

To process the 30-fps video, we used custom code written in MATLAB and Python. In brief, the user first drew zones over six regions of the video frame: cue infrared LED, distractor infrared LED, perch zone, reach zone, pellet zone and eat zone (Extended Data Fig. 1c). The first two zones (cue infrared LED and distractor infrared LED) were used to synchronize Arduino events to the video of mouse behaviour (see previous section 'Synchronizing the video of behaviour with Arduino events'). The perch zone detected movement within the region where the paw rests before the reach. The reach zone detected movement of the paw into the zone between the resting position of the paw and the pellet (Extended Data Fig. 1d). The pellet zone detected the presence of the pellet directly in front of the mouse (Extended Data Fig. 1e). The eat zone detected chewing as an approximately 7-Hz oscillation of the jaw (Extended Data Fig. 1f). Behavioural events were defined by combining behavioural features detected in these various zones. For example, a successful reach was defined as a reach to the pellet, leading to a displacement of the pellet and followed by a long period of chewing (more than several seconds). A drop was defined as a reach to the pellet, leading to a displacement of the pellet and followed by no chewing. A reach that missed the pellet was defined as a reach without dislodging the pellet (this was a rare reach type). A pellet missing reach was defined as a reach, when the pellet was missing. Failed reaches included drops, reaches that missed the pellet and pellet missing reaches. A support vector machine was trained to separate the successes from the drops based on intensity data in the reach, pellet and eat zones. This support vector machine was applied to improve the discrimination of successes and drops. The automated behavioural classification pipeline was 96% accurate at classifying successes, 91% accurate at classifying drops and 98% accurate at classifying misses (Extended Data Fig. 1g).

### Measuring the accuracy of the automated pipeline

To measure the accuracy of the automated behavioural classification pipeline, we compared the output of the automated code pipeline to manually classified reaches (Extended Data Fig. 1g).

### Processing the 255-fps video

The high-speed video was processed using DeepLabCut[51] to track the paw trajectory in 2D. The 2D positions from two perpendicular mirrors were combined to determine the position of the paw in 3D.

### Virus injection

We diluted all AAV to a titre of $10^{13}$ gc ml$^{-1}$ or lower. The following viruses were used: pAAV-EF1a-mCherry-IRES-Flpo (Addgene #55634; packaged in AAV2/retro); pAAV-Ef1a-fDIO hChR2(H134R)-eYFP (Addgene #55639; packaged in AAV2/1); AAV2/8-EF1a-fDIO-ChrimsonR-mRuby2-KV2.1TS (modified from Addgene #124603); pAAV-hSyn1-SIO-stGtACR2-FusionRed (Addgene #105677; packaged in AAV2/8); and pAAV-hSyn-dLight1.1 (Addgene #111066; packaged in AAV2/9).

### Age of mice for virus injection

We used adult mice older than 40 days of age.

## Injection of the AAV carrying retro-Flp into the pDMSt

We injected 300 nl of AAV2/retro-EF1a-mCherry-IRES-Flpo into the pDMSt bilaterally. We targeted the pDMSt at 0.58 mm posterior, 2.5 mm lateral and 2.375 mm ventral of bregma. We lowered the virus-containing pipette (pulled glass pipette) to 0.05 mm below the target site, before retracting the pipette to the target site, waiting 2 min, and then injecting virus at a speed of 30 nl min$^{-1}$. After the injection, we waited 10 min before withdrawing the pipette from the brain.

## Injection of the AAV carrying Flp-dependent channelrhodopsin

We injected 300 nl of AAV2/1-Ef1a-fDIO-ChR2-eYFP into V1 of the left hemisphere. We targeted V1 at 3.8 mm posterior of bregma, 2.5 mm lateral of bregma and 0.65 mm ventral of the pia. After lowering the pipette to the target site, we waited 2 min before injecting. If we detected any leak of the virus out of the cortex, we lowered the pipette another 0.05 mm. We waited 10 min after the injection before withdrawing the pipette from the brain.

## Injection of the AAV carrying Flp-dependent ChrimsonR

We injected 300 nl of AAV2/8-EF1a-fDIO-ChrimsonR-mRuby2-KV2.1TS, where TS indicates soma-targeted, into V1 of the left hemisphere. We targeted V1 as described in a previous section ('Injection of the AAV carrying Flp-dependent channelrhodopsin').

## Surgical virus injection

We prepared all mice for surgery under isoflurane anaesthesia, as previously described[54,55]. In brief, after stereotactically flattening the skull, we drilled the hole in the skull, inserted the virus pipette to the target site, injected the virus, retracted the virus pipette and then sutured the skin. Orally administered carprofen or subcutaneous injections of ketoprofen were used as the analgesic. Mice were allowed to recover for at least 3 weeks before we implanted the headframe.

## Headframe implant and thinning skull over V1

We used isoflurane anaesthesia during the surgery and maintained the temperature of the animal using a closed-loop, thermoregulating heating pad. We covered the eyes in lubricant, removed the hair from the scalp, cleaned the scalp and cut the skin to expose the skull bilaterally around the midline from behind the lambdoid suture to just anterior of bregma. We stereotactically flattened the skull. We used a bone scraper and scalpel blade to scrape and score the skull. We thinned a 1.5 mm by 1.5 mm square of skull centred on V1 using a bone drill by hand. We put a thin layer of Vetbond onto the skull. We positioned the headframe, a thin bar, behind the lambdoid suture and perpendicular to the midline suture, so that the edges of the headframe protruded laterally just in front of the ears of the mice. We glued the headframe to the skull using Krazy Glue. The Krazy Glue is transparent, allowing light to access the thinned skull over V1. After the glue dried, we built up layers of opaque dental cement over all regions of the skull, except the 1.5 mm by 1.5 mm square centred on V1. We built up dental cement around the edges of this 1.5 mm by 1.5 mm square of thinned skull to create a pocket for the placement of the tip of the LED-coupled optical fibre. We used oral carprofen or subcutaneous ketoprofen as the analgesic. We allowed the animals to recover from the surgery for at least 5 days before beginning behavioural training.

## Definition of $d'$

We defined the discriminability index used to measure behavioural performance ($d'$) as

$$d' = z(\text{hit}) - z(\text{FA})$$

where $z(\text{hit})$ is the $Z$-score transformation of the hit rate, and $z(\text{FA})$ is the $Z$-score transformation of the false alarm rate. The hit rate represents the likelihood of observing one or more reaches right after the cue. Graphically, on a curve showing the distribution of the number of reaches in this time window, the hit rate corresponds to the fraction of the area under the curve that lies beyond a certain threshold (one reach in our case). As the hit rate goes up, more and more of the curve is above the threshold and our $Z$-score increases. We can use the inverse of the cumulative density function to calculate the $Z$-score associated with the hit rate. Note that scaling the curve or moving its mean, assuming the same transformation is applied to the threshold (one reach), does not change that fraction of the area under the curve. Thus, we can use the inverse of a standard normal cumulative density function to calculate the $Z$-score from the hit rate. We defined a false alarm as one or more reaches in the time window before the cue. As the hit rate probability goes up, $z(\text{hit})$ increases, and analogously, as the false alarm probability goes up, $z(\text{FA})$ increases. As the false alarm probability goes down and the curves for hits and false alarms become easier to discriminate, $z(\text{FA})$ decreases. Thus, a larger difference in the amount of reaching after the cue relative to before the cue produces a larger $d'$. This is why $d'$ is called the discriminability index. It captures how discriminable two curves are, accounting for both mean and variance. A positive $d'$ indicated more reaches after versus before the cue. To calculate the hit rate, we measured reach rates in the time window 400 ms immediately after cue onset. In Figs. 1 and 3, we used two different time windows before the cue to calculate two false alarm rates. The first false alarm window was 400 ms in duration beginning 400 ms before the cue. The second false alarm window was 400 ms in duration beginning 1 s before the cue. We calculated a $d'$ for each false alarm window, then we used whichever $d'$ was lower. This ensured that we did not miss any preemptive reaching, which should decrease $d'$. In Fig. 2, we used the time window 400 ms in duration beginning 400 ms before the cue to calculate the false alarm rate.

## Defining learning stages

We defined beginner as any session with $d' < 0.25$. We defined intermediate as any session with $0.25 \leq d' < 0.75$. We defined expert as any session with $d' \geq 0.75$.

## Behavioural sessions included or excluded

Because video analysis is computationally intensive, we did not analyse data from every session. Instead, we analysed data from every other day for each mouse, except for mice used to plot the learning curves or when otherwise specified. In these cases, daily analysis was performed. We have included data from all analysed sessions in our figures and statistics. However, we excluded all the behavioural data collected by one mouse trainer who set up the behavioural rig improperly ($n = 5$ mice).

To eliminate the early motor learning stage, when the mouse is still in the process of learning how to grab food pellets (Extended Data Fig. 3), we defined day 1 for the learning curves as the first day when the following two criteria were met: (1) the mouse successfully grabbed and consumed 20 or more pellets during a session lasting 45 min or longer. (2) Pellet occupancy (as described in the previous section 'Training mice to associate a cue with the food pellet') was 60% or less. This second criterion ensures that the mouse experiences both successful reach attempts when the pellet is present after the cue and unsuccessful reach attempts when the pellet is absent before the cue.

If we observed any obvious cheating behaviour, that is, preemptive reaching before the cue at a level above the spontaneous baseline, we excluded the entire session from analysis. This rarely occurred; however, in some cases, the mouse appeared able to consistently detect the approaching pellet without using the cue, despite our extensive efforts to mask the presentation of the pellet. If mice could detect the pellet approaching, they always reached before the cue. Mice never patiently waited over the 0.22-s delay between final pellet presentation and the cue onset. Thus, we were able to detect with high certainty any preemptive reaching (that is, cheating) behaviour.

## Strategy for suppressing pDMSt neural activity

Direct optogenetic inhibition is limited in its efficiency, if the fraction of cells that express the inhibitory opsin and are exposed to sufficient light power is less than 100%. Rather than use a direct optogenetic inhibition of SPNs, we developed an approach to silence the SPNs. The logic was as follows (Extended Data Fig. 5a). Some inhibitory interneurons have promiscuous connectivity and release the neurotransmitter GABA onto SPNs. We reasoned that it might be possible to express an activating opsin in a subset of inhibitory interneurons with the result of strongly inhibiting a very large fraction of SPNs. We targeted the striatal interneurons using the NKX2.1–Cre transgenic mouse line. Approximately 90% of the striatal interneurons express the transcription factor NKX2.1 during development, and SPNs do not express NKX2.1. However, many other neuron types, outside of the striatum, also express NKX2.1 during development. Therefore, we chose an intersectional approach to target the NKX2.1$^+$ cells within the pDMSt specifically. We used Cre recombinase to target the NKX2.1$^+$ cells, and we used Flp recombinase to target the pDMSt. First, we crossed the NKX2.1–Cre transgenic mouse line (Jackson Labs stock #008661) with the Cre-On and Flp-On ReaChR transgenic mouse line (R26 LSL FSF ReaChR-mCitrine, Jackson Labs stock #024846), which expresses a red-activatable variant of channelrhodopsin (ReaChR[56,57]) only when both recombinases, Cre and Flp, are present. In the double transgenic offspring, the Cre within NKX2.1$^+$ cells makes ReaChR expression dependent only on the presence of Flp. Second, we injected Flp recombinase into the pDMSt (see the section 'Injection of AAV carrying retro-Flp into the pDMSt'). Diffusion limited the spread of Flp around the injection site. As a consequence, all infected neurons in the pDMSt expressed Flp, but only the infected NKX2.1$^+$ interneurons also expressed ReaChR (Extended Data Fig. 5b). This led to a high level of ReaChR expression in the striatal interneurons but not in SPNs. Moreover, retro-Flp infected the corticostriatal cue neurons. This enabled the expression of both Flp-dependent ChR2 in corticostriatal projection neurons and Cre-dependent and-Flp-dependent ReaChR in striatal interneurons.

## Fibre implant surgery to optically access the pDMSt

To illuminate the pDMSt for optogenetic manipulations or dLight1.1 (ref. 58) fibre photometry, we chronically implanted optical fibres over the pDMSt. We prepared the mice for surgery, as described above in the section 'Headframe implant and thinning skull over V1'. We drilled two craniotomies above the pDMSt bilaterally (or one craniotomy for unilateral dLight fibre photometry). Each optical fibre was 2 mm long, 0.2 mm in diameter and had a 0.39 NA. We obtained these fibres from ThorLabs or Doric Lenses. We implanted each fibre pointing straight down, so that its tip would be situated at approximately 0.58 mm posterior, 2.3 mm lateral and 2.25 mm ventral of bregma. We glued the fibres to the skull using Loctite gel #454 and catalyst. Then, we built up dental cement around each optical fibre to provide more stability. The top of each fibre was coupled to an optical patch cord (0.39 NA), which connected to a laser for optogenetic stimulation or an LED for fibre photometry.

## Illuminating the pDMSt for striatal silencing

For mice expressing ReaChR in the striatal interneurons of the pDMSt bilaterally, we coupled each implanted optical fibre (one per hemisphere) to a Y-fibre patch cord (0.39 NA) connected to a Coherent Obis laser producing red light (650 nm). We modulated the power of the laser using transistor-transistor logic (TTL) pulses originating from the Arduino that controlled the behavioural rig. The power emitted from each optical fibre tip was 5 mW. The duration of the red-light step pulse was 1 s. The onset of the red-light pulse preceded the onset of the cue by 5 ms.

## Reaching to pDMSt inhibition alone

In mice trained to respond to the optogenetic cue, inhibiting the pDMSt without turning on the cue did not elicit reaching (Extended Data Fig. 5g). These mice did not experience pDMSt inhibition during training. However, when we trained the mice with pDMSt inhibition overlapping the cue during learning (either consistent pDMSt inhibition at every presentation of the cue or randomly interleaved pDMSt inhibition), infrequently ($n = 5$ mice out of 21 mice), a mouse seemed to learn to respond at a delay to pDMSt inhibition alone (for example, 'example mouse C' in Extended Data Fig. 7). To test whether the mouse responded to the cue or pDMSt inhibition alone, in a small fraction of trials, we inhibited the pDMSt without turning on the cue. Only 5 out of the 21 mice exhibited reaching to pDMSt inhibition alone. We did not exclude any mice based on this and included all the mice in the figures. However, we did verify that including or excluding these five mice did not qualitatively change the results (not shown). The reaching to pDMSt inhibition alone was variable day to day. It is possible that this small subset of the mice ($n = 5$) reached to a post-inhibitory rebound after pDMSt inhibition.

## Testing optogenetic strategy for pDMSt silencing

To assess whether ReaChR-mediated activation of NKX2.1$^+$ striatal interneurons elicits inhibitory, GABAergic currents onto SPNs, we conducted acute slice electrophysiology in the pDMSt (Extended Data Fig. 5c). We prepared coronal slices containing the pDMSt from adult NKX2.1–Cre crossed to Cre-ON-Flp-ON-ReaChR double transgenic mice that had received AAV retro-Flp virus injections into the pDMSt over 2.5 weeks before. For details on the slicing protocol, refer to 'In vitro slice electrophysiology' below. We obtained whole-cell recordings of putative SPNs, which did not express ReaChR–mCitrine (as described in 'Strategy for suppressing pDMSt neural activity'). The cells were held at 0 mV in voltage-clamp mode to isolate inhibitory currents. We illuminated the slice with red light (6–7 mW from a red-orange laser emitted at 590 nm). Upon illuminating the slice, we observed clear, fast and reliable outwards currents in the SPNs, consistent with light-induced GABAergic synaptic transmission from striatal interneurons (Extended Data Fig. 5c). To confirm the GABAergic nature of these currents, we applied 10 µM gabazine to the slice, which abolished the outwards current (Extended Data Fig. 5c). We recorded from a total of eight cells within the zone of ReaChR–mCitrine expression and two cells located outside of this zone (Extended Data Fig. 5c).

## In vitro slice electrophysiology

The experiments closely followed the procedures outlined in previous studies[59,60]. Mice were anaesthetized using isoflurane inhalation and subsequently subjected to transcardial perfusion with ice-cold artificial cerebrospinal fluid (ACSF) composed of the following: 125 mM NaCl, 2.5 mM KCl, 25 mM NaHCO$_3$, 2 mM CaCl$_2$, 1 mM MgCl$_2$, 1.25 mM NaH$_2$PO$_4$ and 11 mM glucose, resulting in an osmolarity of 300–305 mOsm kg$^{-1}$. This perfusion was administered at a rate of 12 ml min$^{-1}$ for a duration of 1–2 min. The brain was removed from the skull, and we prepared 250-µm or 300-µm coronal brain slices in ice-cold ACSF. Slices were then placed in a holding chamber at 34 °C for 10 min, containing a choline-based solution with the following composition: 110 mM choline chloride, 25 mM NaHCO$_3$, 2.5 mM KCl, 7 mM MgCl$_2$, 0.5 mM CaCl$_2$, 1.25 mM NaH$_2$PO$_4$, 25 mM glucose, 11.6 mM ascorbic acid and 3.1 mM pyruvic acid. Following this initial incubation, the slices were transferred to a second chamber with ACSF also maintained at 34 °C for a minimum of 30 min. Subsequently, the chamber was shifted to room temperature for the duration of the experiment. During recordings, the temperature was maintained at 32 °C, and carbogen-bubbled ACSF was perfused at a rate of 2–3 ml min$^{-1}$. For whole-cell recordings, we used pipettes (2.5–3.5 MΩ) crafted from borosilicate glass (Sutter Instruments). Cs-based internal solutions were used for voltage-clamp measurements and contained the following components: 135 mM CsMeSO$_3$, 10 mM HEPES, 1 mM EGTA, 3.3 mM QX-314 (Cl$^-$ salt), 4 mM Mg-ATP, 0.3 mM Na-GTP and 8 mM Na$_2$-phosphocreatine, with pH adjusted to 7.3 using CsOH, resulting in an osmolarity of 295 mOsm kg$^{-1}$.

## In vivo extracellular electrophysiology acquisition systems

For in vivo electrophysiology, two different electrophysiology systems were used at two different times in the project. First, we used a Plexon Omniplex recording system with a Plexon headstage and Neuronexus probe (A1x32-Edge-10mm-20-177) to record from eight mice. The Neuronexus probe had 32 linearly arranged recording sites, spaced at a distance of 20 µm between each pair of sites. We acquired data at 40 kHz using the Plexon software PlexControl, passed to a DAC card and PC. Second, we used the WHISPER recording system, custom-built at Janelia Research Campus, to record from 19 mice. We used the same 32-channel Neuronexus probe. Data were amplified and multiplexed by the WHISPER acquisition system, and acquired by the National Instruments USB-6366, X series card. We sampled data at a rate of 25 kHz. We used the program SpikeGLX to acquire data.

## In vivo extracellular electrophysiology recording configuration

While mice were briefly anaesthetized before the electrophysiology recording, we drilled a craniotomy to allow access to the brain (see 'Recording from the visual cortex' or 'Recording from the pDMSt'). We covered the craniotomy with Kwik-Cast, allowed the animals to wake up and returned the mice to the home cage. At the time of the recording and after the head was restrained, we removed the Kwik-Cast covering the craniotomy. Then we built up a temporary well to contain saline at the site of the craniotomy. We used Kwik-Cast to build up this well after the head had been restrained. We placed sterile 1X PBS (pH 7.4) into this recording well. As the reference ground, we used a silver chloride wire resting in this well and in the saline. Thus, all electrode channels within the brain were referenced to this point outside of the brain. We inserted the probe into the brain. We recorded broadband neural activity while mice performed the behaviour. After the recording session, we computationally high pass-filtered the neural data above 300 Hz to remove low-frequency signals and to obtain the high pass-filtered extracellular activity including action potentials. We periodically replaced the 1X PBS during the recording session, as necessary, to prevent the well and craniotomy from drying out. After the end of the recording session and after removing the electrophysiology probe from the brain, we removed the Kwik-Cast well from the skull of the mouse and covered the hole in the skull with a small amount of fresh Kwik-Cast. We returned the mouse to the home cage.

## In vivo acute recordings over days

We recorded acutely from the brain of each mouse over several consecutive days, no more than about 5 days. We then euthanized the mouse, extracted the brain and performed post-mortem histology.

## In vivo electrophysiology in visual cortex

To record from the visual cortex in behaving mice, we anaesthetized already trained and already head-framed mice during an additional, brief surgery (5–10 min). We closed the eyes of the mouse during this brief surgery. We drilled a very small hole through the skull over V1. This hole had a diameter of about 0.05 mm. To do this, we first thinned the skull until it cracked, and then we used the bent tip of a needle to flake off bone until the brain was exposed. We covered the exposed brain using a drop of Kwik-Cast applied to the skull. At the time of the recording, we restrained the head of an awake mouse, removed the Kwik-Cast from the skull, built up a Kwik-Cast well around V1 (as described previously in the section 'In vivo extracellular electrophysiology recording configuration'), added saline to this well, and then placed the electrophysiology probe into the brain, advancing the probe straight down into the brain at a rate of 3 µm s$^{-1}$ or slower. We targeted V1 at approximately 3.8 mm posterior and 2.5 mm lateral of bregma. We placed the probe in one of two positions: (1) we advanced the probe to the bottom of cortex (depth of about 850 µm), such that the deepest channel on the electrode array was just ventral of cortex, or (2) we advanced the probe until only the most superficial channel of the electrode array was still above the pia. We attempted to avoid any large blood vessels. We registered the depth of each channel according to the estimated bottom of the cortex (position 1) or the estimated top of the cortex (position 2). Although this is not the most accurate way to determine channel depth in the visual cortex, none of our scientific questions depended on exactly accurately registering the channel depths. We recorded extracellular activity while the mice behaved.

## In vivo extracellular electrophysiology recording from the pDMSt

To record from the pDMSt in behaving mice, we restrained the head of an already reach-trained mouse. We briefly anaesthetized the mouse by positioning a nose cone, which provided a light level of isoflurane anaesthesia, over the snout of the mouse. We closed the eyes of the mouse and drilled a small hole through the skull. We covered the craniotomy with a small drop of saline (1X PBS, pH 7.4). We built up a well around this craniotomy using Kwik-Cast. We placed the electrophysiology probe and ground wire into this recording well and added more saline. We advanced the electrophysiology probe into the brain at a rate of 5 µm s$^{-1}$ or slower. We targeted the pDMSt at approximately 0.58 mm posterior, 2 mm lateral and 2.63 mm ventral of bregma. To record from mice with a chronically implanted optical fibre positioned over the pDMSt, we angled the electrode and advanced the electrode through the brain diagonally, until the recording electrode sat beneath the chronically implanted fibre. At the time of an earlier surgery, when we had implanted the headframe onto the skull of the mouse, we had stereotactically flattened the skull and left bregma visible by covering bregma only with Krazy Glue, which is transparent (the rest of the skull was covered with dental cement, except over the visual cortex). Hence, we could use bregma to calibrate the location of entry of the recording electrode. We used an electrode angle of 59° pointed ventral and posterior, with respect to horizontal. We used an electrode angle of 32° pointed lateral, with respect to the midline suture. This electrode track nicely follows the dorsomedial edge of striatum, where the V1 axons terminate. We marked the recording site using dye on the recording probe (see 'Marking the recording track'). While advancing the probe, we removed the nose cone providing a light level of isoflurane anaesthesia to the mouse and opened their eyes. The mouse recovered from anaesthesia and performed behaviour, as the recording electrode entered the pDMSt. We recorded pDMSt activity while the mouse behaved, for about 1 h. Afterwards, we retracted the recording probe, removed the Kwik-Cast recording well, covered the craniotomy with Kwik-Cast and returned the mouse to its home cage.

## Marking the recording track in vivo

When recording from the pDMSt, we marked the recording track for viewing by post-mortem histology. On the last day of recording for each different pDMSt recording site, we coated the recording probe in DiI before inserting the probe into the brain. We quickly removed the PBS from the recording well to prevent the PBS from washing away the DiI. Once the probe had entered the brain but before advancing the probe to its final recording site, we added PBS back to the recording well. We always allowed the DiI-covered recording probe to sit at its final site for at least 15 min. We reconstructed the recording track by viewing DiI in histological sections (see 'Post-mortem histology').

## Post-mortem histology

To extract the brain, we deeply anaesthetized the mouse using isoflurane. After testing to be sure that the animal did not respond to a toe pinch, the animal was decapitated. We very quickly extracted the brain from the skull and put the brain into 4% paraformaldehyde, where it remained at 4 °C between 36 h and 48 h. We then transferred the brain into 1X PBS (for sectioning using a fixed tissue slicer) or 30% sucrose (for sectioning using a freezing microtome). We made coronal

sections that were 50 µm thick. We performed immunohistochemistry in two cases: (1) to locate SPNs (see 'Immunohistochemistry against DARPP-32'), or (2) to visualize the location of dLight (see 'Immunohistochemistry to visualize dLight'). Other fluorescent protein signals were not amplified. We mounted the brain sections on slides using a mounting medium containing DAPI. We sliced the entire forebrain starting at the posterior tip of V1 and moving anterior through all of the striatum. We imaged all brain sections and verified virus expression. We used an automated Olympus slide scanner to image the sections (either the VS120 or VS200).

### Immunohistochemistry protocol

First, we washed the brain slices in 1X PBS with 0.1% Tween for 90 min. Second, we washed the slices in 10% Blocking One buffer overnight at 4 °C. Third, we added the primary antibody and let the slices sit overnight at 4 °C. Fourth, we washed the slices in 1X PBS with 0.3% Tween (0.3% PBST) three times for 10 min each. Fifth, we incubated the slices in 10% Blocking One with the secondary antibody overnight at 4 °C. Sixth, we washed the slices in 0.3% PBST three times for 10 min each. Last, we washed the slices in 1X PBS for at least 10 min, before mounting the slices.

### Immunohistochemistry against DARPP-32

We performed immunohistochemistry against DARPP-32 (Extended Data Fig. 5b) using the Novus Biologicals primary antibody (NB110-56929; concentration 1 µg ml$^{-1}$) and an anti-rabbit secondary conjugated to Alexa 594 to localize SPNs (A-11012, Invitrogen; concentration 2 µg ml$^{-1}$).

### Selecting new learning days

We defined new learning days as days during learning before the mouse was an expert ($d' < 0.75$), when the $d'$ calculated for that day was higher than the $d'$ achieved by that mouse on any previous day. The last 10% of trials in each session were discarded, because mice disengaged from the task during this period.

### Measuring the effects of pDMSt inhibition on reach phases

To test whether pDMSt inhibition had any effect on different phases of the reaching behaviour (that is, initial fast ballistic movement of the arm towards the pellet, grasping the pellet, supination of the paw and raising the paw with the pellet to the mouth; Extended Data Fig. 6a–e), we used a combination of DeepLabCut[51] and manual quantification. To measure the trajectory of the initial fast ballistic movement of the arm towards the pellet, we plotted paw trajectories tracked using DeepLabCut[51]. To measure the duration of each phase of the reaching behaviour, we viewed the high-speed video and manually counted the number of frames belonging to each phase of the reach. The $\Delta t$ from the perch to pellet was the time required for the paw to move from its resting position to touching the pellet. The $\Delta t$ grasp was the time required for the fingers of the paw to close completely around the pellet. The $\Delta t$ grasp to mouth was the time required for the mouse to lift the pellet into the mouth.

### Spike detection and single-unit sorting

We examined the raw physiology signal for periods when the mouse was chewing. Chewing sometimes produced large artefacts in the data that were easily identified. As mice chew at about 7 Hz, the chewing artefacts were periodic at 7 Hz, although these artefacts also contained high-frequency content. The artefacts were much larger than any spikes. We removed any chewing artefacts by subtracting the common mode signal across all physiology channels, because the chewing artefact was identical on all channels. We verified that any spikes detected during these artefacts were identical in shape and size to the spikes detected outside of these artefacts, for a number of example single units when only one large unit was recorded per channel. We

filtered the physiology data between 300 Hz and 25 kHz. We then used UltraMegaSort to detect spikes and cluster single units, as described elsewhere[54,55].

### Identifying putative SPNs

We identified putative SPNs as in ref. 33. First, for each unit, we averaged all of its spikes to get the average waveform. Second, we defined the spike amplitude as the maximum size of the negative deflection. Third, we defined the width of the spike waveform at half-maximum (called 'width' in Fig. 5b) as the time delay between the falling and rising time points at half the spike amplitude. Fourth, we measured the average firing rate of the unit over the entire experiment. We used these features to classify the unit as one of the following types (see Fig. 5b for an example session with different unit types).

- SPN: width of the spike waveform at half-maximum ≥ 0.22 ms and mean firing rate < 4 Hz
- Tonically active neuron: width of the spike waveform at half-maximum ≥ 0.22 ms and mean firing rate ≥ 4 Hz
- Fast spiking: width of the spike waveform at half-maximum < 0.22 ms and mean firing rate ≥ 1.25 Hz
- Low-firing-rate thin: width of the spike waveform at half-maximum < 0.22 ms and mean firing rate < 1.25 Hz

### Defining the probability that a reach was preceded by the cue

We previously used $d'$ to represent the behaviour. $d'$ is a commonly used behavioural metric that compares reaching in the time window immediately after the cue (window A) to reaching in the time window before the cue (window B). However, reaches are sparse in this behaviour, and hence many trials are required to calculate a meaningful $d'$. The hit rate used to calculate $d'$ was essentially $P$(reach|cue). An alternative analysis approach is to define the probability that a reach was preceded by the cue, within some time window. We called this the probability $P$(cue|reach). We plotted $P$(cue|reach) to understand how the reaching changes within a single day's training session (Fig. 2f). $P$(cue|reach) increased within the day's training session. For the summary datasets across mice, we used the time window within 0.4 s of cue onset, for consistency with the $d'$ definition in Fig. 1. Thus, $P$(cue|reach) was the probability that a reach was preceded by the cue within a 0.4-s time window. However, when analysing the example session in (Fig. 2e, top), summarized in (Fig. 2f, top), we expanded the time window after the cue to 1.5 s, allowing us to calculate a meaningful $P$(cue|reach) for this single session. In contrast to $P$(cue|reach), the probability that a reach was followed by the cue (within 0.4 s) decreased within a single day's training session (0.048 ± 0.003 over the first fourth of the session, and 0.038 ± 0.002 over the last half of the session, $P = 0.01$ from a two-proportion $Z$-test, $n = 58$ new learning days from 10 mice).

### Control mice for illumination of the pDMSt during learning

To test whether silencing the pDMSt during learning affects behaviour, we trained two groups of mice at the same time (Fig. 3). The first group of mice ($n = 9$) experienced real silencing of the pDMSt. The second group of mice ($n = 7$) were controls that did not experience silencing of the pDMSt. These control animals were negative littermates from the NKX2.1–Cre transgenic mouse line cross to the ReaChR transgenic mouse line. To test whether the learning deficit observed in the pDMSt silencing group was simply due to brain damage as a result of virus injections or fibre implants, we performed identical virus injection and fibre implant surgeries on the control mice. The experimenters performing surgeries and training the mice were blinded to the genotype of each mouse from before the first surgery and throughout training. The pDMSt silencing group and control groups were handled identically. We used red light to illuminate the pDMSt bilaterally in the control mice, but this red light did not silence the pDMSt in the absence of ReaChR expression.

## Illumination of the pDMSt during learning (loss of one mouse)

One mouse in the pDMSt silencing cohort in Fig. 3 had to be eliminated for health reasons, before switching the cohort to the 'recovery' training stage post-pDMSt inhibition.

## Identifying sessions where the mouse learned

We identified training sessions in which the mouse improved at cued reaching over the course of the session by evaluating if $d'$ at the end of the session was more than 0.1 greater than $d'$ at the beginning of the session. To allow for cases in which the mouse improved either earlier or later in the session, we made three calculations:

$$\Delta d'_1 = d'_{\text{last 75\% of session}} - d'_{\text{first 25\% of session}}$$
$$\Delta d'_2 = d'_{\text{last 50\% of session}} - d'_{\text{first 50\% of session}}$$
$$\Delta d'_3 = d'_{\text{last 25\% of session}} - d'_{\text{first 75\% of session}}$$

If either $\Delta d'_1$, $\Delta d'_2$ or $\Delta d'_3$ were greater than 0.1, we classified the session as one in which the mouse learned.

## Injections of muscimol into the superior colliculus

We injected 1.5 µg µl$^{-1}$ muscimol (M1523, Sigma-Aldrich) dissolved in 0.9% NaCl in ddH$_2$O (Extended Data Fig. 6g–k). Injections were stereotactically targeted to the superior colliculus at coordinates 4.6 mm posterior to bregma, 0.8 mm lateral to the midline and 1.9 mm deep. To avoid the sinus and a chronically implanted headframe, we used an angled approach (either 18° or 48° from vertical), advancing the pipette laterally to medially and dorsally to ventrally. We used two different injection systems: a Drummond NanoJect for four mice and a WPI injector for the remaining four mice. We briefly anaesthetized the mice, performed a small craniotomy (on the first injection day) and injected muscimol at 30–40 nl min$^{-1}$. After injection, we waited 2 min before retracting the pipette and waking the mouse. The total anaesthesia duration was less than 15 min. Mice were allowed to recover fully in their home cage for 10–20 min, resuming normal behaviour, before being transferred to the behavioural rig for 1-h-long cued reaching sessions. The mice were then returned to the home cage. We interleaved control days (no injection) with muscimol or saline injection days over several successive days.

Mice were excluded if they were unable to perform spontaneous reaches after the muscimol injection, as cue–reach associative memory could not be assessed. We titrated muscimol volumes to minimize the disruption to spontaneous reaching. The muscimol injection volumes in Extended Data Fig. 6g–k were 115 nl, 100 nl and 100 nl (mouse 1); 50 nl and 50 nl (mouse 2); 90 nl (mouse 3); and 20 nl (mouse 4). The saline injection volumes in Extended Data Fig. 6g–k were 100 nl saline plus dye (mouse 1); 70 nl saline (mouse 2); and 60 nl dye plus saline (mouse 4). We injected DiI as the dye. Three mice failed to recover spontaneous reaching on all muscimol injection days and were excluded. Another mouse was excluded, because it did not perform cued reaching on the control days. We were unable to perform a saline injection in mouse 3 or a dye injection in mouse 2, because the headframes came off, after which the mice were immediately euthanized. We processed all the brains as described in 'Post-mortem histology'.

For the four animals that did not recover spontaneous reaching after the muscimol injection, we observed gross motor defects, including spinning in the home cage and, on 2 of the muscimol injection days, seizure-like activity manifest as running-like movements of the forelimbs and hindlimbs. (In this latter case, we immediately euthanized the mice.) The spinning behaviour resolved within a few hours, but during this time, the mice did not perform spontaneous reaches when placed into the behavioural rig and hence could not be used to collect data about the cue–reach association.

## Statistics on muscimol injections into the superior colliculus

We used a linear mixed-effects model to assess whether muscimol injections affected a behavioural metric (that is, cued reach rate, uncued reach rate or $d'$; Extended Data Fig. 6k). To account for the non-independence of observations within the same mouse and potential baseline differences between mice, a random intercept was incorporated for each mouse. An overall intercept was also included to capture general trends. The model was

$$\text{metric}_{ij} = \beta_0 + \beta_1 \times \text{Condition}_{ij} + u_i + \epsilon_{ij}$$

where $\beta_0$ is the overall intercept, $\beta_1$ is the fixed-effect coefficient, Condition$_{ij}$ indicates muscimol or control (including no injection and saline days) for the $i$-th mouse at the $j$-th observation, $u_i \sim \mathcal{N}(0, \sigma_u^2)$ represents the random intercept for the $i$-th mouse, and $\epsilon_{ij} \sim \mathcal{N}(0, \sigma_\epsilon^2)$ is the residual error, implemented in MATLAB using the fitlme function.

## Statistics on learning curves with or without pDMSt inhibition

We used a linear mixed-effects model to assess whether pDMSt inhibition affected the change in $d'$ on days 15–20 relative to day 1. To account for the non-independence of observations within the same mouse and potential baseline differences between mice, a random intercept was incorporated for each mouse. An overall intercept was also included to capture general trends. The model was

$$\Delta d'_{ij} = \beta_0 + \beta_1 \times \text{Condition}_{ij} + u_i + \epsilon_{ij}$$

where $\beta_0$ is the overall intercept, $\beta_1$ is the fixed-effect coefficient, Condition$_{ij}$ indicates the condition of control or pDMSt inhibition during learning for the $i$-th mouse at the $j$-th observation, $u_i \sim \mathcal{N}(0, \sigma_u^2)$ represents the random intercept for the $i$-th mouse, and $\epsilon_{ij} \sim \mathcal{N}(0, \sigma_\epsilon^2)$ is the residual error, implemented in MATLAB using the fitlme function. When plotting the learning curves, on days excluded from the analysis because the mouse cheated, we interpolated $d'$ using neighbouring days or filled in the $d'$ from the last day before cheating.

## Natural visual discrimination behaviour

We designed a behavioural paradigm in which mice learned to discriminate between two visual stimuli: a cue, paired with food pellet delivery, and a distractor, unpaired with the pellet (Extended Data Fig. 8). Both stimuli were spatially unstructured and delivered via the same 1-mm-diameter optical fibre coupled to a 473-nm blue LED (maximum output of 40 mW) positioned several inches above the head of the mouse. The LED remained off during baseline periods and was activated only during stimulus presentation. The cue consisted of a gradual ramp in blue-light intensity, increasing from 0 mW to 40 mW over 0.5 s, with pellet delivery coinciding with the ramp onset. The distractor was a 6-Hz flicker, comprising six rapid light ramps (0–40 mW) over 1 s. The cue and distractor were randomly interleaved and presented with approximately equal probabilities.

## Natural visual discrimination data analysis

Our analysis of the natural visual discrimination was analogous to our analysis of the optogenetic cue (Extended Data Fig. 8). We measured reach rates within a 400-ms window starting after the onset of either the cue or the distractor. To assess discrimination performance, we calculated the 'rate ratio': the ratio of the reach rate following the cue to the reach rate following the distractor. Histograms of the rate ratio were generated across days and across individual mice. We used a linear mixed-effects model to compare the rate ratio on days 10–15 as a function of the condition, that is, whether the animal experienced pDMSt inhibition during learning. To account for the non-independence of observations within the same mouse and potential baseline differences between mice, a random intercept was incorporated for each mouse.

An overall intercept was also included to capture general trends. The model was

$$\text{Rate ratio}_{ij} = \beta_0 + \beta_1 \times \text{Condition}_{ij} + u_i + \epsilon_{ij}$$

where $\beta_0$ is the overall intercept, $\beta_1$ is the fixed-effect coefficient, $\text{Condition}_{ij}$ indicates the condition for the $i$-th mouse at the $j$-th observation, $u_i \sim \mathcal{N}(0, \sigma_u^2)$ represents the random intercept for the $i$-th mouse, and $\epsilon_{ij} \sim \mathcal{N}(0, \sigma_\epsilon^2)$ is the residual error, implemented in MATLAB using the fitlme function. To compare the rate ratio across mice as a function of the condition, we used the Wilcoxon rank-sum test.

### Changes in behaviour from trial to trial

To examine trial-to-trial changes in behaviour that underlie learning, we selected training sessions in which the mouse learned (see 'Identifying sessions where the mouse learned'). We then considered the individual cue presentations and reach attempts comprising these sessions. To determine how the outcome of one trial affected the next, we considered sequences of three neighbouring trials: trial $n - 1$, trial $n$ and trial $n + 1$. This three-trial sequence analysis avoids issues of regression to the mean. We measured how behavioural changes from trial $n - 1$ to trial $n + 1$, contingent on the behavioural experience of trial $n$. We defined behaviour as a 2D quantity, the rate of reaching in the cued window versus the rate of reaching in the uncued window. The cued window was defined as the 400-ms time window immediately after cue onset. The uncued window was defined as the time window beginning 3 s before cue onset and ending 0.25 s before cue onset. To plot how the behaviour changed in this 2D space, we ran a bootstrap by resampling, with replacement, all trial sequences, in which the behaviour of trial $n$ matched a particular type (that is, cued success, cued failure, uncued success or uncued failure; see the next paragraph). If we began with $m$ trials of this particular type, we resampled $m$ trials at each iteration of the bootstrap. For each iteration of the bootstrap, we subtracted the average behaviour on trial $n - 1$ from the average behaviour on trial $n + 1$. This is represented by the following: mean(behaviour on trial)$_{n+1}$ (resample $i$)) − mean(behaviour on trial$_{n-1}$ (resample $i$)), where $i$ is the set of resampled trials for iteration $i$ of the bootstrap. Thus, this bootstrap analysis represents the change in the joint distribution of cued and uncued reach rates. We plotted 100 runs of the bootstrap as the scatter plots in Fig. 4 (each dot is the result of one iteration of the bootstrap). In the top row of Fig. 4, we also plotted a shaded region that represents the 2D histogram of the change in this joint distribution, after running 1,000 iterations of the bootstrap and filtering the resulting 2D histogram with a Gaussian filter with standard deviation equal to 0.0096 along the $x$ axis (Δreach rate uncued) and 0.024 along the $y$ axis (Δreach rate cued).

We classified the behavioural experience of trial $n$ as one of four types:
(1) Cued success: on trial $n$, the mouse
 (i) Did not reach before the cue
 (ii) Made a successful reach within 1 s after cue onset
(2) Cued failure: on trial $n$, the mouse
 (i) Did not reach before the cue
 (ii) Made a failed reach (that is, dropped pellet, reached but failed to touch the pellet or the pellet was missing at the time of the reach) within 1 s after cue onset
(3) Uncued success: on trial $n$, the mouse
 (i) Did not reach before the cue
 (ii) Did not reach in the 1.5-s time window after the cue
 (iii) Made a successful reach between 3.5 s and 7 s after the cue (note that successful reaches are not possible before the cue, when the pellet is missing)
(4) Uncued failure: on trial $n$, the mouse
 (i) Made a failed reach before the cue

 (ii) And was not chewing at the beginning of the trial (we excluded trials when the mouse was chewing at the beginning of the trial, because, if the mouse had its forelimb outstretched to chew, the mouse could potentially detect the approaching pellet with its already outstretched forelimb)
 (iii) Or made a failed reach between 3.5 s and 7 s after the cue
 (iv) Did not reach in the 1.5-s time window after the cue
 (v) Did not make any successful reaches at any time in this trial (that is, all reaches were failures)

To measure the effects of pDMSt optogenetic inhibition, we compared three-trial sequences when the optogenetic inhibition was on or off in trial $n$ ('inhibition on' or 'inhibition off'). To ensure that the inhibition off trials were interleaved with the inhibition on trials, we took inhibition off trials that were followed by an inhibition on trial at the trial position $n + 2, n + 3, n + 4$ or $n + 5$. Analogously, to ensure that the inhibition on trials were interleaved with the inhibition off trials, we took inhibition on trials that were followed by an inhibition off trial at the trial position $n + 2, n + 3, n + 4$ or $n + 5$.

Note that the time window of pDMSt optogenetic inhibition overlaps the cued success (Fig. 4c, first column) but does not overlap the uncued success (Fig. 4c, third column). This may explain why the pDMSt optogenetic inhibition disrupted the behavioural update after a cued success but not after an uncued success.

### No outcome-independent behavioural change

To test whether there was any systematic change in the behaviour that did not depend on the behavioural experience of trial $n$, we plotted the change in behaviour from trial $n - 1$ to trial $n + 1$, given any type of trial $n$ behavioural experience (Fig. 4b). Any type of trial includes trials when the mouse reached successfully, failed or did not reach. There was no systematic change.

### Effect of pDMSt inhibition on the current trial

To test whether pDMSt inhibition affects the current trial, we plotted the change in behaviour from trial $n - 1$ to trial $n + 1$, given (1) any type of trial $n$ behavioural experience, and (2) pDMSt inhibition during the cue on trial $n + 1$ versus no inhibition on trial $n + 1$ (Fig. 4d). pDMSt inhibition on trial $n + 1$ (beginning 5 ms before the cue and continuing for 1 s) did not produce a shift in behaviour from trial $n - 1$ to trial $n + 1$, consistent with data elsewhere in this paper showing no effect of pDMSt inhibition on the ongoing cued reaching behaviour (for example, Fig. 2d).

### Varied timing of pDMSt inhibition

To determine when pDMSt neural activity was required for trial-to-trial behavioural updates, we varied the timing of the 0.5-s optogenetic inhibition relative to the cue and reach. Inhibition was applied at one of three time points: (1) starting 0.5 s before cue onset, (2) simultaneously with cue onset, or (3) 0.3 s after cue onset. For each inhibition timing, we analysed sequences of three consecutive trials (trial $n - 1$, $n$ and $n + 1$) where the reach on trial $n$ occurred at different times with respect to the pDMSt inhibition. Figure 4e shows the change in reaching behaviour from trial $n - 1$ to trial $n + 1$ for successful reaches on trial $n$. The $y$ axis in Fig. 4e is identical to the $y$ axis in Fig. 4a–d and represents the change in reach rate in a 400-ms window following cue onset. The circles represent the mean across trials, and the vertical lines show the standard error (mean ± s.e.m.). For clarity, the line representing the mean − s.e.m. was omitted. The black dots are when trial $n$ was a control trial; the red dots are when trial $n$ contained pDMSt inhibition. Successful reaches before cue onset led to a decrease in cued reaching on trial $n + 1$, whereas successful reaches after cue onset increased cued reaching on trial $n + 1$. To ensure sufficient reach counts for statistical power, we used different reach time windows based on reach frequency. For example, we needed to use a long 1.2-s window for low-frequency

reaches before the cue. Hence, in the left panel of Fig. 4e, we used a 1.2-s-long reach time window. Because cued reaches occurred at a higher rate after the cue, we could use a shorter reach time window for the middle panel of Fig. 4e. We used a 0.2-s-long reach time window for the points at $x$ axis positions 0.2 s, 0.3 s, 0.4 s and 0.5 s, but we had to use a longer reach time window of 0.5 s for the point at $x$ axis position 0.75 s owing to lower reach counts. For the right panel of Fig. 4e, we used a 0.2-s-long reach time window for the points at $x$ axis positions 0.2 s, 0.225 s, 0.25 s, 0.275 s, 0.3 s, 0.4 s and 0.5 s. We used a 1-s-long reach time window for the points at $x$ axis positions 1 s, 1.1 s and 1.5 s owing to lower reach counts. Figure 4f displays the difference between the red and black points from Fig. 4e, plotted according to the time difference between the midpoint of pDMSt inhibition (middle of the 0.5-s window) and the midpoint of the reach time bin. We overlaid all the points from the panels in Fig. 4e to construct Fig. 4f.

## Control for behaviour change (backwards time control)

If the change in behaviour from trial $n - 1$ to trial $n + 1$ depends on the behavioural experience of trial $n$, then the effect on trial $n + 1$ should be manifest forwards in time but not backwards in time. If trial $n + 1$ showed the same shift in behaviour when 'time moved backwards', this would suggest a correlational structure in the data but not any causal effect of the behavioural experience of trial $n$. To test this, instead of conditioning trial $n + 1$ on trial $n$, we conditioned trial $n + 1$ on trial $n + 2$. We measured the shift in behaviour from trial $n - 1$ to trial $n + 1$, that is, before the particular behavioural experience of trial $n + 2$. This abolished the increase in cued reaching observed after a cued success, and this abolished the increase in uncued reaching observed after an uncued success (Extended Data Fig. 9a).

## Optogenetically inhibiting the pDMSt using GtACR2

We used a second, orthogonal optogenetic method to confirm that inhibiting the pDMSt disrupts the behavioural updates from trial to trial. We directly expressed soma-targeted GtACR2, a blue-light-stimulated inhibitory opsin, in SPNs. We injected an AAV carrying Cre-dependent GtACR2 (see 'Virus injection of GtACR2 into the pDMSt and ChrimsonR into the visual cortex') into the pDMSt bilaterally in the double transgenic offspring of a cross between the D1–Cre transgenic mouse line and the Adora2a–Cre transgenic mouse line. This led to expression of the inhibitory opsin GtACR2 in both direct and indirect pathway neurons of the pDMSt. We illuminated the pDMSt bilaterally with blue light from a 473-nm laser. The duration of the step-pulse illumination was 1 s and began 5 ms before cue onset. The power of the blue light at the tip of the patch cord was 8 mW. To activate the cue neurons in the visual cortex and avoid any antidromic activation of these visual cortex cue neurons by the blue light in the pDMSt, we expressed soma-targeted ChrimsonR in the cue neurons. ChrimsonR is a red-activatable excitatory opsin. We illuminated the thinned skull over the visual cortex with a red LED coupled to an optical fibre (output power of 35 mW and diameter of the optical fibre of 1 mm). The duration of red light illumination was 0.25 s. We used a constant step pulse of red light to activate the cue neurons. We interleaved the GtACR2-mediated inhibition of the pDMSt on random trials while mice learned to respond to the cue. We aimed to minimize confounds of GtACR2 axonal stimulation by using soma-targeted GtACR2, by using a low light power (8 mW), and by excluding entire sessions if the GtACR2 stimulation led to an increase in cued reaching of more than 20% of the control reach rate (excluded 43 of 87 sessions). We then performed the same trial-to-trial analysis as in Fig. 4. We observed qualitatively the same effects of inhibiting the pDMSt using GtACR2 (Extended Data Fig. 9d) as when we inhibited the pDMSt using ReaChR (Fig. 4).

## Viral injections of pDMSt GtACR2 and visual cortex ChrimsonR

We targeted the pDMSt and visual cortex for injections, as described above. We injected 150 nl of the virus AAV2/8-hSyn-SIO-stGtACR2-FusionRed mixed with 150 nl of the virus AAV2/retro-EF1a-mCherry-IRES-Flpo into the pDMSt bilaterally. We injected 300 nl of this mixture into the pDMSt of each hemisphere. We injected 300 nl of the virus AAV 2/8-EF1a-fDIO-ChrimsonR-mRuby2-KV2.1TS into V1.

## Dopamine fibre photometry in the pDMSt (virus injections)

For the surgery protocol, see the section 'Virus injections surgical details'. We unilaterally injected the pDMSt with AAV9-syn-dLight1.1. We injected the AAV2/retro-EF1a-mCherry-IRES-Flpo into the pDMSt at the same time. We mixed the Flp and dLight viruses in a ratio of 1:1. We then injected 300 nl of this mixture into the pDMSt. We targeted the pDMSt at 0.58 mm posterior, 2.5 mm lateral and 2.375 mm ventral of bregma. We then injected V1 with AAV2/8-EF1a-fDIO-ChrimsonR-mRuby2-KV2.1TS, as described in 'Injection of the AAV carrying Flp-dependent ChrimsonR'. We chose to trigger the optogenetic cue using the red-light-activated ChrimsonR instead of the blue-light-activated opsin ChR2, in the case of these mice for dopamine fibre photometry, because we wanted to avoid any leak of blue light into the dLight1.1 excitation channel. We injected adult mice older than 40 days of age.

## Dopamine fibre photometry in the pDMSt (optogenetic cue)

We activated the ChrimsonR-expressing neurons in the visual cortex as the optogenetic cue. See the section 'Red light optogenetic cue' for details.

## Dopamine fibre photometry in the pDMSt (acquisition setup)

We implanted an optical fibre unilaterally over the pDMSt for dopamine fibre photometry. We implanted this fibre over the pDMSt ipsilateral to the virally expressing cue neurons in the visual cortex, because V1 provides a predominantly unilateral projection to the pDMSt (see the section 'Fibre implants to optically access the pDMSt' for details about the optical fibre implants and targeting of the pDMSt). We coupled the implanted fibre to a Doric Lenses patch cord (0.37 NA). This was coupled to a Doric Fluorescence MiniCube (iFMC5_E1(460-490)_F1(500-540)_E2(555-570)_F2(580-680)_S) for fluorescence imaging. The excitation LED wavelength was band-passed between 460 nm and 490 nm, and the emission light was band-passed between 500 nm 540 nm for green imaging. The MiniCube also enabled red imaging. For red imaging, the excitation LED wavelength was between 555 nm and 570 nm, and the emission light was band-passed between 580 nm and 680 nm. We used the red channel only as an autofluorescence control. Because the heads of the mice were restrained, motion artefacts and artefacts relating to any bending of the patch cord were limited. We modulated the excitation light emitted by the LED. We modulated this light at a constant frequency of 167 Hz, and we sampled the emission light at 2,000 Hz. We used a LabJack T7 to drive the LED and sample data from the photodetector on the Doric MiniCube. We used a custom code in MATLAB to acquire data from and write data to the LabJack T7.

## Dopamine fibre photometry and $Z$-score

We band-passed the collected green light between 120 Hz and 200 Hz (the excitation light was modulated at 167 Hz). Next, we used the MATLAB package Chronux to get a spectrogram. Chronux uses the multi-taper method to calculate the spectrogram. We passed the following parameters to Chronux: (A) moving window of 0.1 s, shifted every 0.01 s to provide a smoothed output, (B) time-bandwidth product of 3, and (C) 2 tapers. Third, we measured the time-varying power to get a representation of the putative dopamine-dependent fluorescence of dLight1.1. We calculated the $Z$-score of this power using a rolling baseline window with a duration of 30 s. We median-filtered this $Z$-score.

## Immunohistochemistry to visualize dLight

We followed the protocol described above in the section 'Immunohistochemistry protocol'. As the primary antibody, we used anti-GFP

from Abcam (#ab13970; concentration 2.5 µg ml⁻¹). As the secondary antibody, we used an anti-chicken antibody conjugated to Alexa 488 from Thermo Fisher (A-11039; concentration 10 µg ml⁻¹).

## Definition of the post-outcome period
A mouse found out whether a reach was successful at the moment when the paw encountered or failed to encounter the food pellet. If the mouse dropped the pellet, the drop typically occured very shortly (less than 0.1 s) after the paw first encountered the pellet. We aligned reaches to the moment when the arm is outstretched. Hence, the outcome was manifest and known around this time point. Thus, we defined the post-outcome period (POP) as the 5-s time window beginning at the outstretched arm.

## Trial type definitions for in vivo physiology analysis
We defined a cued reach as any reach occurring within 3 s of the cue onset. We defined an uncued reach as any reach occurring from 5 s to 16 s after the cue onset, a window that also captures reaches occurring before the onset of the next trial's cue. As mice learned to respond to the cue, cued reaches became restricted to the brief 400-ms window immediately after the cue, but while mice were learning, there was greater variability in the timing of the apparently cued reach. Therefore, we did not analyse reaches between 3 s and 5 s after cue onset, because they were ambiguously either cued at a long delay or uncued. We defined a success as any reach resulting in successful pellet consumption. We defined a failure as any reach not resulting in successful pellet consumption, including cases when the mouse dropped the pellet, reached in a time window when the pellet was missing or reached without dislodging the pellet.

## Training and test sets
We aimed (step 1) to classify neuronal responses into different groups and (step 2) to use these groups to decode the behavioural trial type (that is, cued success, cued failure, uncued success or uncued failure) based on the neural activity (Fig. 5 and Extended Data Fig. 10). To avoid any circular logic or studying noise, we divided the dataset into training and test sets. The training set was a randomly selected 50% of trials acquired for each neuron of each behavioural trial type. For example, if we recorded 50 cued success trials, 40 cued failure trials, 30 uncued success trials and 60 uncued failure trials for neuron 1, then the training set was a random 25 cued success trials, a random 20 cued failure trials, a random 15 uncued success trials and a random 30 uncued failure trials for neuron 1. We used these same trials for all other neurons recorded simultaneously with neuron 1. The test set was the other half of trials. We performed all of step 1 (classification of neurons into different groups) based on the training set only (Extended Data Fig. 10). We then performed all of step 2 (decoding the behaviour based on the neural activity) based on the test set only (Fig. 5j–l). Hence, any patterns detected by the grouping in step 1 are only useful in step 2, if these patterns are consistent across the training and test sets and do not represent noise.

## Two approaches to analyse the SPN activity patterns
We observed that some neurons were more active after a success than after a failure, whereas other neurons were more active after a failure than after a success. To investigate this observation more rigorously, we took two different approaches to organizing the neural activity patterns of the recorded SPNs. Approach 1 was fitting a GLM to the activity pattern of each neuron, followed by clustering of the GLM coefficients (Extended Data Fig. 10a–e). Approach 2 was performing a tensor regression to relate a tensor (or matrix) representing the activity patterns of the neurons to the different behavioural conditions (Extended Data Fig. 10f–l). Both approaches ultimately provided a similar view of the neural data, that is, one group of cells was more active after a success, and a second, different group of cells was more active after a failure,

consistent with our observation by eye. We explain each of these two approaches in greater detail below. We used only trials in the training set for the GLM fitting and tensor regression (see 'Training and test sets').

## Generalized linear model
We built a GLM to analyse how behavioural events predict the neural activity of each recorded neuron. The behavioural events were:
(1) Cue
(2) Distractor LED
(3) Reach (moment of arm outstretched)
(4) Successful outcome (moment of arm outstretched)
(5) Failed outcome is dropped pellet (moment of arm outstretched)
(6) Failed outcome is pellet missing (moment of arm outstretched)
(7) Cued successful outcome (moment of arm outstretched)
(8) Cued failed outcome is dropped pellet (moment of arm outstretched)
(9) Cued failed outcome is pellet missing (moment of arm outstretched)

We binned the neural activity into 0.1-s time bins, and we represented each behavioural event as 1's or 0's across the 0.1-s time bins. We shifted each of the nine behavioural events in time steps of 0.1 s to produce more time-shifted behavioural events (from 2 s before the event to 5 s after the event, $9 \times 71 = 639$ time-shifted behavioural events). We then used a custom code in Python wrapping scikit-learn to find a weight or GLM coefficient (Extended Data Fig. 10a) associated with each of these time-shifted behavioural events. We used a linear link function between the time-shifted behavioural events and the neural activity. To fit the GLM, we used fivefold cross-validation and held out 10% of the data for testing. The resulting GLM coefficients attempted to relate the time-shifted behavioural events to the neural activity. The coefficients associated with each type of behavioural event provide a picture of how that behavioural event predicts neural activity in time. To get the coefficients for a failed outcome, we averaged the coefficients for the two types of failures, (A) dropped pellet and (B) reach to a missing pellet.

Our goal is to find a GLM that is a good fit to the data. We used regularization to prevent overfitting. Regularization adds a penalty that is a function of the magnitude of the GLM coefficients. Hence, with regularization, more parsimonious solutions are preferred. There are different approaches to regularization. We performed a hyperparameter sweep over various regularization parameters to find the regularization parameters resulting in a GLM with the highest $R^2$ regression score function (coefficient of determination):

$$R^2 = 1 - \frac{\text{SS}_{\text{res}}}{\text{SS}_{\text{tot}}}$$

where $\text{SS}_{\text{res}}$ is the sum of squares of residuals after subtracting the model fit, and $\text{SS}_{\text{tot}}$ is the total sum of squares (proportional to the variance of the data). These two regularization parameters were used: α and l1_ratio. At α = 0, this is ordinary least squares, and there is no regularization of the model. At α ≠ 0 and l1_ratio = 0, this is Ridge regression. At α ≠ 0 and l1_ratio = 1, this is Lasso regression. Otherwise, we used ElasticNet (see scikit-learn documentation). We tested α = 0 and all combinations of values for the regularization parameters: α = [0.01, 0.1, 1] and l1_ratio = [0, 0.1, 0.5, 0.9, 1]. We performed this hyperparameter sweep and fit the GLM separately for each neuron. All code is freely available on GitHub (https://github.com/kimerein/k-glm).

## Clustering the GLM coefficients in the POP
To study whether there is neural activity in the striatum that represents both the reach outcome and its context, we considered the GLM coefficients assigned to the POP (Extended Data Fig. 10b). The POP is the time period after the arm is outstretched (see 'Definition of the post-outcome period') and continuing for 5 s. We took the GLM coefficients from 0 s to 5 s for each of these four behavioural events:

(1) Successful outcome (success)
(2) Failed outcome (failure)
(3) Cued successful outcome (cue × success)
(4) Cued failed outcome (cue × failure)

We called the POP coefficients for each of these behavioural events a 'kernel'. We smoothed the kernels with a 0.08-s time bin, then max-normalized the kernels. Note that there are now four kernels per neuron. We concatenated the four kernels to make a data vector for each neuron. We considered only neurons with POP coefficients greater than zero. (The excluded neurons had GLM coefficient assignments related to other behavioural events, for example, the cue, or GLM coefficient assignments before the POP period but no GLM coefficient greater than zero in the POP period for the four behavioural events listed above.) Finally, we performed $k$-means clustering of these vectors to partition them into two clusters (see $t$-distributed stochastic neighbour embedding (t-SNE) with labels 'Clust 1' and 'Clust 2' in Extended Data Fig. 10d). For visualization purposes only, we plotted these two clusters in a low-dimensional space using t-SNE in MATLAB (t-SNE parameters: Euclidean distance, perplexity = 150; Extended Data Fig. 10d).

## Setting up the tensor regression
We used only the training set to train the regression and later validated using the test set. The goal of the tensor regression (Extended Data Fig. 10f–l) was to predict the behavioural condition (that is, cued success, cued failure, uncued success or uncued failure) from the neural activity. The model can be considered a multilinear (3D) reduced-rank multinomial regression. We attempted to predict the current behavioural condition from the neural activity of the 1,000 SPNs. Typically, there is not a unique solution to this problem, so model comparison was used to choose a rank for the model. Backpropagation via the ADAM optimizer was used to optimize the coefficient weights.

Furthermore, we aimed to find interpretable patterns in the data. Hence, we searched for a regression that could be decomposed into a low-rank sum of rank−1 outer products (that is, a Kruskal tensor). Thus, we searched for a low-dimensional representation that captures the major features of the relationship between the behavioural condition and the neural data. The low dimensionality of this representation or decomposition simplifies our interpretation of the regression and improves the interpretability of the solution found by the optimization algorithm.

We set up the regression as follows. For simplicity, we trial-averaged the responses of each neuron within each of the four behavioural conditions (Extended Data Fig. 10f):
(1) Successful outcome (success)
(2) Failed outcome (failure)
(3) Cued successful outcome (cued success)
(4) Cued failed outcome (cued failure)

We then time-shifted the failure responses (2 and 4 above) to align the timing of the dopamine dip after a failure (approximately 1.6 s after the arm outstretched) to the timing of the dopamine peak after a success (approximately 0.83 s after the arm outstretched). Although dopamine was measured in a separate group of mice by dLight fibre photometry, we observed that the timing of the post-success dopamine peak and post-failure dopamine dip were quite consistent across mice (not shown). Therefore, we chose the timing of the peak or dip from the averaged data across mice and used those time points to shift the neural data before the tensor regression.

We did not have a trial dimension, because we trial-averaged. For each behavioural condition, there were $N$ neurons by $T$ time points. Putting together the four behavioural conditions, we ended up with a 3D matrix with dimensions, $N$ neurons by $T$ time points by $C$ behavioural conditions (Extended Data Fig. 10g). This 3D matrix, or tensor, was the input to the regression.

We performed a multinomial logistic regression, because we are trying to predict a categorical variable, not a numeric variable, in this case. The categorical variable is the behavioural condition (that is, cued success, cued failure, uncued success or uncued failure). We used custom code wrapping PyTorch in Python to regress the behavioural condition on the input matrix. The output of the model is in the form of a Kruskal tensor, that is, a set of components, where each comprised three 1D vectors, or factors: an $N$-dimensional, $T$-dimensional and $C$-dimensional vector. Taking the outer product of each set of vectors and summing the resulting 3D arrays makes a rank-$R$ beta weight tensor. The inner product of the input tensor with this beta weight tensor produces the output logits for the multinomial regression model. Vectors in the Kruskal tensors can be thought of as the weights, or loadings. By considering these vectors, we can observe the loadings onto each modality (that is, neurons (Extended Data Fig. 10j, left), time points (Extended Data Fig. 10j, middle) and behavioural conditions (Extended Data Fig. 10j, right)). We enforced a non-negativity constraint on the optimized Kruskal tensor weights corresponding to the neuron vectors (that is, factors) only. The other two vectors (that is, factors for time points and behavioural conditions) were allowed to be positive, negative or zero valued. The final tensor regression model was selected to be of rank 2 and thus produced 2 components (see 'Selecting the rank of the tensor regression'). One component was associated with a specific pattern of activity after a success versus failure. The second component was associated with a different pattern of activity after a success versus failure. These two components tended not to share neurons (Extended Data Fig. 10j, left), suggesting that they represented two different groups of cells. All code is freely available on GitHub (https://github.com/kimerein/tensor_regression).

## Tensor regression optimization
We randomly initialized the $N$ neurons by $T$ time points by $C$ behavioural conditions tensor, which represents the regression (see 'Setting up the tensor regression'), by sampling the parameters from the uniform distribution between 0 and 1, scaled by a constant. This constant is a hyperparameter called Bcp_init_scale in the code (see https://github.com/kimerein/tensor_regression). We set Bcp_init_scale to 0.625. We then optimized the tensor, using a learning rate of 0.007 and minimizing the cross-entropy loss using the ADAM optimizer (see torch.nn.CrossEntropyLoss and torch.optim.Adam), until convergence.

## Tensor regression regularization
We used Ridge (L2) regularization, which adds a penalty proportional to the squared magnitude of the parameters. This penalty is added to the loss function, which the optimization attempts to minimize (see 'Tensor regression optimization').

## Selecting the rank of the tensor regression
Before running the optimization, we must manually select the rank, or number of components, of the tensor regression (Extended Data Fig. 10h,i). The rank can be thought of as roughly analogous to the number of components in principal components analysis or reduced-rank regression. To choose the rank, we re-ran the tensor regression optimization many times, obtaining a solution with a different rank each time. We re-ran the tensor regression optimization ten times for each of the following ranks: 1, 2, 3, 4 and 5. We present the results in Extended Data Fig. 10h,i. First, we found that the loss (we used the cross-entropy loss; see torch.nn.CrossEntropyLoss) was not much worse when the solution was a two-rank solution versus a three-rank, four-rank or five-rank solution (Extended Data Fig. 10i). Therefore, we chose to present a two-rank solution (Extended Data Fig. 10i, arrows), which is simpler to interpret.

## Choosing a specific tensor regression solution
Next, we considered the ten different, two-rank solutions produced by running the tensor regression optimization ten times. We noticed that

one solution loaded the two components onto two different and largely non-overlapping groups of neurons. We measured the overlap as the 'joint loading penalty', $J$, defined as the pairwise sum of factor loadings onto the same neuron over the pairwise difference of factor loadings onto the same neuron (Extended Data Fig. 10j), that is,

$$J = \frac{\sum_{n \in N} \sum_{i,j \in F} |w_{n,i} + w_{n,j}|}{\sum_{n \in N} \sum_{i,j \in F} |w_{n,i} - w_{n,j}|}$$

where $n$ is a neuron in the set of neurons $N$; $i, j$ are pairs of factors in the set of factors $F$ given $i \neq j$; and $w_{n,i}$ is the loading (or weight) of factor $i$ onto neuron $n$ for the $N$-dimensional 'neuron' vector component of the Kruskal tensor. Note that $w_{n,i}$ is always positive, as described above ('Setting up the tensor regression'). Hence, as the response of a neuron is described more unevenly by the different factors belonging to different components, the penalty $J$ decreases. We chose the solution to the tensor regression optimization that minimized $J$. This was a solution that loaded the neuron factors of two components onto two largely separate and non-overlapping groups of neurons (Extended Data Fig. 10h, arrow). Note that this solution also utilized the two components equally overall, as measured by the 'component weight', that is, the sum of the absolute value of all mean-subtracted parameter weights (Extended Data Fig. 10i, arrows).

## Validation of tensor regression

The tensor regression describes the relationship between the neural data and the behaviour based on the training set. To validate our tensor regression, we asked whether this solution is useful to describe the relationship between the neural data and the behaviour for the test set. The test set contains a set of trials independent from the training set. We used the tensor regression to predict behavioural successes versus failures from the neural activity of the test set. The regression correctly predicted behavioural successes versus failures for the test set (Extended Data Fig. 10k), suggesting that there is something detected by the regression that is consistent across the training and test sets. We shuffled the neuron ID, and this markedly degraded the prediction. We shuffled the time points, and this dramatically degraded the prediction. Shuffling both neuron ID and time points further degraded the prediction (Extended Data Fig. 10l).

## The simpler approach to the neuron groups 1 and 2

Both approaches (approach 1: clustering GLM coefficients, and approach 2: tensor regression) produced two groups of neurons, which have different response properties. We analysed these two groups of neurons, populating all parts of Fig. 5, based on each approach, and we found that either approach (approach 1: clustering GLM coefficients, or approach 2: tensor regression) produced qualitatively similar results (not shown). However, we decided to use a simpler approach (Extended Data Fig. 10m,n) to separate the neurons into two groups for our presentation in Fig. 5. All approaches revealed consistent structure in the data that was able to predict the behaviour from the neural activity. We arrived at this simpler approach as follows. We observed that component 1 from the tensor regression indicated higher activity that tends to decrease after a success (Extended Data Fig. 10j). We captured this pattern using the 'modulation index' after a success (Extended Data Fig. 10m). The modulation index, $m$, was defined as

$$m = \frac{c_{2\,to\,5s} - c_{0\,to\,2s}}{|c_{2\,to\,5s}| + |c_{0\,to\,2s}|}$$

where $c_{2\,to\,5s}$ is the average GLM coefficient from 2 s to 5 s after the arm is outstretched, and $c_{0\,to\,2s}$ is the average GLM coefficient from 0 s to 2 s after the arm is outstretched. For a success, we calculated $m_{success}$ for the success GLM coefficients and $m_{cued\,success}$ for the cued success GLM coefficients. We averaged $m_{success}$ and $m_{cued\,success}$ to get the modulation index after a success, presented in Extended Data Fig. 10m,n.

We also observed that component 2 from the tensor regression indicated slightly increasing and sustained activity after a failure (Extended Data Fig. 10j). We captured a pattern of sustained modulation after a failure using the 'sustained metric' (Extended Data Fig. 10m). The sustained metric, $s$, was defined as

$$s = |c_{1\,to\,5s}|$$

where $c_{1\,to\,5s}$ is the average GLM coefficient from 1 s to 5 s after the arm is outstretched. We calculated $s_{failure}$ for the failure GLM coefficients and $s_{cued\,failure}$ for the cued failure GLM coefficients. We averaged $s_{failure}$ and $s_{cued\,failure}$ to get the sustained metric after a failure, presented in Extended Data Fig. 10m,n. The $k$-means clustering of GLM coefficients produced a division that qualitatively matched these observations (see purple versus cyan dots representing neurons in Extended Data Fig. 10n, top). For simplicity, we decided to just draw a line that separated the purple neurons from the blue neurons in Extended Data Fig. 10n, bottom. We used this line to divide neurons for the analysis presented in Fig. 5. Both of the more complicated approaches (that is, clustering GLM coefficients and tensor regression) motivated our decision to use this line (and not some other boundary) to separate the neurons in Fig. 5 into two groups. Only the data in the training set was used to draw the separation boundary in Extended Data Fig. 10n, bottom, whereas conclusions about its utility were drawn from its application to the test set.

## Decoding the behaviour from average unit firing rates

We used only the test set to attempt to decode trial identities (Fig. 5k). To determine whether the neural activity of SPNs in the POP encodes the four behavioural conditions, that is, cued success, cued failure, uncued success or uncued failure, we measured, in each of these behavioural conditions separately, the trial-averaged firing rate of each SPN over the time window 1–5 s after the outstretched arm. We excluded the 1-s window immediately after the outstretched arm to ensure that the cue offset precedes the analysed time window by more than 0.75 s (Fig. 5i). We were not interested in the immediate cue-evoked response but rather whether the cue information continues to be represented after the outcome is known. We considered neurons belonging to either group 1 or group 2, as classified by the methods described above using only the training set for the classification. We ran a bootstrap with 100 iterations to plot how group 1 versus group 2 neuronal firing rates represent the four behavioural conditions (Fig. 5k). At each iteration of the bootstrap, from the group 1 neurons, we randomly sub-sampled $n$ neurons with replacement, and from the group 2 neurons, we randomly sub-sampled $n$ neurons with replacement. We then averaged the firing rates of all group 1 neurons and plotted this as the value along the $y$ axis in Fig. 5k. We averaged the firing rates of all group 2 neurons and plotted this as the value along the $x$ axis in Fig. 5k. There were four behavioural conditions for each sub-sampled set of $n$ neurons. Hence, the 400 points in Fig. 5k represent the average firing rates of group 1 versus group 2 neurons, for each of the behavioural conditions. We found that this mapping, at least partially, separated the cued successes from uncued successes, and both success types from failures. To quantify the quality of this separation, we used linear discriminant analysis (LDA) to attempt a three-way separation of behavioural conditions (cued success versus uncued success versus failure) based on the points in Fig. 5k. We measured the accuracy of the LDA prediction. Higher prediction accuracies indicated better separation. We reported the accuracy of the LDA prediction for different numbers of neurons sub-sampled, $n$ (Fig. 5k, bottom-right).

## Shuffled average unit firing rates

To determine whether the separation of neurons into groups 1 and 2 provides any meaningful information, we took all neurons identified

as belonging to group 1 or group 2, then shuffled the identities of these neurons before attempting the decoding of the behavioural condition from the neural activity. Figure 5k, top right, shows what happens as a result of this shuffling. Note that successes, and, in particular, the uncued success, are no longer separable from failures. The shuffle decreased the separation of the four behavioural conditions and the quality of the decoding. This indicates that the assignment of neurons into groups 1 or 2 provides added information that helps to decode the current behavioural condition. However, note that some information remains in the activity of all the neurons combined (along the diagonal $y = x$ in Fig. 5k, top right). We also performed a second type of shuffle. For this second shuffle, we maintained the unit identities but shuffled the average firing rates with respect to the behavioural conditions. For example, if neuron 1 had average firing rates of 0.5, 2, 4 and 0 spikes per second for the four behavioural conditions of cued success, cued failure, uncued success and uncued failure, respectively, then after shuffling, neuron 1 had average firing rates of 4, 0, 0.5 and 2 spikes per second for the four behavioural conditions of cued success, cued failure, uncued success and uncued failure, respectively. As expected, this second shuffle also disrupted the decoding of the current behavioural condition (Fig. 5k, bottom-right).

## Decoding the behaviour from single-trial firing rates

We used only the test set to attempt to decode trial identities (Fig. 5l). As described above, the average firing rates of the neurons could be used to decode the behavioural condition (that is, cued success, cued failure, uncued success and uncued failure). To test whether single-trial firing rates provided sufficient information to perform similar decoding, we measured the firing rate of each neuron on each trial averaged over the time window 1–5 s after the outstretched arm. We ran a bootstrap with 100 iterations. We randomly sub-sampled $n$ neurons with replacement from the group 1 neurons, and we randomly sub-sampled $n$ neurons with replacement from the group 2 neurons. Then, we randomly sampled one single trial from each unit, for each behavioural condition. For each behavioural condition, we averaged the $n$ single trials. We plotted the average activity from neurons belonging to group 1 on the $y$ axis (Fig. 5l), and we plotted the average activity from neurons belonging to group 2 on the $x$ axis (Fig. 5l). Therefore, there are 100 points plotted (100 bootstrap iterations) for each behavioural condition. We used LDA to attempt a three-way separation of these points based on the behavioural condition (cued success versus uncued success versus failure). We plotted the accuracy of the LDA prediction of the behavioural condition, as a function of the number of trials sub-sampled (Fig. 5l, bottom-right).

## Shuffled single-trial firing rates

First, we shuffled the identities of the group 1 and group 2 neurons, before attempting to decode the behavioural condition from neural activity (Fig. 5l, top right). This disrupted the decoding. Second, we randomly permuted the time window-averaged firing rates of single trials with respect to the behavioural conditions of those single trials (Fig. 5l, bottom right). This shuffle also disrupted the decoding.

## Reporting summary

Further information on research design is available in the Nature Portfolio Reporting Summary linked to this article.

## Data availability

Summary datasets are available at https://dataverse.harvard.edu/dataset.xhtml?persistentId=doi:10.7910/DVN/QPQEC9. Example datasets for running the code at this same location are also available. Because the total amount of raw data is over 10 TB, and this volume is not well supported by the Harvard Dataverse, we have not uploaded all raw data to the Harvard Dataverse, but any raw data will be made freely available on request to the corresponding authors. Source data are provided with this paper.

## Code availability

All custom codes are freely available on GitHub, as listed below. An explanation of the top-level scripts, along with example datasets to run the code are available at https://dataverse.harvard.edu/dataset.xhtml?persistentId=doi:10.7910/DVN/QPQEC9. The MATLAB analysis code: https://github.com/kimerein/integrate-phys-and-beh and https://github.com/kimerein/KR_Analysis_Toolbox; the Python GLM code: https://github.com/kimerein/k-glm; the automated analysis of reaching in low-speed video: https://github.com/kimerein/reach-behavior-analysis and https://github.com/kimerein/reach-Behavior; the multi-unit processing of data from Plexon and WHISPER systems: https://github.com/kimerein/MU-analysis; the Python tensor regression: https://github.com/kimerein/tensor_regression; photometry acquisition: https://github.com/kimerein/photometry; the Arduino code: https://github.com/kimerein/behaviorRig; Python alignment of the high-speed video to events in behaviour: https://github.com/kimerein/integrate-phys-and-beh; and UltraMegaSort spike sorting: https://github.com/kimerein/Mat_Code/tree/master/.

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

**Acknowledgements** This work was funded by the US National Institutes of Health (U19NS113201 to B.L.S. and K99MH127471 to K.R.) and a Helen Hay Whitney Foundation postdoctoral fellowship (to K.R.). We thank the Sabatini laboratory and Neurobiology department at Harvard Medical School for helpful discussions.

**Author contributions** K.R. and B.L.S. designed the experiments. K.R. conducted the analyses. K.R. designed the behavioural rig. K.R., M.I., S.T., A.C., S.S., W.K. and M.A.A. trained the mice. K.R., M.I., S.T., A.C., S.S., W.K. and M.A.A. performed the surgeries. K.R., M.I., S.T., A.C. and W.K. processed the data. K.R. performed the in vivo electrophysiology. K.R. and M.I. performed the dopamine fibre photometry. W.W. performed the in vitro electrophysiology. J.Z. wrote a GLM analysis package. R.H. wrote a tensor regression analysis package. K.R. and B.L.S. wrote the manuscript.

**Competing interests** The authors declare no competing interests.

**Additional information**
**Correspondence and requests for materials** should be addressed to Bernardo L. Sabatini.

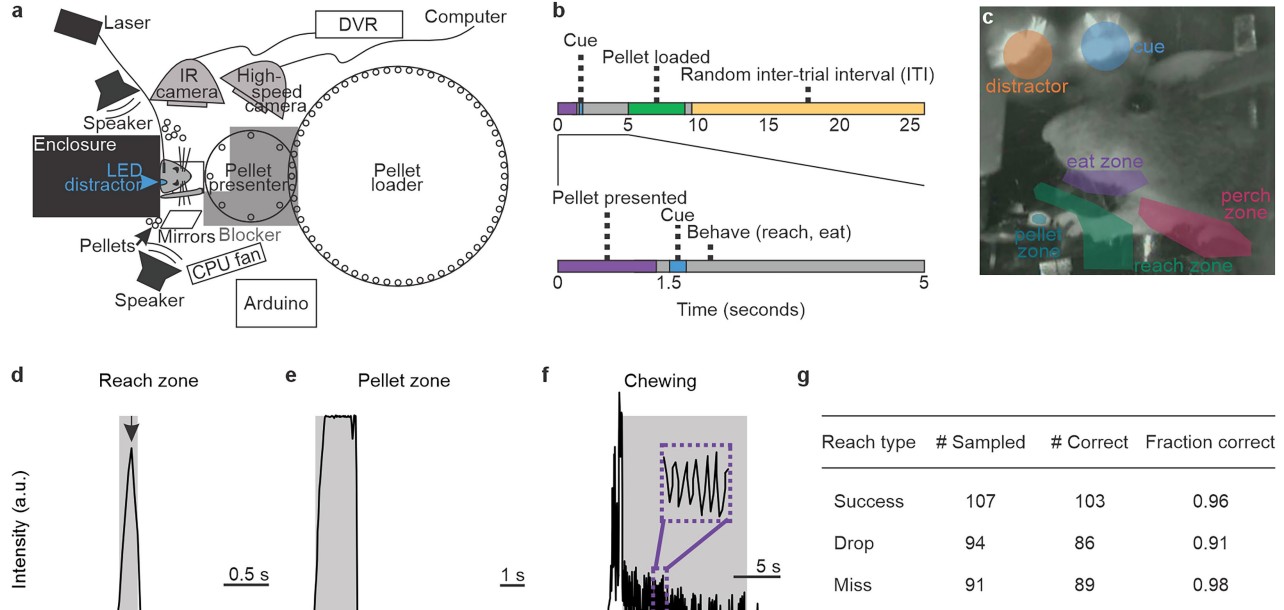

**Extended Data Fig. 1 | Behavior paradigm pairs optogenetic activation of corticostriatal neurons in the visual cortex with presentation of a food pellet obtained by a forelimb reach. a**, Automated rig to train mice. Mice are head-fixed at a short distance from the food pellet. Food pellets are presented and loaded automatically using stepper motors controlled by an Arduino. Arduino also controls the timing of the LEDs and lasers for optogenetic stimulation, triggers the LED distractor, and triggers high-speed video acquisition. Two cameras: one labeled infra-red (IR) camera for low-speed, continuous video acquisition, and one for high-speed 255 frames per second (fps) video acquisition triggered at the beginning of each trial. Speaker masks the sound of the stepper motors. CPU fan obscures the smell of the approaching food pellet. Other food pellets mask the smell of the approaching food pellet. Mirrors are positioned below and to the side of the mouse, enabling high-speed 3D tracking of the paw position using DeepLabCut. Entire rig is enclosed in large light-tight box to prevent the mouse from seeing food pellets. Inside of the box is pitch-black. **b**, Trial structure: Pellet moves into position in front of the mouse over 1.28 s. Following a 0.22-s delay, cue turns on. 8 s later, pellet moves out of reach. Future food pellets are loaded onto the back of the pellet presenter disk. Random inter-trial interval (ITI) ranges from 0 to 16.5 s (Methods, "Training mice to associate a cue with the food pellet"). **c**, Analysis of low-speed video to monitor behavior events. Zones are drawn onto the video by user. Behavior events identified by signal processing of intensity signals within these zones (Methods, 'Processing the 30-fps video'). **d-f**, Example signals from zones in **c**. Intensity in arbitrary units. **d**, Intensity increases when forelimb enters reach zone. **e**, Intensity increases when pellet enters pellet zone. **f**, Chewing produces periodic signal at ~7 Hz in chewing zone. **g**, Accuracy of automated classification of reach outcomes. "Correct" as compared to human classifier.

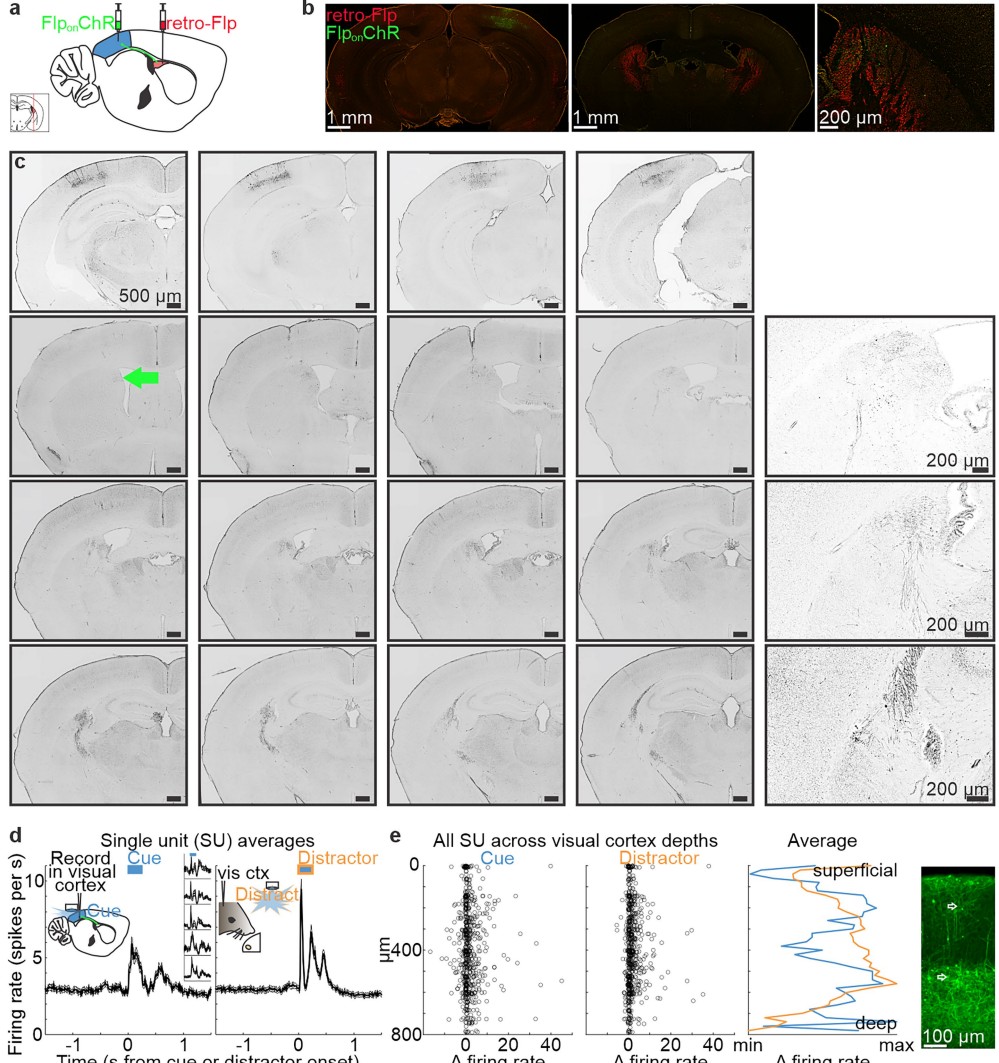

**Extended Data Fig. 2 | Optogenetic activation of corticostriatal neurons in visual cortex serves as the cue. a**, Virus injections to retrogradely label striatum-projecting neurons in visual cortex, called the cue neurons (Methods, 'Virus injection'). **b**, Histology of example mouse injected with AAV/retro carrying Flp recombinase (red, Flp) into pDMSt bilaterally and injected with AAV carrying FlpOn Channelrhodopsin2 (green, ChR) into the visual cortex unilaterally. Green axons visible in pDMSt. **c**, Cue neuron cell bodies in visual cortex (top row) and projection pattern of cue neuron axons in striatum (next 3 rows, with close-ups at right). Green arrow shows anterior-most extent of cue neuron axons in only the medial-most part of striatum. **d**, Recordings in visual cortex to verify optogenetic activation of ChR-expressing cue neurons. *left*, Average±s.e.m firing rate of all single units (SU) measured by multi-channel extracellular electrophysiology in visual cortex (n = 640 SU from 5 mice). Blue bar represents the duration of LED illumination of visual cortex through a

thinned skull. Insets are data from different individual mice over same X axis time window (Y axis range: 0–7 Hz, 0–12 Hz, 0–12 Hz, 2–9 Hz, 1–11 Hz, from top to bottom). *right*, Response of same neurons to the LED distractor, which is an external visual stimulus with the same blue color and duration as the cue. **e**, Cue- (*left-most panel*) or distractor- (*middle-left panel*) evoked change in firing rates of all individual SU across layers of the visual cortex, ordered from superficial to deep. Change in firing rate is the average firing rate over 0.25 s just after the cue minus the average firing rate over 2 s just before the cue in spikes per s (or aligned to distractor onset). (*middle-right panel*) Average cue- (blue) or distractor- (orange) evoked change in SU firing rate across depths, as min-subtracted, max-normalized and smoothed by 20 µm bin. (*right-most panel*) Close-up of visual cortex (V1) histology showing ChR-expressing cue neurons in layers 5 and 2/3 (white arrows point to example cells).

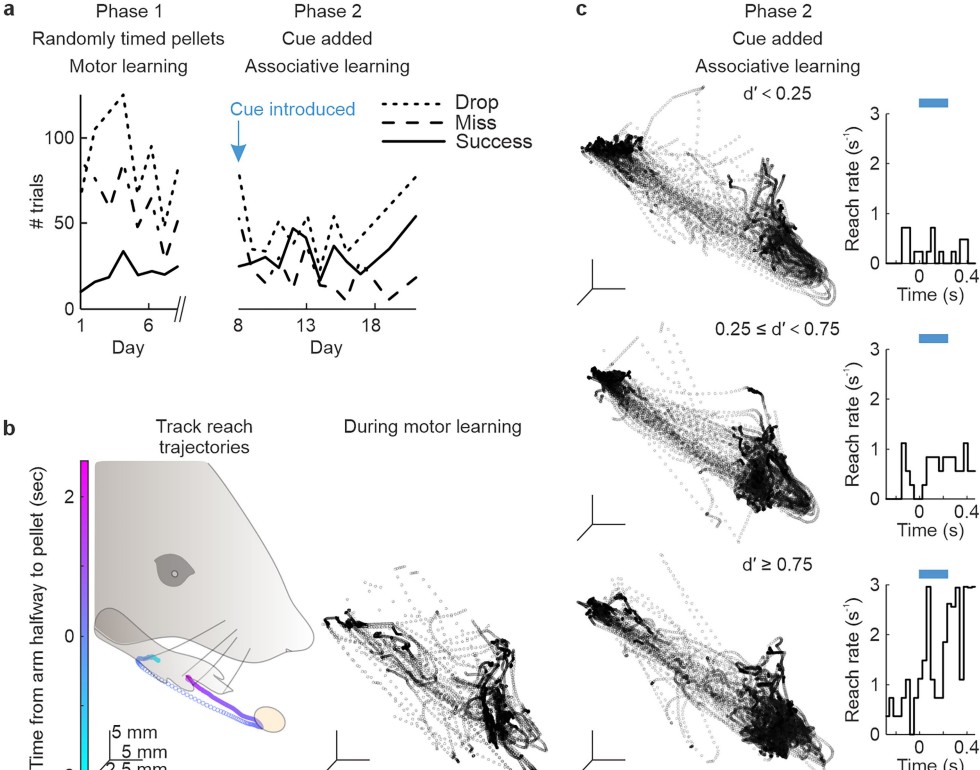

**Extended Data Fig. 3 | Two distinct phases of learning, motor learning in Phase 1 and associative learning in Phase 2.** In Phase 1, we train hungry mice to perform stereotyped forelimb reaches to obtain the food pellet. In this phase, food pellets are presented at random times. In Phase 2, we train mice to associate the cue with the presentation of the food pellet. Hence, in this phase, mice learn to associate the cue with the forelimb reach. **a**, Reach outcomes from an example mouse over Phases 1 and 2. Each trial is one cue presentation. Drop means the mouse dislodged the pellet but failed to consume it. Miss means the mouse reached but did not touch the pellet. Success means the mouse successfully grabbed and consumed the pellet. Failures (drops and misses) decrease during Phase 1 motor learning. No further improvements in success rate in Phase 2. **b**, 3D paw tracking at 255 frames per second (fps). *left*,

Average trajectory of reaches from Phase 2 sessions from an example mouse (n = 412 reaches from 3 sessions). All reaches aligned to the time when the forepaw is part-way to the pellet during the initial ballistic movement of the forelimb toward the pellet. *right*, Example single reaches during Phase 1 from the same example mouse, showing variable trajectories and a non-stereotyped reach. **c**, Reaches from same example mouse as **b** during Phase 2 after pairing the cue with the food pellet. *left*, Example single reach trajectories overlaid. *right*, Reach rate over time aligned to the cue (blue bar represents the cue). *top to bottom*, Each row is a different example session from beginner, intermediate, and expert stages of learning about the cue. Note no further refinement of reach trajectories, despite the mouse shifting the timing of the reach to the time window immediately after the cue.

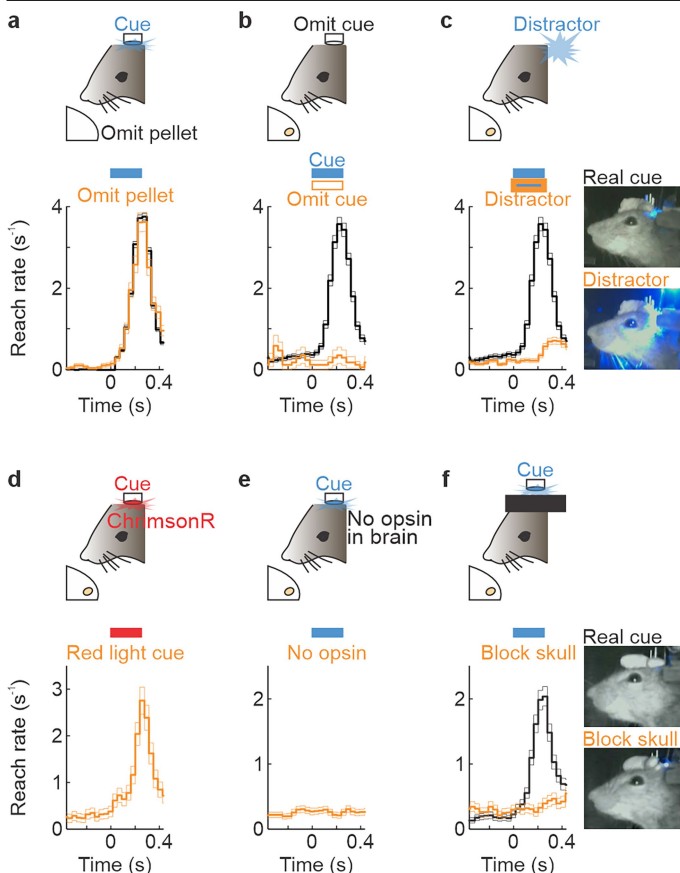

**Extended Data Fig. 4 | Mice attend to the optogenetic cue.** All panels show reach rate as in Fig. 1. **a**, Omit the food pellet, but present the cue. Black: Pellet presented. Orange: Pellet omitted on random trials. n = 11805 black trials, 1637 orange trials from 18 mice. We excluded trials when the mouse dislodged the pellet before the cue. **b**, Omit the cue, but present the food pellet. Black: Cue turns on. Orange: Cue omitted on random trials. n = 3268 black trials, 246 orange trials from 18 mice. **c**, Compare reaching in response to the cue with reaching in response to the distractor LED. Black: Aligned to cue. Orange: Aligned to distractor LED. n = 3268 black trials, 3268 orange trials from 18 mice. Video frames at right show that distractor LED is brighter than real cue. **d**, Response to a red light cue in mice expressing the red-activatable opsin ChrimsonR in visual cortex. Poor visual detection of red light in mice, yet the mice still learn to respond to the optogenetic cue. n = 862 orange trials from 3 mice. **e**, Response to the blue light cue when the visual cortex does not express the activating opsin Channelrhodopsin2, ChR. Orange: Aligned to the cue, from mice that lack ChR in visual cortex. n = 3225 orange trials from 4 mice. **f**, In mice trained to respond to the blue light optogenetic cue, block the thinned skull to prevent blue light from accessing the brain. Video frames at right show that the blue light turns on but does not penetrate the blocked skull. Black: Control day before blocking the skull. Orange: The next day when we blocked the skull. n = 2357 black trials, 1733 orange trials from 18 mice.

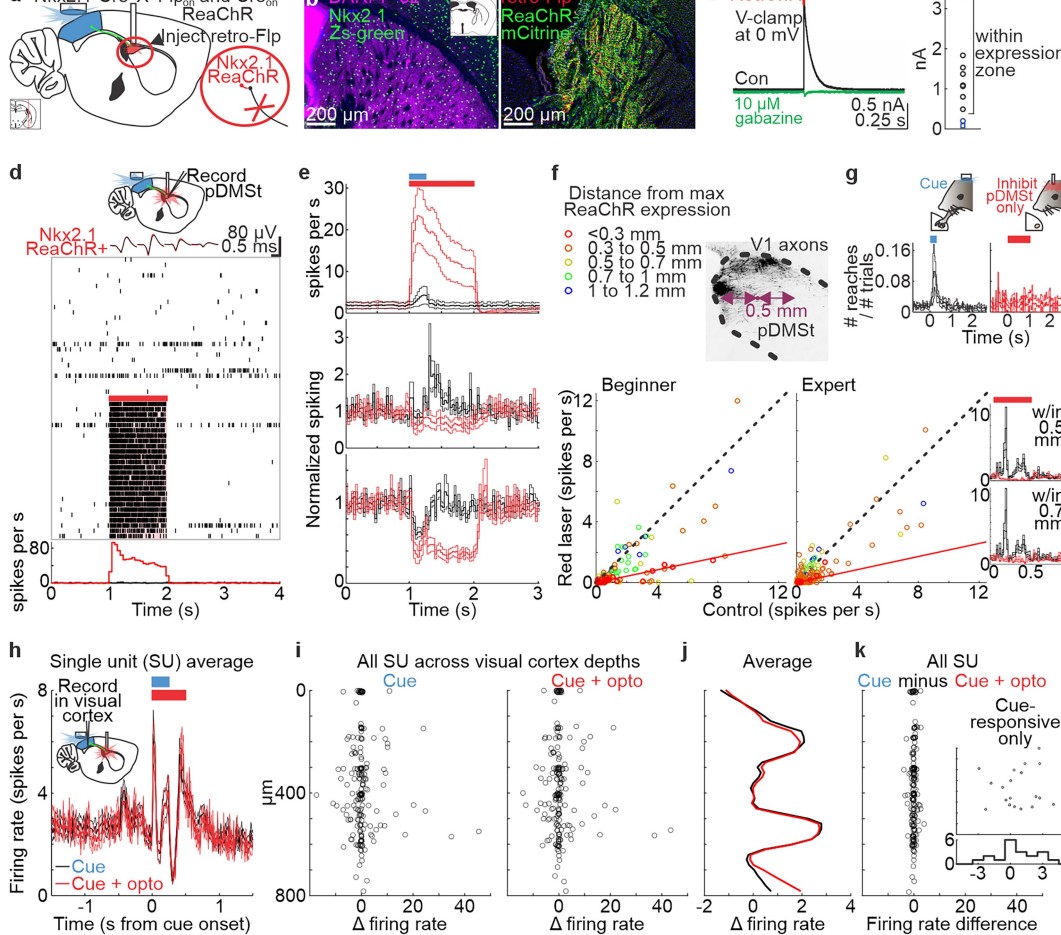

**Extended Data Fig. 5** | See next page for caption.

**Extended Data Fig. 5 | Method to optogenetically inhibit pDMSt.**
**a**, Schematic sagittal section of mouse brain at medial-lateral position shown by red line in inset box at bottom-left. Injections into a double transgenic mouse expressing Cre in Nkx2.1+ striatal interneurons and red-activatable Channelrhodopsin (ReaChR), where ReaChR expression is conditional on Cre recombinase and Flp recombinase being present in the cell. Thus, Flp injections into pDMSt produce ReaChR expression only in the Nkx2.1+ striatal interneurons of pDMSt. Close-up circle: Striatal interneurons (red) project to and inhibit the striatal projection neurons (black), which represent the sole output of pDMSt. Hence, red light-mediated activation of striatal interneurons is expected to suppress pDMSt output. **b**, *left*, Example coronal section of pDMSt showing immunohistochemistry for DARPP-32 marker of striatal projections neurons (pink) in a Cre-dependent Zs-green reporter transgenic mouse line that expresses Zs-green in the Nkx2.1+ striatal interneurons (green). *right*, Expression of ReaChR-mCitrine (green) that results from AAV/retro Flp-mCherry injection (red) into pDMSt of double transgenic mouse line described in **a**. **c**, Acute in vitro slice electrophysiology to test whether ReaChR activation of Nkx2.1+ striatal interneurons produces inhibitory synaptic transmission onto striatal projection neurons (SPNs). *left*, Example whole-cell voltage-clamp (V-clamp) recording from putative SPN in pDMSt. Black: Average outward current aligned to red light illumination of slice expressing ReaChR in Nkx2.1+ striatal interneurons. Green: Gabazine block suggests that ReaChR-evoked outward current is GABAergic. *right*, Summary of short-latency, likely monosynaptic outward currents across all putative SPNs (n = 10) patched within (black) or outside of (blue) ReaChR expression zone. Blue square was a cell with outward current delayed by 10 ms (not putative monosynaptic). **d**, In vivo multi-channel extracellular electrophysiology in pDMSt to test optogenetic inhibition of pDMSt. *top*, Schematic showing recording in pDMSt from awake mice experiencing blue light-mediated activation of visual cortex cortico-striatal neurons as the cue and red light-mediated activation of striatal interneurons to inhibit the striatal output neurons. *middle*, Spike waveforms from 4 neighboring electrode channels from an example red light-activated single unit, indicating no difference in that unit's spike waveform when the red laser was on (red) or off (black). Raster plot shows rows of vertical lines indicating spiking activity. Each row is aligned to red light onset. Each line is a spike. Red light trials were randomly interleaved during the experiment but are separated here for visual clarity. Red bar shows duration of red laser illumination of pDMSt. *bottom*, Peri-stimulus time histogram (PSTH) illustrating the average activity of this example neuron in control trials (black) versus trials with red laser (red). **e**, Mean±s.e.m. across single units in pDMSt measured by in vivo electrophysiology. Cue onset at 1 s (blue bar shows cue duration). Red bar shows duration of red laser illumination of pDMSt. *top*, Units enhanced by red light (n = 17 from 6 mice from sites within 0.7 mm of peak of ReaChR expression). *middle*, Units that increased their activity after the cue (n = 17 from 6 mice from sites within 0.7 mm of peak of ReaChR expression). *bottom*, All other units (n = 99 from 6 mice from sites within 0.7 mm of peak of ReaChR expression). **f**, Spiking activity of single units in pDMSt from 8 beginner and 2 expert mice comparing control conditions (X axis) to activity during red laser illumination of pDMSt (Y axis). Units below the dotted line were suppressed by red light. Colors indicate the distance of the recording site from the peak of ReaChR expression, determined post-mortem by comparing the dye-labeled electrode recording track to the expression of ReaChR-mCitrine in fixed post-mortem slices. Histology at top-right shows example visual cortex (V1) axons in pDMSt, for reference. Note that the spread of pDMSt inhibition measured empirically matches well with the spread of V1 axons in pDMSt. PSTH insets to right of expert plot show mean±s.e.m. firing rate of all single units within 0.5 mm or 0.7 mm of peak of ReaChR expression from 2 expert mice (X axis unit is seconds). **g**, Behavior in cue-trained mice (n = 3) comparing reaching in response to the cue (*top*) versus reaching in response to the optogenetic inhibition of pDMSt, in the absence of the cue (*bottom*). **h**, Recordings in visual cortex during red light illumination of pDMSt. Inset schematic shows recording in visual cortex while mice behave and experience red light illumination of pDMSt. Plot shows the average±s.e.m firing rate across all single units (SU) recorded in visual cortex, as in Extended Data Fig. 2d (n = 196 from 3 mice). Trials with cue plus illumination of pDMSt (red) are overlaid on control trials with cue only (black). Cue onset at 0 s (blue bar shows cue duration). Red bar shows duration of red laser illumination of pDMSt. Note no difference in the activity of the visual cortex, with or without the red laser illumination of pDMSt. **i**, Change in firing rate of SU (n = 196 from 3 mice) in visual cortex, as in Extended Data Fig. 2e. Change in firing rate is average firing rate over 0.25 s just after the cue minus the average firing rate over 2 s just before the cue in spikes per s. *left*, Trials with cue only. *right*, Trials with cue plus red laser illumination of pDMSt. **j**, Average cue-evoked change in SU firing rate as in Extended Data Fig. 2e, but here black is the response to the cue only and red is the response to the cue plus red laser illumination of pDMSt. **k**, Comparing the two panels in **i** across all SU in visual cortex. Inset scatter: Firing rate difference for the cue-responsive units only. Inset histogram: For cue-responsive units only, the change in rate when the red laser illumination of pDMSt was added. Note only small changes distributed around zero.

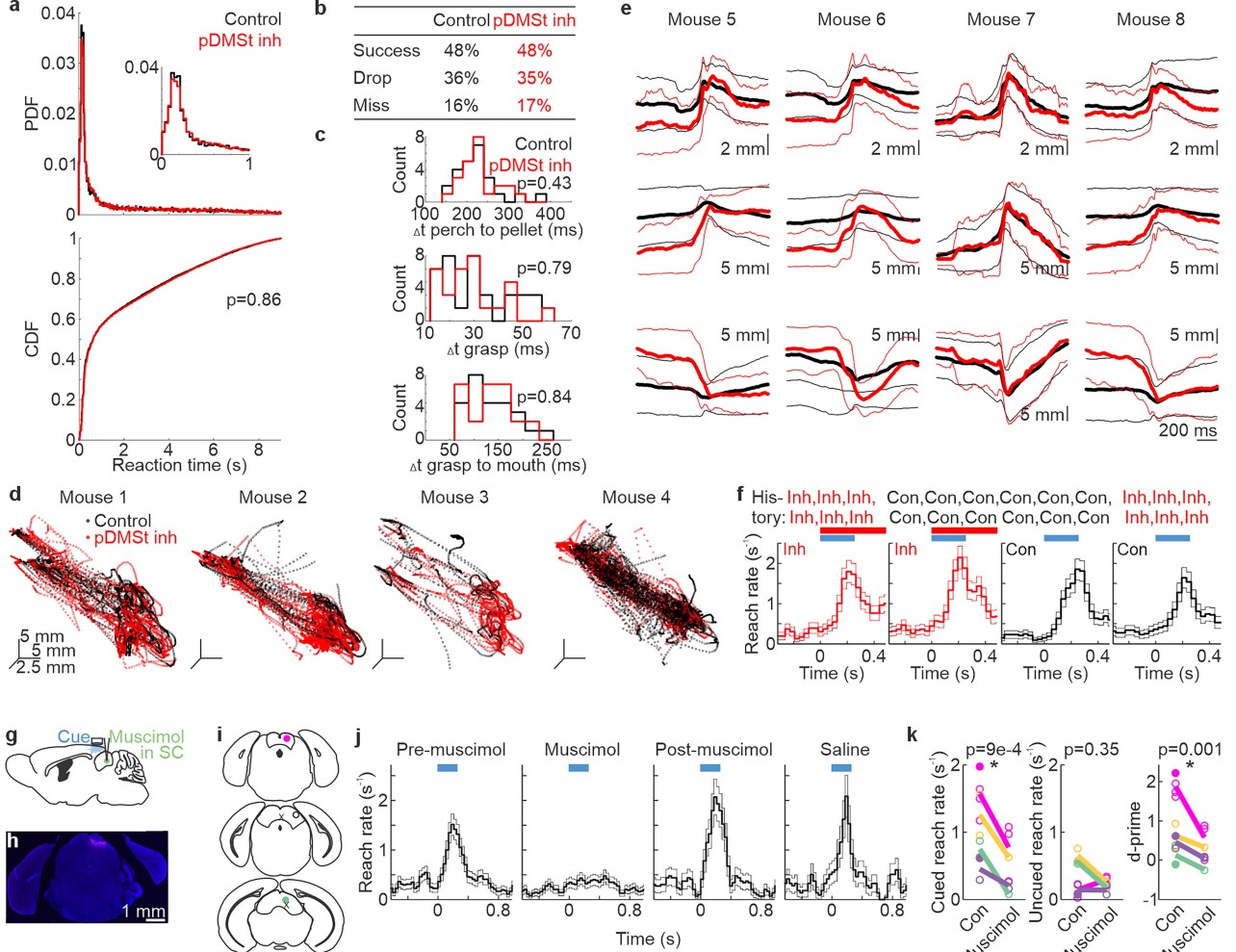

**Extended Data Fig. 6 | pDMSt inhibition does not affect motor kinematics of reach or subsequent recall, but muscimol injections into superior colliculus disrupt recall (i.e., the initiation of cued but not spontaneous reaches after learning) in 4 mice. a-f,** In panels a-f, red represents trials with pDMSt inhibition over the 1-s time window starting 5 ms before the cue, and black represents interleaved control trials. We observed no effects of pDMSt inhibition on motor kinematics of the reach during or after learning; hence, here we present a data set combining days during and after learning. **a,** Reaction time ($t_{arm}$) of first reach after the cue ($t_{cue}$ at t = 0). *top,* Probability density function (PDF) of reaction times across all trials (n = 21858 control trials and 15109 pDMSt inhibition trials from 15 mice). Inset: Close-up from 0 to 1 s. *bottom,* CDF of reaction times. Comparison of black to red p-value is from the Kolmogorov-Smirnov test. **b,** Outcome of first reach after the cue (n = 21858 control trials and 15109 pDMSt inhibition trials from 15 mice). **c,** Histograms showing the durations of different epochs of the reach (n = 24 randomly selected control trials from 10 days from 2 mice, n = 23 randomly selected pDMSt inhibition trials from same 10 days from same 2 mice). *top,* Time from paw resting on the starting perch to the paw touching the pellet. *middle,* Time for paw to close around the pellet. *bottom,* Time to lift the pellet from the pellet presenter disk into the mouth. **d,** 3D trajectories of individual reaches in the 1-s time window immediately after the cue from 4 example sessions from 4 different mice. **e,** Mean and standard deviation of raw reaching trajectories in X, Y and Z dimensions for control trials (black) and during pDMSt inhibition (red). **f,** Frequent pDMSt inhibition after learning does not affect memory recall.

Mean±s.e.m of reach rate across trials, as in Fig. 1, contingent on trial history. List above plot shows pDMSt inhibition or control on previous 6 trials. Black or red color of plot shows pDMSt inhibition (red) or control (black) on current trial. **g-k,** Mice were excluded if muscimol injections caused a complete loss of reaching, as cue-reach associative memory could not be assessed without reaching behavior. **g,** Schematic showing the muscimol injection into the superior colliculus (SC) after mice learned the optogenetic cue. **h,** Example injection site visualized with fluorescent dye (pink) in a post-mortem histological section stained with DAPI (blue). **i,** Schematic of muscimol injection sites in 2 of 4 mice (pink and green). The other 2 mice died before dye injection but had stereotactically targeted injections (see Methods). "O" indicates a recovered injection site from an excluded mouse that did not learn cued reaching; however, spontaneous reaching was unaffected in this mouse. "X" marks injection sites in 2 other excluded mice, which did not recover spontaneous reaching within several hours after muscimol injection (Methods). **j,** Mean±s.e.m. of trials showing cued reaching responses across days: pre-muscimol, muscimol injection day, post-muscimol day, and saline injection day. Data are shown for the 4 mice that recovered spontaneous reaching immediately after muscimol injection. **k,** Summary of cued reach rate (*left*), uncued reach rate (*middle*), and d-prime values (*right*) across 4 mice comparing control days (including saline, shown as filled circles) to muscimol days. Each dot represents one day; different colors and lines represent individual mice. The p-values are from linear mixed effects models (Methods).

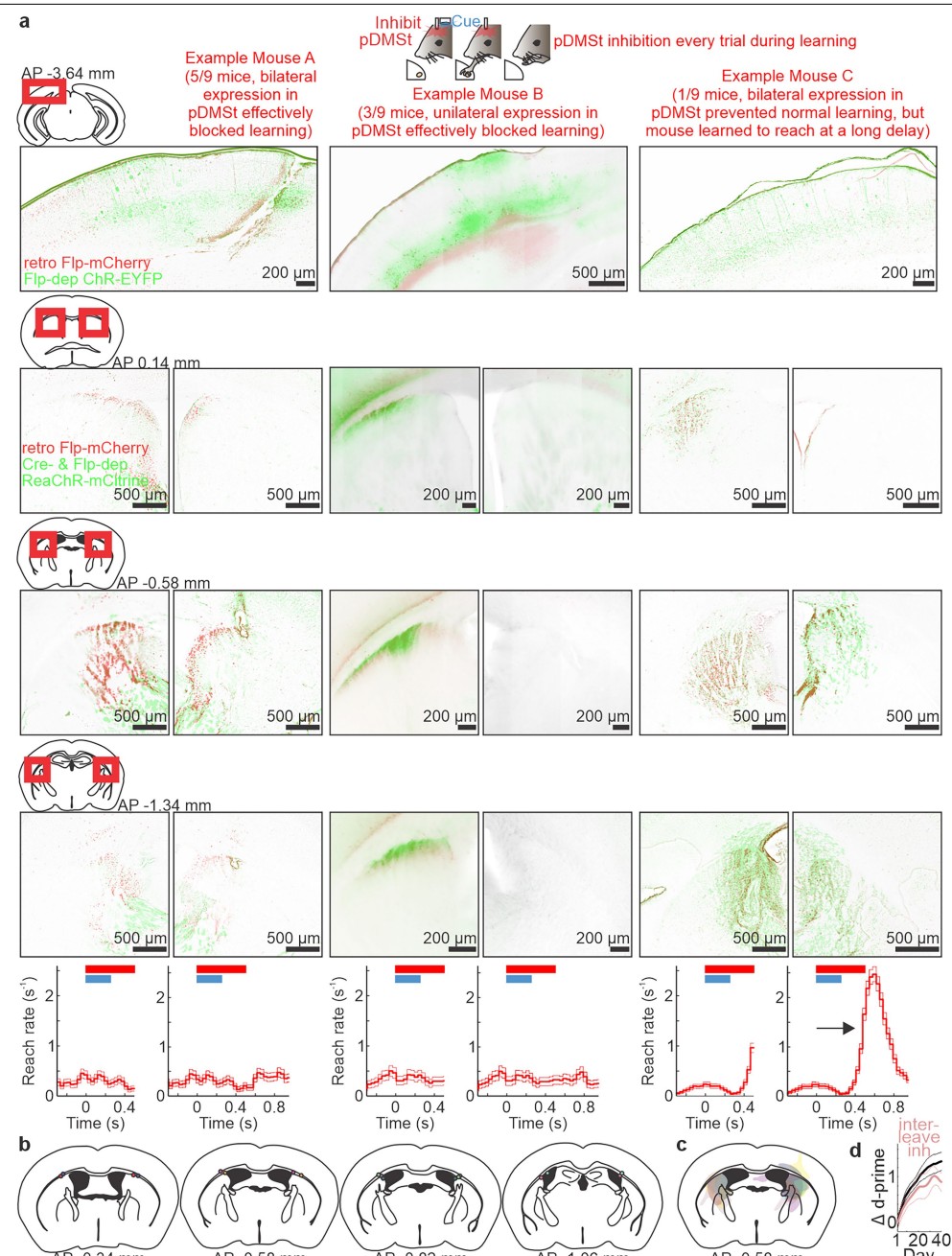

**Extended Data Fig. 7 | Details of the mice experiencing pDMSt inhibition at every cue presentation throughout training. a**, Histology and behavior from mice experiencing pDMSt inhibition over 1-s time window starting 5 ms before cue onset at every presentation of the cue over weeks of training. Schematic sections with red boxes show brain locations of the histology below. **Top row**, Visual cortex histology showing Flp-mCherry (red) and Flp-dependent ChR-EYFP (green). **Rows 2-4**, Anterior to posterior sections of striatum showing Flp-mCherry (red) and ReaChR-mCitrine (green). **Bottom row**, For each of the 3 example mice, two plots, one showing time window matching Fig. 3b, and another showing extended time window continuing after the end of the red laser to inhibit pDMSt. **Left 2 columns labeled Example Mouse A**, 5 of 9 mice had bilateral expression in pDMSt and failed to learn to respond to cue (see reach rate, bottom). Histology is from example mouse in this group. **Middle 2 columns labeled Example Mouse B**, 3 of 9 mice had unilateral expression in

pDMSt. All of these mice failed to learn to respond to cue. **Right 2 columns labeled Example Mouse C**, 1 of 9 mice had bilateral expression in pDMSt and failed to learn to reach within 400 ms time window immediately after the cue but learned to reach at a long time delay. **b**, Schematic of tip placement of bilateral fibers for illumination of pDMSt across these 9 mice. Each color is a mouse. **c**, Schematic of green expression (ReaChR-mCitrine) in striatum across these 9 mice. **d**, Mean±s.e.m. learning curves for control mice (never experienced pDMSt inhibition during learning, black) vs. mice that experienced interleaved pDMSt inhibition during learning (light pink). Only animals that ultimately learned were included, defined as those achieving a d-prime consistently greater than 0.6 within 40 days, as the focus here is on the rate of learning. Learning curves were smoothed using a 15-day uniform bin. Note slower learning when pDMSt inhibition was interleaved.

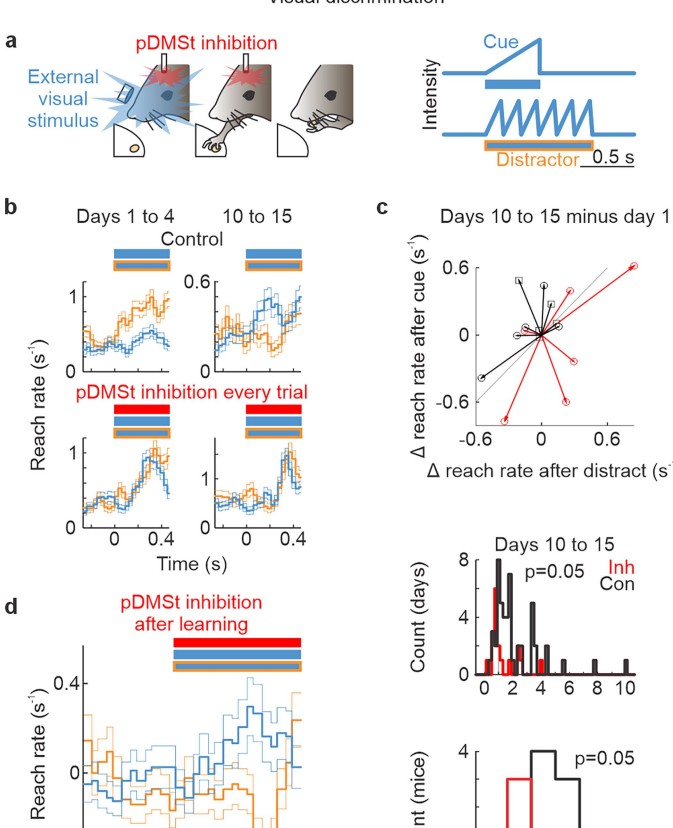

**a**

pDMSt inhibition

External visual stimulus

Cue

Distractor  0.5 s

Intensity

**b**

Days 1 to 4      10 to 15

Control

pDMSt inhibition every trial

Reach rate (s⁻¹)

Time (s)

**c**

Days 10 to 15 minus day 1

Δ reach rate after cue (s⁻¹)

Δ reach rate after distract (s⁻¹)

Days 10 to 15

p=0.05  Inh Con

Count (days)

p=0.05

Count (mice)

Rate ratio

**d**

pDMSt inhibition after learning

Reach rate (s⁻¹)

Time (s)

**Extended Data Fig. 8 | pDMSt inhibition impairs learning but not recall of a natural visual discrimination. a**, Schematic of the external visual stimuli used in the paradigm: a cue paired with a food pellet and a distractor not paired with the food pellet. Both stimuli were emitted from the same LED, with identical spatial structure but distinct temporal profiles. The plot (right) shows the temporal differences: the cue is a slow ramp of light over 0.5 s, while the distractor is a 6 Hz flicker. The maximum LED power was 40 mW, delivered through a 1 mm diameter fiber. Cue or distractor stimuli were presented randomly but with approximately equal probability. Bilateral pDMSt inhibition was applied during or after learning. **b**, Mean±s.e.m. of reach rates across trials following the cue (blue) or distractor (orange) for training days 1 to 4 and 10 to 15. *Top*, control mice (n=from 5 mice, 3298/3136 cue/distractor trials for days 1 to 4, 3140/2361 trials for days 10 to 15). *Bottom*, mice with pDMSt inhibition (1 s, 5 mW) applied during every presentation of the cue or distractor (from 6 mice, n = 3092/2817 cue/distractor trials for days 1 to 4, 1939/1429 trials for days 10 to 15). **c**, *top*, Change in reach rate in a 400 ms window after the distractor (X axis) versus the cue (Y axis). Most control mice (black circles, n = 5 mice) learned to increase reaching after the cue compared to the distractor, while mice with pDMSt inhibition (red squares, n = 6 mice) did not. Black squares show data from pDMSt-inhibited mice after inhibition was removed, allowing recovery of natural learning (n = 5 mice, recovery data not collected from 1 mouse). *Middle*, Histograms of the ratio of reach rates (cue to distractor) across training days 10 to 15. *Bottom*, Histogram of the same ratio, averaged across days 10 to 15, for each mouse. P-values from a linear mixed effects model (top) and Wilcoxon rank-sum test (bottom). **d**, Baseline-subtracted reach rates (0.5 s baseline) following the cue and distractor during trials with pDMSt inhibition applied after, not during, learning (from 5 mice, n = 1969/1418 cue/distractor trials).

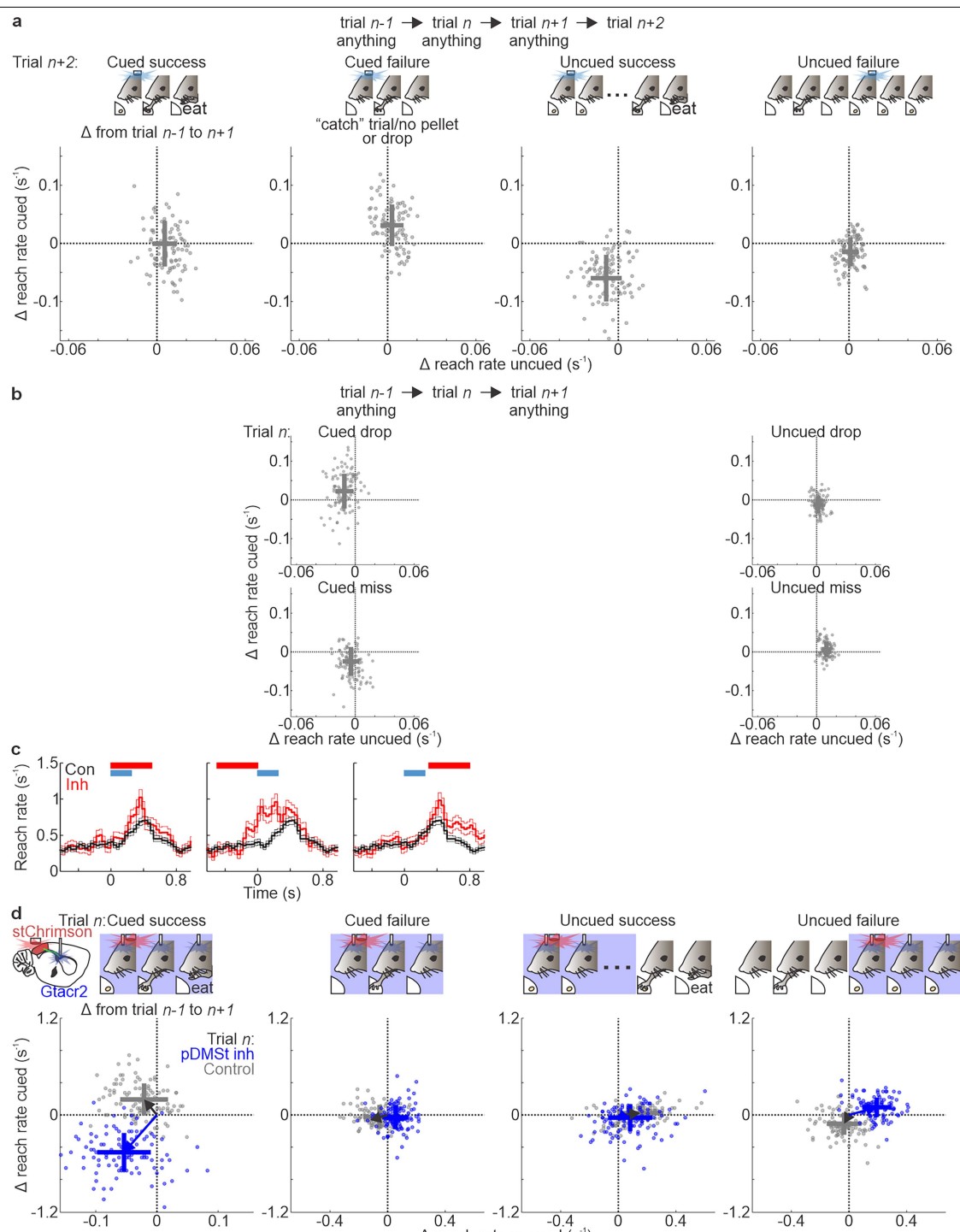

**Extended Data Fig. 9 | Controls for trial-to-trial reinforcement based on the outcome, including backwards time control and alternative pDMSt silencing approach. a**, To test whether the trial-to-trial update observed in Fig. 4 is manifest forward but not backward in time, we measured the effect on trial $n+1$ of the trial outcome on trial $n+2$. We compared trial $n+1$ to trial $n-1$, as in Fig. 4, but here we considered trial sequences conditioned on the outcome of trial $n+2$. The behavioral experience on trial $n+2$ was: *Left*, A cued success (n = 2587 trials from 37 mice). *Left-middle*, A cued failure (n = 3198 trials from 37 mice). *Right-middle*, An uncued success (n = 1660 trials from 37 mice). *Right*, An uncued failure (n = 6110 trials from 37 mice). **b**, Effect of trial $n$ outcome on the next trial, comparing trial $n+1$ to trial $n-1$, as in Fig. 4. Here we divide failures into two different types: the mouse grabbed then dropped the pellet, or the mouse reached but failed to touch the pellet, called a miss. **c**, Effect of varying the timing of pDMSt inhibition on trial $n$. Reach rate plotted as in Figs. 1–3. Note that pDMSt inhibition can sometimes evoke longer-latency

reaches (>250 ms reaction time) when the inhibition does not begin simultaneously with the cue. This may occur because the mice learn to respond to the rebound from pDMSt silencing. In Fig. 4e,f, we compare the change in behavior from trial $n+1$ to trial $n-1$, contrasting control and pDMSt inhibition conditions on trial $n$. To ensure consistent reach timing, we only include trials where the reach occurs in the same time window on trial $n$ for both control and pDMSt inhibition conditions. **d**, Layout as in Fig. 4. Here the optogenetic inhibition of pDMSt was by GtACR2 inhibition (Methods, 'Optogenetically inhibiting the pDMSt using GtACR2'). We used ChrimsonR to activate the cue neurons in visual cortex in these mice. Blue dots are from the trials with GtACR2 inhibition. Gray dots are from the control trials. n = 104 cued success control trials, 88 cued success GtACR2 inhibition trials, 192 cued failure control trials, 279 cued failure GtACR2 inhibition trials, 91 uncued success control trials, 113 uncued success GtACR2 inhibition trials, 228 uncued failure control trials, 248 uncued failure GtACR2 trials from 4 mice. Qualitatively similar results to Fig. 4.

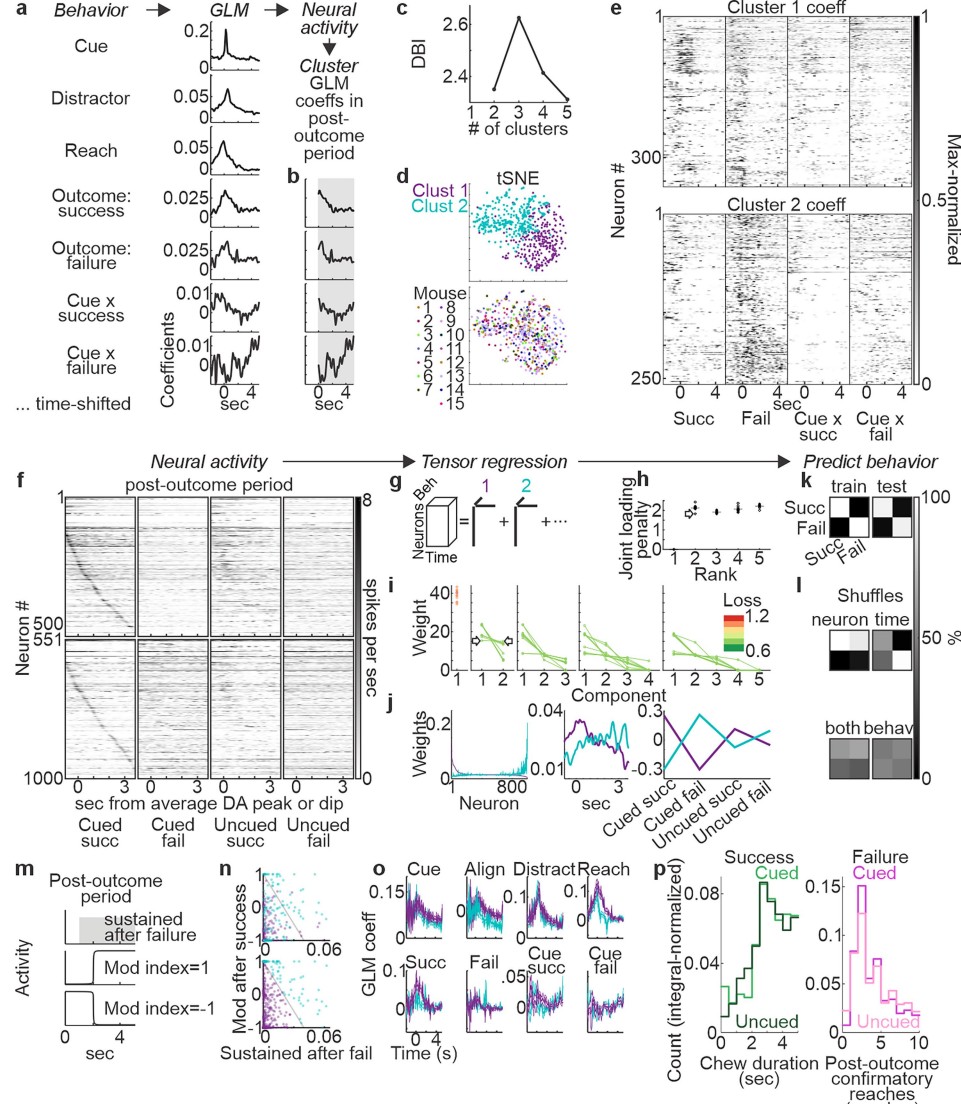

**Extended Data Fig. 10** | See next page for caption.

**Extended Data Fig. 10 | Two approaches to cluster the pDMSt neuronal responses in the post-outcome period, and no significant behavioral difference after cued versus uncued success in the post-outcome period.** Panels **a-o** show neural activity. Panel **p** shows behavior. Panels **a-e** show the first approach, a generalized linear model (GLM). Panels **f-l** show the second approach, tensor regression. This figure analyzes only putative striatal projection neurons (SPNs) (Methods, 'Identifying putative SPNs'). This figure uses only the training set (half of the data set) to cluster the pDMSt neuronal responses. Fig. 5 uses the other half of the data set (the test set) to decode behavior from neural activity. **a**, We built a GLM to describe how each neuron's activity relates to behavior events (Methods, 'Generalized linear model'). A GLM attempts to use behavior events to predict neural activity. The result is a set of coefficients, or weights, assigned to each neuron for each behavior event. These weights capture the pattern of that neuron's response to the behavior event. Below "Behavior", we list the behavior events. Below "GLM" and to the right of each behavior event, we show the resulting GLM coefficients. These are the coefficients averaged across all neurons. 0 s is the time of the behavior event. For "outcome: success", "outcome: failure", "cue x success" and "cue x failure", 0 s is $t_{arm}$, the moment that the arm is outstretched during the reach. **b**, Note that the first three GLM coefficients ("cue", "distractor", "reach") are not aligned to the outcome, so we ignored them for subsequent analysis. We took the GLM coefficients *after* an outcome ("outcome: success", "outcome: failure", "cue x success" and "cue x failure") in the post-outcome period (>0 s, gray shaded area). For each neuron, we made a vector that puts together these 4 sets of coefficients. We call this vector the "outcome profile" of the neuron. Neurons lacking any GLM coefficients greater than zero in the post-outcome period do not have an outcome profile and were excluded. We clustered the outcome profiles of all remaining neurons using k-means clustering. **c**, The Davies-Bouldin Index (DBI) for different numbers of k-means clusters. Lower values are better. **d**, The result of k-means clustering for 2 clusters. Each dot is one neuron. *top*, tSNE of the outcome profiles. *bottom*, Same tSNE, but here neurons are colored according to which mouse brain contained that neuron. **e**, GLM coefficients after an outcome. Neurons missing if they did not have any GLM coefficients greater than zero in the post-outcome period (Methods, 'Clustering the GLM coefficients in the POP'). **f**, Tensor regression attempts to predict the behavior trial type (cued success, cued failure, uncued success or uncued failure) from the neural activity of all neurons together. Like principal components analysis (PCA), the tensor regression produces multiple components. (See Methods, 'Setting up the tensor regression' for more details.) Here we show the trial-averaged activity of all of the neurons sorted by component 1 > component 2 (*top row*) or component 2 > component 1 (*bottom row*). Within each row, we further sorted the neurons according to the time delay of the peak response near a cued success. **g**, Schematic describing tensor regression, i.e., regress behavior trial type against neural activity, then represent the result as a sum of components. Each component is the outer product of 3 rank-1 tensors (more details in

Methods, 'Setting up the tensor regression'). We ran an optimization to find the tensor regression solution (Methods, 'Tensor regression optimization'). This solution is not unique, so different initial conditions produce different results. **h** and **i** summarize the results over multiple optimization runs. **h**, The joint loading penalty penalizes solutions in which one neuron relies too heavily on more than one component. We chose a solution with a low joint loading penalty, which is a parsimonious solution that loads different components onto different sets of neurons. (See Methods, 'Choosing a specific tensor regression solution') **i**, We tried different numbers of components (Methods, 'Selecting the rank of the tensor regression'). The 2-component solutions had a loss similar to the more complicated 5-component solutions. Therefore, for simplicity, we selected a 2-component solution. **j**, Result of the tensor regression. *left*, Loadings onto neurons for component 1 (purple) versus component 2 (cyan). Note that the two components target largely non-overlapping groups of neurons. *middle*, Loadings onto timepoints for component 1 (purple) versus component 2 (cyan). *right*, Loadings onto behavior trial types for component 1 (purple) versus component 2 (cyan). **k**, To determine whether the tensor regression simply clusters noise, we asked the tensor regression to predict the behavior trial type from the neural activity in the test set (see Methods, 'Training and test sets'). Results are shown as a confusion matrix for the training (*left*) and test (*right*) sets. **l**, Shuffles accompanying **k**. *top left*, Neuron ID shuffle. *top right*, Timepoints shuffle. *bottom left*, Shuffle both neuron IDs and timepoints. *bottom right*, Shuffle behavior trial type. **m**, Metrics to summarize the post-outcome period GLM coefficients (also see Methods, 'The simpler approach to the neuron groups 1 and 2' used in Fig. 5). Sustained after failure is the absolute value of the average coefficient in the time window 1 to 5 s after the time of the arm outstretched, $t_{arm}$. Modulation index (mod index) is the GLM coefficient average from 2 to 5 s minus the GLM coefficient average from 0 to 2 s after $t_{arm}$, divided by the sum of these two quantities. **n**, Response of each neuron summarized by the metrics explained in panel **m**. Each dot is a neuron. *top*, Colors are from Cluster 1 (purple) and Cluster 2 (cyan) in panel **d**. *bottom*, For simplicity, we drew a line to roughly separate the purple and blue neurons of Clusters 1 and 2. We used this line to divide the neurons into two groups, called Consensus Group 1 and Consensus Group 2. These Consensus Groups were used to make Fig. 5 (see more explanation in Methods, 'The simpler approach to the neuron groups 1 and 2' used in Fig. 5). **o**, Average±s.e.m. of GLM coefficients across neurons. Neurons grouped into Consensus Group 1 (purple) and Consensus Group 2 (cyan). "Align" shows cue coefficients after subtracting pre-cue baseline (i.e., t < 0 s). **p**, Integral-normalized histograms of behavior metrics from the post-outcome period. *left*, Chewing duration after a successful reach, comparing cued to uncued successes. P-value from Wilcoxon rank sum test is 0.6. *right*, Number of additional, confirmatory reaches after a failed reach, comparing cued to uncued failures. P-value from Wilcoxon rank sum test is 0.04. n = 3685 cued successes, 916 uncued successes, 4724 cued failures, 2414 uncued failures from 17 mice.

# Reporting Summary

## Statistics

For all statistical analyses, confirm that the following items are present in the figure legend, table legend, main text, or Methods section.

| n/a | Confirmed | |
|---|---|---|
| ☐ | ☒ | The exact sample size (*n*) for each experimental group/condition, given as a discrete number and unit of measurement |
| ☐ | ☒ | A statement on whether measurements were taken from distinct samples or whether the same sample was measured repeatedly |
| ☐ | ☒ | The statistical test(s) used AND whether they are one- or two-sided *Only common tests should be described solely by name; describe more complex techniques in the Methods section.* |
| ☐ | ☒ | A description of all covariates tested |
| ☐ | ☒ | A description of any assumptions or corrections, such as tests of normality and adjustment for multiple comparisons |
| ☐ | ☒ | A full description of the statistical parameters including central tendency (e.g. means) or other basic estimates (e.g. regression coefficient) AND variation (e.g. standard deviation) or associated estimates of uncertainty (e.g. confidence intervals) |
| ☐ | ☒ | For null hypothesis testing, the test statistic (e.g. *F*, *t*, *r*) with confidence intervals, effect sizes, degrees of freedom and *P* value noted *Give P values as exact values whenever suitable.* |
| ☐ | ☒ | For Bayesian analysis, information on the choice of priors and Markov chain Monte Carlo settings |
| ☐ | ☒ | For hierarchical and complex designs, identification of the appropriate level for tests and full reporting of outcomes |
| ☐ | ☒ | Estimates of effect sizes (e.g. Cohen's *d*, Pearson's *r*), indicating how they were calculated |

*Our web collection on statistics for biologists contains articles on many of the points above.*

## Software and code

Policy information about availability of computer code

Data collection | All custom code available at https://github.com/kimerein. Arduino code for behavior rig at https://github.com/kimerein/behaviorRig. SpikeGLX (v3) and PlexControl (v1) for in vivo physiology. Igor (v6.02) for slice physiology. FlyCapture (v2) for high-speed video collection. OlyVIA (v4.1) and VS120 or VS200 slide scanner for histology imaging.

Data analysis | Chronux (v3), DeepLabCut (v2.2), UltraMegaSort (v2000). All custom codes freely available online at https://github.com/kimerein. Tensor regression package at kimerein's Github and Richard Hakim's Github. Links here:
MATLAB analysis code: https://github.com/kimerein/integrate-phys-and-beh
https://github.com/kimerein/KR_Analysis_Toolbox
Python GLM code: https://github.com/kimerein/k-glm
Automated analysis of reaching in low-speed video:
https://github.com/kimerein/reach-behavior-analysis
https://github.com/kimerein/reachBehavior
Multi-unit processing of data from Plexon and WHISPER systems:
https://github.com/kimerein/MU-analysis
Python tensor regression: https://github.com/kimerein/tensor_regression
Photometry acquisition: https://github.com/kimerein/photometry
Arduino code: https://github.com/kimerein/behaviorRig
Python align high-speed video to events in behavior:
https://github.com/kimerein/integrate-phys-and-beh
UltraMegaSort spike sorting:
https://github.com/kimerein/Mat_Code/tree/master/UltraMegaSort

For manuscripts utilizing custom algorithms or software that are central to the research but not yet described in published literature, software must be made available to editors and reviewers. We strongly encourage code deposition in a community repository (e.g. GitHub). See the Nature Portfolio guidelines for submitting code & software for further information.

## Data

Policy information about [availability of data](availability of data)

All manuscripts must include a [data availability statement](data availability statement). This statement should provide the following information, where applicable:

- Accession codes, unique identifiers, or web links for publicly available datasets
- A description of any restrictions on data availability
- For clinical datasets or third party data, please ensure that the statement adheres to our [policy](policy)

> We provide summary data sets at https://dataverse.harvard.edu/dataset.xhtml?persistentId=doi:10.7910/DVN/QPQEC9. We also provide example data sets for running the code at this same location. Because the total amount of raw data is well over 10 TB, and this volume is not well supported by the Harvard Dataverse, we have not uploaded all raw data to the Harvard Dataverse, but any raw data will be made freely available upon request.

# Field-specific reporting

Please select the one below that is the best fit for your research. If you are not sure, read the appropriate sections before making your selection.

☒ Life sciences ☐ Behavioural & social sciences ☐ Ecological, evolutionary & environmental sciences

For a reference copy of the document with all sections, see [nature.com/documents/nr-reporting-summary-flat.pdf](nature.com/documents/nr-reporting-summary-flat.pdf)

# Life sciences study design

All studies must disclose on these points even when the disclosure is negative.

| | |
|---|---|
| Sample size | Fig. 1: no pre-determination of sample size; choice of sample size: initially, we trained 6 mice in the behavior to ensure that animals could learn, but in the figure, we include the relevant mice collected after training mice for 4 years; sample size is sufficient, because clear differences in reaching across stages of learning. Fig. 2: no pre-determination of sample size; choice of sample size: we included all relevant mice after collecting data for 4 years; sample size is sufficient to see changes from first to second half of session, yet no observed change between control and interleaved pDMSt inhibition trials. Fig. 3: choice of sample size: we predetermined the sample size of mice in each cohort based on the max behavioral throughput given available experimenters; control and pDMSt silencing mice were run concurrently in batches of control plus pDMSt silencing mice; sample size is sufficient to see a learning deficit in pDMSt silencing mice (see Fig. 3 for statistics). Fig. 4: no pre-determination of sample size; choice of sample size: we included all relevant mice after collecting data for 4 years; sample size is sufficient, because bootstrap analysis indicates similar results and variability when we analyze the group of 37 mice or analyze the group of 11 mice, and hence adding more mice does not change the result. Fig. 5: no pre-determination of sample size; choice of sample size: after building a physiology rig, we recorded from all mice passing through the behavior pipeline over more than a year, before we switched to analysis of this data set without knowing the results; sample size is sufficient, because successful decoding of behavior condition indicates that sufficient data was acquired to decode held-out test data. |
| Data exclusions | Fig. 1: includes only mice that learned the task, excluding mice that failed to learn (but all mice are shown in final panel of Fig. 3), as planned. Fig. 2: all relevant mice and sessions included, as described in figure legend. Fig. 3: one mouse missing from recovery, as this mouse died. Fig. 4: all relevant mice included, except 5 mice were excluded from all analyses throughout the paper, because the behavior rig was not set up properly (as explained in Methods). Fig. 5: all relevant mice included. |
| Replication | Fig. 1: most mice learned the task, but others did not (Fig 3 final panel shows all of the control mice that we attempted to train, including mice that failed to learn). Fig. 2: for in vivo physiology, we measured pDMSt silencing in two separate cohorts of mice recorded on different physiology rigs separated in time by more than one year -- hence, the pDMSt silencing method replicates; Fig. 2 suggests that the lack of effect of pDMSt silencing at the end of the session replicates across mice. Fig. 3: experimenters were blinded to genotype, and figure made with all relevant mice, but no specific replication of this result with another group of mice. Fig. 4: included all mice (n=37 mice), but no specific replication using another group of mice. Fig. 5: included all mice, and decoder trained on cross-validated data and tested on held-out data set. |
| Randomization | Genotype of the animal determined its belonging to the control or experimental (pDMSt silencing) group. We used mice as available including the double transgenic pDMSt silencing mice (Nkx2.1-Cre X ReaChR), preferring to use transgene-negative cage mates as the paired controls. |
| Blinding | Experimenters were blinded to animal genotype in Fig. 3. There was no blinding to genotype in the other figures, but note that training with the cue is automated and open-loop with little opportunity for experimenter influence. |

# Reporting for specific materials, systems and methods

We require information from authors about some types of materials, experimental systems and methods used in many studies. Here, indicate whether each material, system or method listed is relevant to your study. If you are not sure if a list item applies to your research, read the appropriate section before selecting a response.

## Materials & experimental systems

| n/a | Involved in the study |
|-----|----------------------|
| ☐ | ☒ Antibodies |
| ☒ | ☐ Eukaryotic cell lines |
| ☒ | ☐ Palaeontology and archaeology |
| ☐ | ☒ Animals and other organisms |
| ☒ | ☐ Human research participants |
| ☒ | ☐ Clinical data |
| ☒ | ☐ Dual use research of concern |

## Methods

| n/a | Involved in the study |
|-----|----------------------|
| ☒ | ☐ ChIP-seq |
| ☒ | ☐ Flow cytometry |
| ☒ | ☐ MRI-based neuroimaging |

## Antibodies

| | |
|---|---|
| Antibodies used | DARPP-32: Novus Biologicals primary antibody (Product # NB110-56929), GFP: anti-GFP from abcam (Product # ab13970), Anti-chicken: secondary antibody conjugated to Alexa488 from ThermoFisher (Product # A-11039), Anti-rabbit: secondary antibody conjugated to Alexa594 from ThermoFisher (Product # A-11012) |
| Validation | DARPP-32: There are no validation statements on the website, and we did not specifically validate the antibody. GFP: The abcam website states that the product was tested and does not cross-react with other proteins that differ from GFP by just a few point mutations (e.g., YFP). Anti-chicken: ThermoFisher website cites 2722 references. Anti-rabbit: ThermoFisher website cites 3293 references. |

## Animals and other organisms

Policy information about studies involving animals; ARRIVE guidelines recommended for reporting animal research

| | |
|---|---|
| Laboratory animals | 65 males and 62 females, including WT, Nkx2.1-Cre transgenic mouse line (Jackson Labs Stock #008661), Cre-On and Flp-On ReaChR transgenic mouse line (R26 LSL FSF ReaChR-mCitrine, Jackson Labs Stock #024846), Adora2a-Cre (GENSAT B6.FVB(Cg)-Tg(Adora2a-cre)KG139Gsat/Mmucd), D1-Cre (GENSAT B6.FVB(Cg)-Tg(Drd1a-cre)EY262Gsat/Mmcd). All mice (all strains) were initially injected between 2 and 6 months of age and trained 3 weeks after virus injection. Some animals continued to participate in the behavior up to 1.5 years of age. |
| Wild animals | no wild animals |
| Field-collected samples | no field-collected samples |
| Ethics oversight | President and Fellows of Harvard College IACUC |

Note that full information on the approval of the study protocol must also be provided in the manuscript.

