## [Peer Review File · Nature]

Striatum supports fast learning but not memory recall

Corresponding Author: Professor Bernardo Sabatini

Version 0:

Reviewer comments:

Referee #1

(Remarks to the Author)

In this manuscript, Reinhold et al use an optogenetically-cued head-fixed reaching task to study the role of corticostriatal inputs and striatal neurons (in the posterior dorsomedial striatum tail, pDMSt) in learning and “recall”. They find that while pDMSt neurons have specific patterns of activation during task trials according to the action/outcome, optical manipulations suggest the pDMSt is important for the learning but not recall of the task. The most potentially provocative conclusion is that the striatum is no longer required once the task has been learned.

Some of the questions addressed here have been tackled in other work, albeit with different methodology: (1) another publication lesioned the pDMSt projection neurons (Ruediger and Scanziani eLife 2020) and found impairment in learning of a visual discrimination task. (2) multiple papers from the Hikosaka group have shown value coding of visual stimuli in the analogous region of the caudate tail in non-human primate, using electrophysiology (eg Kunimatsu et al, PNAS 2021), and also shown the role of projections to the substantia nigra and superior colliculus in the associative value coding of visual stimuli (eg Amita et al, 2020). The same group has used muscimol to demonstrate a role of the caudate tail in performance of the visual/value task. Finally, several other labs have examined the role of corticostriatal inputs and SPNs (not in the tail of the striatum, however) in the learning of associative tasks, or in the performance of such tasks, once learned.

This study has a somewhat unusual design, which may make some skeptical of the interpretability of the findings. However, the authors have included many experimental and analytical controls in the Supplemental figures which strengthen the manuscript and are important in understanding their findings. Overall, the questions and experiments outlined in the paper are interesting and relevant to those interested in reinforcement learning and the role of the tail of the striatum. However, as currently presented the manuscript is hard to parse.

Major Comments:

1. Though the data collected is impressive, I found the presentation to be challenging to read and process. I had to put in a lot of extra time to find the experiments/analyses I suspected the authors performed, but were not mentioned in the main text. Examples include the GtACR experiments, the no-virus controls, and the validation that LED light (outside the head) produced similar responses to the activation of visual cortex to pDMSt neurons, but there are others. The main figures are highly compressed and there are numerous supplements, and quite a bit of critical information is in the Figure legends (including the Supplementary Figure legends) or even in the Methods (not mentioned at all in the main text), making the Results section hard to read by itself. I am not sure if this amount of information can be conveyed in this format with another approach, or just doesn't lend itself to this format.
2. The authors use a unique “cue” in their cued reaching task – optogenetic stimulation of visual cortex neurons that project to pDMSt, and go on to use this cued task to demonstrate the role of the pDMSt. However, would transient inhibition of pDMSt neurons prevent traditional visually cued learning?
3. The optogenetically cued task is quite complicated, and much of the important information for understanding the task and training parameters is in the Supplement or Methods sections. As a reader I would have benefited from having the setup schematic (found in the Supplement) in the main figure.
4. Many important controls are mentioned after the main behavioral findings are presented (eg that reaches increase at the optogenetic “cue”). I would have found it helpful to mention the control findings in parallel with the main findings, as I became impatient while reading, wondering if the authors had done appropriate controls (which they had), and how these compared to the primary analyses. Also, in line 142, it says sessions were excluded if they “failed these controls” – what does that mean and how often did it happen?
5. The authors test the idea that pDMSt projection neurons are important for learning of the cued reaching task not by direct

optogenetic inhibition of SPNs, but by activation of GABAergic interneurons (Nkx2.1 + ReaChR). Is it possible that in addition to inhibiting SPNs, the released GABA inhibits local dopamine terminals and might impair learning by reducing the dopamine release that occurs with cued/correct trials? By careful reading of the Supplements and Methods I discovered that the authors also performed experiments with GtACR in SPNs (this is not mentioned at all in the main text), which would tend to address this question. However, as this experiment is not explained in the main text it's very labor-intensive as a reader to determine if it was done the same way or was up to the same standards as the experiments in the main text.

6. Did the pDMSt inhibition experiments produce any other behavioral effects (eg overall motor activity/freezing), outside of cued reaching?

7. In line 233, the authors introduce the section with dLight photometry and extracellular electrophysiology in the pDMSt by saying that they made recordings in "beginner" and "expert" mice. Recording in both of these phases seems important given the findings with optogenetic manipulations in Figure 3. However, I could not figure out what recordings or comparisons were made in "beginner" or "expert" mice, and if there were any differences in neural activity patterns in these two phases. I think this type of comparison would be very important in light of their negative results for optical manipulations during "recall" or performance of the task, once learned.

8. The authors find that once the task is learned, inhibiting the pDMSt doesn't impair performance, and conclude that axon collaterals of visual cortex neurons that go to locations other than pDMSt may be important at this stage. However, they don't provide any direct/positive evidence for this speculation. In the discussion they suggest it might be via pathways to other visual cortical neurons and/or the superior colliculus, but don't provide direct evidence for this idea.

Minor comments:

1. The authors describe the optogenetic cued trials, stating on line 117 that "the pellet was unavailable before cue onset" and then that the pellet "became available shortly before cue onset" (0.22 seconds). These statements seem to be contradictory. I suspect the latter statement is more accurate and this sentence should just state that the pellet became available 0.22 seconds before each cue onset.

2. Is it meaningful that the performance of controls in the ReaChR experiments (Figure 3), in terms of reaches/sec after the cue, or d prime, was lower than in animals shown in Figure 1?

Referee #2

(Remarks to the Author)

This interesting report demonstrates the selective necessity of neurons in the tail of the posterior dorsomedial striatum for learning that supports the ability to use optogenetic activation of the visual input to this structure as a discriminative cue signaling reinforcement of a reaching action. Mice can learn to perform a reaching action in response to optogenetic activation of visual cortical input to the pDMSt. Inhibition of pDMSt projection neurons (via activation of GABAergic interneurons) during the reach behavior disrupts learning of this task, but, interestingly, does not affect performance after learning. Thus, pDMSt neurons are needed to learn this discriminative cue task, but may not be necessary for recalling or using such discriminative memories to guide behavior. High density neuronal recordings provide information on the endogenous activity of these neurons, showing they encode information needed for successful reinforcement of the discriminative cue-action task. Although there has been a long history of assessment of striatal involvement in instrumental learning and memory, this study adds information on this particular subnucleus of the striatum and its potential selective involvement in learning but not memory recall. The use of an optogenetic cue potentially isolates the pathway through which visual cue information can get to the pDMSt to support visual discriminative instrumental learning. The combination of optogenetic methods is innovative. The addition of high density recordings adds an important layer of converging evidence for the learning findings and provides useful context for how pDMSt SPNs might contribute. There are many strengths to the experimental design, including controls to validate the mice were using the optogenetic stimulation as a cue and that the optogenetic inhibition effects are not secondary to effects on kinematics. Validation of the optogenetic manipulations is a strength. The studies are very nicely done and rigorous. The paper is well written. The results are novel and potentially important, but limited in 3 ways. First, it is clear that the pDMSt SPNs mediate learning in this task, but whether they do not mediate memory retrieval is not fully demonstrated. Second, it is not clear if these findings apply selectively to optogenetic cue discriminative learning, or whether this region is also important for natural discriminative stimulus instrumental learning. Third, there is no information on the potentially unique or complementary contributions of the functionally and anatomically distinct subtypes of striatal projection neurons. Additionally, more information is needed to understand what region of the striatum is targeted and distinguish the findings here from what is already known about the posterior DMS and tail of the striatum. I expand on these below.

1. Selective function in learning and not memory recall. There are several potential alternative explanations for the lack of an effect of pDMSt on performance during the memory-recall tasks. One simple one is differences in the amount of inhibition. Inhibition was given on all trials for learning experiments v. a random subset of trials for performance/memory recall experiments. This difference could explain why inhibition of pDMSt SPNs was effective when applied during learning, but not memory recall. Is the putative selective effect on learning due to more inhibition or the lack of effect on recall due to an insufficient amount of inhibition? A stronger and/or longer inhibition during this test phase (even with something simple like muscimol) would provide converging and stronger evidence for the null effect.

It is well known that instrumental behaviors of the sort studied here can be controlled by multiple memory systems that rely on distinct brain circuits, centered on distinct subregions of the striatum. Action-outcome (in this case cue-action-outcome) requiring the pDMS or S-R habits requiring the DLS being a classic example of this. Another is model-free v. model-based reinforcement learning. The nature of the memory/memories used to guide behavior was not evaluated here. It is presumed to be a sensory-motor association, but it is not clear that this rather than an outcome-based memory is controlling the behavior. Most likely this behavior could be control by both of these memory systems. It would be very interesting to probe

this more deeply. Understanding the type of learning supported in this region and how this region supports learning would be impactful. But more importantly, the possibility that this behavior could be controlled by multiple memory systems is critical for interpreting the null effects of inhibition during the performance phase. If the behavior can be controlled by 2 (or more) different forms of memory, then it could well be the case that pDMSt SPNs do mediate the recall of one type of memory (maybe the action-outcome association), but upon inhibition of these neurons perform shifts to be controlled by another (perhaps stimulus-response, model-free, or sensorimotor) memory. Test of outcome sensitivity (e.g., with devaluation), as have been used classically and more recently to delineate striatal function in instrumental learning and memory may be useful in this regard. Such tests have previously shown the pDMS to be critical for both the learning and performance of action-outcome memories. I am tending to assume that the learned association here is instrumental, given the contingent setup. However, it is also plausible that this task can be performed in a Pavlovian manner, further leading to questions about what memory/memories are learned here and what aspects of these memories are pDMSt neurons involved in learning and/or using.

An explanation for the null effect of pDMSt inhibition on memory recall/performance tests is that it due to axon collaterals of cortico-striatal neurons triggering the cued action via another brain pathway, but this is not substantiated.

2. Do the findings only apply to this form learning? It seems important to know whether the proposed selective function of the pDMSt in learning, not performance of discriminative cue instrumental learning is specific to learning induced by optogenetic activation of visual input to the pDMSt or would apply to other forms of visual cue discriminative instrumental learning or learning of discriminative cues of other modalities. The latter may be likely given prior evidence that mice can learn to respond to optogenetic stimulation of auditory inputs to the tail of the striatum. Prior work has shown that mice can perceive optogenetic manipulations to the brain and use them to guide actions (<https://www.eneuro.org/content/9/3/ENEURO.0216-22.2022>). This seems highly relevant and should be cited here. The results of this prior work bring up questions here about whether activation of the visual cortex projection to the pDMSt is critical or whether similar results would be obtained with optogenetic stimulation of other parts of the brain. It is also important to understand the role of this region in natural cue discriminative instrumental learning.

3. SPN subtypes. The striatum is well known to have anatomically distinct direct and indirect projection pathways to basal ganglia output structures. Neither the recording nor manipulation components of this study distinguished between cell types, which is important given functional differences in these output pathways. Some understanding of the potentially unique, similar, or complementary functions of the direct v. indirect pathway SPNs would enhance the potential impact of this report.

4. Histological validation of viral expression and fiber placement should be in the main text figures so readers don't have to go hunting for this important information. In addition to representative examples, it would also greatly help to have schematic maps of viral expression spread and fiber placement showing the placements for all subjects. In some cases, it appears as if there might be some spread to the GPe.

5. Relatedly, how distinct is this subregion of the striatum from the previously studied posterior DMS and tail of the striatum? The pDMS has been considerably implicated in instrumental learning similar to that evaluated here (e.g., work from Corbit, Janak, Balleine, Yin etc.), including the performance of instrumental behavior. The tail of the striatum has also been implicated in instrumental behavior- including the performance of instrumental discriminations similar to those studied here (e.g., Znameskiy & Zador 2013). Some discussion of these findings in light of this prior work is needed to better contextualize the findings and understand what is novel here.

6. P values are provided in the figures, but full statistical reporting needed to evaluate the support for the claims appears missing from the main manuscript. It is not always clear if trials or subject is the N unit.

7. Numbers of subjects that are male v. female is needed in the figure legends.

Referee #3

(Remarks to the Author)

Reinhold et al. developed a unique paradigm to study how visual cortex projections to the posterior dorsomedial striatum tail (pDMSt) contribute to learning an association between a cue and a presentation of a food pellet a mouse can reach and retrieve.

The authors provide a solid demonstration that mice can learn to use an optogenetic cue to initiate a reaching behavior to retrieve a food pellet. This cue consists of optogenetic activation of pDMSt projecting V1 neurons. The authors confirm that the animals are using this cue to learn that a pellet has been presented through a comprehensive set of behavioral controls. The authors add a second optogenetic manipulation in the pDMSt to determine whether V1 to pDMSt projections are involved in learning the cue-behavior association.

The physiological relevance of the experiments is uncertain and there are several major concerns about the logic of the conclusions. Given this, without further work (for example, comparing these results to inhibition of superior colliculus as mentioned in the discussion) the current conclusions seem incremental. These comments are outlined below:

(1)

The authors are careful to point out that optogenetic activation of pDMSt-projecting V1 neurons also activates other regions

due to axon collaterals of these neurons. However, this “off target” activation presents an important confound in their main conclusions. Specifically, the authors conclude that the specific projection from V1 to pDMSt is involved in learning the association between a cue and behavior using Nkx2.1 ReaChR in the pDMSt targeted to striatal interneurons and stimulated on every presentation of the cue for 1 second beginning 5ms before cue onset over 20 consecutive days of training. This optogenetic manipulation was shown to both activate and inhibit neurons in the pDMSt, which is consistent with direct activation of interneurons and the resulting indirect inhibition of pDMSt projection neurons. The assumption is that this inactivation is having its effect on learning by blunting the excitatory input from V1 neurons activated by the optogenetic cue. However, given that V1 activation is also activating other regions, the possibility exists that activation of these other brain regions also (or exclusively) encodes the cue and leads to learning, possibly by projecting back to the pDMSt. Thus, pDMSt inhibition is not specific to the studied pathway and could be disrupting input received from other regions influenced by the optogenetic cue.

The pDMSt manipulation is not sufficient to prove that the V1 to pDMSt pathway is exclusively learning this association as the impact of the pDMSt manipulation itself may have effects on other brain regions, such as other regions of the striatum or basal ganglia output regions (e.g., could the manipulation have an aversive effect which leads to lower reaching rate?). It would be informative to record from and manipulate other V1-recipient regions, as well as to understand how the pDMSt manipulation impacts other brain areas. They also perform control experiments where they demonstrate that pDMST stimulation does not affect the frequency of uncued movements.

We wonder if the same manipulation in other projection neurons of V1 or other areas of the striatum have a same or different effect?

We wonder why wasn't optogenetics used to test the necessity of striatum for specific epochs of the behavior (for example just during the cue period or only after the behavior was completed).

The authors should also discuss the possibility that in the post-learning phase striatum is still involved, but after learning the network is more robust to the optogenetic manipulation. Obviously this would change the narrative as evident by even the title “Striatum supports fast learning but not memory recall”. Striatum not being involved in post-learning is the most novel aspect of this study since as the authors state “a dominant theory is that the sensory cortex-to-striatum synapses are the storage site of learned cue-action associations.”

(2)

The paper is framed as an investigation into how the V1 to pDMSt pathway contributes to associating a cue and a behavior. To study this, the authors use an optogenetic activation of pDMSt projecting V1 neurons. The authors explain that this optogenetic intervention is used because visual cues themselves may activate many parallel pathways. However, this raises the question of whether this specific pathway is actually relevant to learning of cue-behavior associations in a physiological context (i.e., in the absence of the optogenetic intervention). It seems possible that the authors designed an experiment akin to a brain-machine interface study, which does not claim to mimic physiological forms of learning. This distinction is critical, as readers should be aware of whether to interpret the results as evidence of what is going on during learning of visual cue-behavior associations or as a specific study of a learning process that is imposed by the experimenter but does not mimic a physiological process

Performing the manipulation of pDMSt activity during an actual visual cue - behavior association learning would answer this. If the effect is the same, then we learn that this pathway can be used in physiological learning contexts. If the effect is different, then we should think of this experimental design akin to a brain-machine interface design which does not aim to mimic a physiological learning process.

(3)

The authors show that pDMSt manipulation during the second half of a daily training session does not impact cue-behavior learning within the session. While this does support their conclusion that changes in the first half a training session do not show regression when pDMSt manipulation is introduced, it is confusing when contrasted to the trial-to-trial results. Shouldn't we see a deviation during the second half of the training session, with less learning occurring compared to control? Alternatively, is it possible that pDMSt manipulation has no effect within a training session, but impacts “offline” learning? Investigating how much learning occurs “online” during training vs. “offline” between sessions, and the impact of pDMSt manipulation on these two forms of learning would be informative.

The author's definition of short term memory is a little questionable. They define short term memory as “Short term memory, defined here as an improvement in task performance acquired during the daily ~1 hour-long behavioral training session, might depend on pDMSt activity”. Isn't trial to trial effects a version of short-term memory?

(4)

I am not convinced by the conclusions from the results in Figure 5 that pDMSt neurons can represent differences between success/failure or cued/uncued and don't find it as strong evidence supporting the author's argument that pDMSt neurons are required to learn the association between a cue and a behavior.

The results in Figure 5 seem entirely expected. Movements during success and failure trials are distinct (second half of the

movement is different, e.g., mouse doesn't move hand to mouth, doesn't chew, etc.). Therefore, it seems like these results can simply be interpreted as evidence that different movements are represented by different activity in the pDMSt, without anything to do with success or failure. An analysis of non-chewing, post-movement periods with careful behavior tracking to confirm there are no movement differences would be more convincing evidence of success/failure coding.

Second, it also seems guaranteed that there is a difference in activity in pDMSt activity during cued or uncued trials, as the cue is an optogenetic activation of a population of V1 neurons that directly excites pDMSt neurons. Therefore, it seems unfair to call this "cue-related" activity as this confounds typical behavior cues and direct optogenetic activation of inputs. Therefore the logic in the paper of pDMSt encoding a cue in this context seems circular and confusing.

Furthermore, this paper is studying how a cue is associated with an event in the real world, which in this case is the presence of a food pellet. They state "According to reinforcement learning (RL) theory, reinforcement of an association between the cue and action depends on the outcome, such that only actions resulting in beneficial outcomes are reinforced". It was not clear why analyses were conditioned based upon success or failure. I would have thought analyses conditioned on whether the presence of a pellet was associated with a cue (and the temporal properties of this association) would have been more appropriate. Perhaps, an even more interesting conditioning would have been if the animals received tactile information about the presence of the pellet.

(5)

The analysis of the behavior is not comprehensive enough to support the claims. The authors state: "Mice learned to use this internal, optogenetic cue to guide the timing of the reach, without changing the previously established reach kinematics" (Line 122). However, a very clear drop in the ratio of drops to successes occurs after the cue is introduced. How can this change occur if not by a change in movement kinematics? In general, the analysis of behavior is severely limited. The quantifications of behavior are success/drop/miss rate and duration of 3 reach phases. Given the availability of tools to track detailed movements (which is already done in the work with DeepLabCut), quantification of the actual kinematics seems warranted for this level of work. The authors do provide reaching trajectories, which is a powerful visual tool, but further quantification is necessary. This should include consideration of both the spatial trajectory and velocity profile, including, e.g., quantification of single trial reaching amplitude, x/y/z reaching direction amplitude, magnitude of velocity & acceleration, consistency of spatial trajectory and/or velocity profile. Additionally, the pellet reaching task involves dexterous fine movements for grasping, which is not properly considered. Changes in how the digits open/close should be examined if claims about the movement kinematics changing/non changing are made, especially because these changes in digits may be controlled separately from the control of the outward reaching movement.

(6)

The main result is a deficit in learning with consistent pDMSt manipulation. However, the consistency of the results when considering individual animals is weak and the entire effect in Figure 3 seems to be driven solely by 3/7 mice (Figure 3c). The figure 3e histogram with mouse count should only include the proper control mice (7 control mice), not grouped in with other mice which seem to skew the result.

They should also discuss why some animals still don't learn even after the perturbation has ended.

(7)

I am unsure about the logic behind some of the method's decisions.

For example, the rationale for using a reaching task is unclear, would results be different for a different behavior, licking?

(8)

If this pathway is critical for cue association then I would be very interested if it was also involved in related phenomenon such as reversal learning and extinction.

(9)

If the 5s post outcome period is relevant as posited in Figure 5, pDMSt inhibition should be applied to this period to see if it is causal. It would be useful to determine when pDMSt inhibition impacts trial-to-trial behavior. What if applied right after reaching for 1s, what if applied 1s after reaching for 1s, etc. This would have fully taken advantage of their optogenetic method.

Minor comments:

Line 67: "To overcome the challenge imposed by the low firing rates of the SPNs (Hikosaka et al., 1989; Barnes et al., 2005; Tang et al., 2021), we recorded the activity patterns of one thousand putative SPNs in pDMSt during the behavior." I am unsure what the challenge is, is it in recording low FR neurons with in vivo electrophysiology? In understanding their function? Low firing rate neurons with bursty activity may have higher SNR in some contexts in comparison to high FR baseline neurons, so not sure what this general statement is saying.

Line 71: "SPNs in pDMSt encoded the combination of the reach, the outcome, and the cued versus uncued context of the reach. This combination predicted the behavioral change from trial to trial during learning, consistent with a specific function of pDMSt in incremental trial-to-trial learning." I may be missing something, but I don't see the analysis of physiology

measures predicting across-trial changes.

Line 273: "Moreover, the neural activity in pDMSt after a success contains lingering information about the presence or absence of the cue more than several seconds after the cue ends." This statement is based on the decoder which is able to classify trials based on the post-outcome period (5s after outstretched hand). However, there is no evidence that only the start (e.g., 200ms) of this period is used for classification. If a statement about lingering information for several seconds is made, the authors should show that using activity from smaller time bins continues to correctly classify cued/uncued trials after the cue ends (0-500ms after cue ends, 500-1000ms after cue ends, 1000-1500ms after cue ends, and so on).

Why is the activity in Supplemental Figure 2c and Supplemental 5h so different? The frequency of the response is completely different and 5h looks more like the distractor in figure 2c. This suggests that some quantification of how reliable the neural responses are to optogenetic stimulation in V1 is critical. How consistent is the induced "cue" activity between trials or between animals? If the frequency components are different across mice, might this impact how the activity travels to striatum, as temporal integration is an important characteristic of striatal responses? Could this explain high animal-to-animal variability?

Would like to see a plot of reaching times relative to when a pellet is randomly presented before any optogenetic cue pairing. Also would be interesting to see the same plot after pairing with optogenetic cue is learned (i.e., reaching to randomly presented pellet, no cue, but after cue is learned).

A "block" of inhibition trials, with inhibition on every trial was used to show that inhibiting pDMSt disrupts learning (Fig. 3), why was a block not used to show no effect on memory recall (Fig. 2). Would a block of trials have a different effect than interleaved trials? Additionally, it would add more understanding of the effect if the impact of inhibiting pDMSt was examined when the task parameters change and the animal needs to update the action (if the location of the food pellet is moved).

Was there a reason for choosing 250ms for the visual cortex stim? Would shorter work? Why is the pDMSt inhibition 1s long? Do the animals reach or react after the stim?

Figure 1: Is the reaction time with a peak of ~300-400ms expected? Is this similar to a visual stimulus reaction time?

Would like to know more information and numbers about the animals/sessions that were excluded, including the % of mice that failed the controls

Is there histology to confirm the targeting of inhibition in the pDMSt to areas that receive V1 innervation and are being stimulated? Where do the V1 collaterals go?

Would MSN activation directly in the pDMSt cause the behavior?

Does the pDMSt perturbation inhibit or perturb the probability of random, spontaneous reaches?

Color schemes should be followed more faithfully

What are the faint lines in Figure 3d?

Many of the plots in Figure 4 show some effect along the x and/or y axis - why are these other effects not discussed? How are they interpreted?

Version 1:

Reviewer comments:

Referee #1

(Remarks to the Author)

I appreciate the effort the authors put into this revision. I found the revised manuscript to be (1) strengthened with the addition of new data and analyses, and (2) much more clearly presented, with good referencing to controls and supplementary figures.

(Remarks on code availability)

NA

Referee #2

(Remarks to the Author)

I appreciate the work the authors have done to address the reviewers' concerns. This is substantial and has benefited the manuscript. The authors have largely addressed my concerns. I continue think that that the nature of the memory and striatal cell type are important factors here. Conclusions may be different if the behavior required a particular memory system, or were a specific cell type inhibited. Nonetheless, I agree these are substantial questions beyond the scope of this report.

General comments. In what follows the reviewers' words appear in blue.

Thank you for reviewing our manuscript. In this study, we demonstrated that activity in the posterior dorsomedial (pDMSt) part of the striatum is essential for learning, including trial-to-trial updates, but not for recalling a cue-response memory. The reviewers had many positive comments and feedback, including stating that “the data collected is impressive”, “the combination of optogenetic methods is innovative”, and “the studies are very nicely done and rigorous”. Nevertheless, they raised important concerns. The main criticisms were:

1. It is not clear how the results using the optogenetic cue (activation of visual cortex) relate to learning mechanisms that underlie the learning of an **external visual cue**.

To address this concern, we investigated whether activity in the posterior dorsomedial striatum is required for learning which external visual stimulus predicts the availability of a reward. Our findings confirm that activity in pDMSt is indeed essential to learn to that one visual stimulus signals pellet availability while another does not. In this newly added behavior, reaching is cued by an external visual cue that is a temporally varying pattern of light. A specific temporal pattern of light (the cue) must be distinguished from a different temporal pattern of light (the distractor). Only the presentation of the cue indicates that a food pellet is available. Control animals learned to discriminate between the two visual stimuli and learned to reach for the pellet only after the cue was presented. However, animals experiencing the optogenetic silencing of pDMSt at the time of presentation of the visual stimuli failed to learn the visual discrimination (**Supp. Fig. 9**). Moreover, once control animals had learned the discrimination, the cued response was unaffected by pDMSt silencing. These results mirror the results of pDMSt silencing in the original task, in which the optogenetic activation of visual cortex served as the cue. Thus, both tasks require pDMSt neural activity for learning but not recall, strengthening our conclusion that “Striatum supports fast learning but not memory”.

2. Our results indicate that an alternative brain pathway that is independent of pDMSt activity must underlie memory recall. However, in the original submission we did not provide **positive evidence for an alternative brain pathway underlying memory recall**.

To address this concern, we have now used muscimol to test the hypothesis that the superior colliculus underlies memory recall. We find that it does, as silencing this structure prevented recall of the learned task.

3. Reviewers 1 and 3 raised doubts about the novelty of the findings, citing previous studies that used the drug muscimol to inhibit the striatum and reveal selective deficits in reward-reinforced learning. We now discuss more fully the advantages of the optogenetic approach that we employ.

Because we used temporally precise perturbations, we were able to reveal the function of pDMS in trial-to-trial, outcome-dependent behavioral updating. Furthermore, we were able to pinpoint that it is the activity during and immediately following the action-outcome period that is necessary for behavioral improvement. In addition, this approach allowed us to demonstrate that as soon as a behavioral improvement is acquired, the striatal activity becomes dispensable for manifestation of this learning. As far as we know, none of these conclusions have been previously reported in the literature. Furthermore, these conclusions were impossible to reach using chronic inhibition or lesions approaches.

Previous studies using traditional loss-of-function methods to inhibit the posterior striatum did not address its role in short-term memory or outcome-dependent, rapid trial-to-trial updates. These processes require a fast and reversible approach, which is where our optogenetic loss-of-function method excels. Traditional methods lack the temporal precision needed to isolate specific cognitive processes—such as cue detection, action initiation, outcome-dependent behavioral updates, and short-term memory recall—that unfold over seconds to minutes during learning.

In contrast, our dual optogenetic cue and loss-of-function approach allows precise targeting of individual processes, such as outcome-dependent updates or short-term recall. Using this method, we demonstrate that the posterior striatum does *not* contribute to cue detection, action initiation, or short-term memory recall, but *does* play a critical role in rapid, outcome-dependent behavioral updates.

Thank you for these comments. Addressing them led to a significant expansion of the work and its importance. It also helped us more clearly position our results within the field.

Reponses to specific comments

Referee #1

In this manuscript, Reinhold et al use an optogenetically-cued head-fixed reaching task to study the role of corticostriatal inputs and striatal neurons (in the posterior dorsomedial striatum tail, pDMSt) in learning and “recall”. They find that while pDMSt neurons have specific patterns of activation during task trials according to the action/outcome, optical manipulations suggest the pDMSt is important for the learning but not recall of the task. The most potentially provocative conclusion is that the striatum is no longer required once the task has been learned.

Thank you for this summary. We agree that the conclusion is provocative and therefore should be of interest to the field.

Some of the questions addressed here have been tackled in other work, albeit with different methodology: (1) another publication lesioned the pDMSt projection neurons (Ruediger and Scanziani eLife 2020) and found impairment in learning of a visual discrimination task. (2) multiple papers from the Hikosaka group have shown value coding of visual stimuli in the analogous region of the caudate tail in non-human primate, using electrophysiology (eg Kunimatsu et al, PNAS 2021), and also shown the role of projections to the substantia nigra and superior colliculus in the associative value coding of visual stimuli (eg Amita et al, 2020). The same group has used muscimol to demonstrate a role of the caudate tail in performance of the visual/value task. Finally, several other labs have examined the role of corticostriatal inputs and SPNs (not in the tail of the striatum, however) in the learning of associative tasks, or in the performance of such tasks, once learned.

These are important studies of which we were aware. However, there are crucial distinctions between these and ours. First, none of this previous work examined (1) incremental trial-to-trial learning or (2) short-term memory recall of behavior improvements acquired within a day. The approaches that they used made it impossible to address these questions. To elaborate on each of these points:

1. We used an optogenetic approach with high temporal precision to selectively inhibit striatal activity for only 1 second during and immediately after the cue presentation. This allowed us to test if fast learning, on the timescale of seconds to minutes, requires striatal activity. We found that fast learning updates, which occur trial-to-trial and are outcome-dependent, do require striatal activity. Furthermore, we find that these fast, striatum-dependent updates accumulate over days and weeks to produce the long timescales of learning that were investigated in previous work. To the best of our knowledge, this dissection of the fast learning update is novel, and a necessary function of striatal activity in trial-to-trial updating has not been demonstrated. In addition, because the cue was carried by optogenetic activation of cortico-pDMS neurons, we were able to restrict the inactivation experiments to the region of the striatum that we defined as relevant to the task, greatly increasing the specificity of our findings. Lastly, as the reviewer stated, we were able to show that the same inhibition has no effect after learning has occurred.
2. None of the previous studies examined the contribution of posterior striatum to memory recall of a memory acquired recently (i.e., within the last hour). The optogenetic approach we developed provides temporal precision that enables us to address whether the improvement that a mouse has acquired within a single day’s training session is independent of the striatal neural activity at the end of the training session (i.e., after tens of minutes). We find that the short-term memory is *already* independent of striatum after this short time period. To the best of our knowledge, this is also a novel finding. Two other studies using an optogenetic inhibition of striatum found that striatum is required for learning but did not dissect whether this is because striatum supports the learning update or if striatum is required to express the acquired improvements in performance via short-term memory recall.

To address the specific papers cited.

1. Ruediger and Scanziani could not address fast learning and short-term memory, because they used an irreversible lesion approach. Furthermore, they studied the importance of neurons in the visual cortex *that project to* the pDMSt but not the importance of the pDMSt itself.
2. The Hikosaka work in monkeys is foundational and an inspiration for our study. Amita et al. is an exciting study that uses a *gain-of-function* experiment to show that neural activity in caudate tail is sufficient to bias behavior. However, in *loss-of-function* studies from Hikosaka and colleagues, they could not study fast trial-to-trial learning and its dependence on caudate tail, because they used muscimol, which acts over tens of

minutes, to inhibit caudate tail. Furthermore, our conclusion differs somewhat from Hikosaka's conclusion. Hikosaka generally proposes a memory function for caudate tail, but we do not find that pDMSt activity is necessary for memory recall. There are various possible explanations for this difference (e.g., monkey vs. mouse), but note that Hikosaka finds unilaterally inactivating one hemisphere with muscimol slows saccades to the left or right, but the monkeys can still respond to the cues in many contexts (Kim and Hikosaka, 2013, *Distinct basal ganglia circuits controlling behaviors guided by flexible and stable values*). It may be that unilateral inactivation leads to a bias in a lateralized task, but the important point is that the monkeys *can still saccade* to the correct target after muscimol inactivation.

This agrees with our finding that animals can still respond to the cue during pDMSt silencing. Another important difference is in the task design. Our mouse behavior represents a go or no-go (reach or no reach) design rather than a competition between lateralized target directions, and we inactivate pDMSt *bilaterally*. Thus, it may be that we do not observe any reaction time or bias effect arising from an imbalance in activity between the two hemispheres. Importantly, this discussion highlights the value to the field of using various methodologies to study striatal function.

3. Most other studies have focused on more anterior parts of the striatum and have found different functions for different parts. The posterior striatum in the mouse, which we study here, is relatively understudied, and its function in learning versus memory was not known before our work. Furthermore, it is the only region to receive visual inputs.
4. Work from the Zador lab showed that cortico-striatal plasticity correlates with learning but did not test the necessity of striatal activity for learning or memory recall. Furthermore, the model that they propose is fundamentally at odds with our findings. They, and most models of the striatal contributions to learning, posit that it is plasticity of cortico-striatal synapses that mediates the changes in behavior. This model is rejected by our findings.
5. Neely et al. find a role for dorsomedial striatum in learning that is consistent with our results. However, they do not attempt to probe the contribution of striatal activity to fast trial-to-trial updates versus short-term memory recall.

Moreover, we added three new experiments that provide important further insights.

First, we use the optogenetic loss-of-function method to identify *when* neural activity in pDMSt is needed for the learning update, by varying the timing of pDMSt inhibition with respect to the cue and the reach. This takes advantage of the unique temporal precision of our optogenetic loss-of-function method.

Second, we find a brain structure that contributes to memory recall *after* learning.

Third, we develop a new behavior that uses an *external visual cue*, and we show that learning this behavior also depends on pDMSt. Therefore, we are studying a general brain mechanism for learning about sensory cues. This is a major advance, because, to our knowledge, no one has previously described a pDMSt-dependent visual discrimination behavior in the mouse. Prior work has identified *visual cortex*-dependent behaviors (e.g., Ruediger), but a visual discrimination that is specifically dependent on pDMSt is new.

We have improved the discussion of our work in the context of these important antecedent studies.

This study has a somewhat unusual design, which may make some skeptical of the interpretability of the findings. However, the authors have included many experimental and analytical controls in the Supplemental figures which strengthen the manuscript and are important in understanding their findings. Overall, the questions and experiments outlined in the paper are interesting and relevant to those interested in reinforcement learning and the role of the tail of the striatum.

Thank you. We added references to these important controls in the main text, improving the ease with which a reader can find relevant controls.

However, as currently presented the manuscript is hard to parse.

Thank you for the feedback. We have tried to improve the presentation in this resubmission.

Major Comments:

1. Though the data collected is impressive, I found the presentation to be challenging to read and process. I had to put in a lot of extra time to find the experiments/analyses I suspected the authors performed, but were not

mentioned in the main text. Examples include the GtACR experiments, the no-virus controls, and the validation that LED light (outside the head) produced similar responses to the activation of visual cortex to pDMSt neurons, but there are others. The main figures are highly compressed and there are numerous supplements, and quite a bit of critical information is in the Figure legends (including the Supplementary Figure legends) or even in the Methods (not mentioned at all in the main text), making the Results section hard to read by itself. I am not sure if this amount of information can be conveyed in this format with another approach, or just doesn't lend itself to this format.

Thank you for the feedback. We have worked to improve the presentation by making references to these controls more prominent in the main text.

2. The authors use a unique “cue” in their cued reaching task – optogenetic stimulation of visual cortex neurons that project to pDMSt, and go on to use this cued task to demonstrate the role of the pDMSt. However, would transient inhibition of pDMSt neurons prevent traditional visually cued learning?

As described above, we developed a new behavioral paradigm using a traditional visual cue to answer this question (see **Point 1 in the General Comments** section above). We find that natural visual cue learning is by a mechanism quite similar to learning the optogenetic cue. Hence the optogenetic cue seems to be a reasonable model for visual discrimination learning.

Moreover, we believe that it is meaningful to study *how* the brain can accomplish behavior. Each behavior is unique. Our study clearly shows that the brain is able to use a mechanism that depends on the pDMSt to learn but not recall after learning, for both an internal cue and an external cue. This demonstrates an important aspect of posterior striatal function, i.e., that it tends to participate during learning but not memory.

We feel that the addition of another, distinct behavior further strengthens our conclusion. This conclusion, “*Striatum supports fast learning but not memory*”, is now valid for two different behaviors using two quite different cues.

3. The optogenetically cued task is quite complicated, and much of the important information for understanding the task and training parameters is in the Supplement or Methods sections. As a reader I would have benefited from having the setup schematic (found in the Supplement) in the main figure.

Thank you for this suggestion. We have added a setup schematic to **Figure 1**.

4. Many important controls are mentioned after the main behavioral findings are presented (eg that reaches increase at the optogenetic “cue”). I would have found it helpful to mention the control findings in parallel with the main findings, as I became impatient while reading, wondering if the authors had done appropriate controls (which they had), and how these compared to the primary analyses. Also, in line 142, it says sessions were excluded if they “failed these controls” – what does that mean and how often did it happen?

Thank you. We now discuss the controls in parallel with the main findings. Mice failed these controls in less than 15% of the sessions. We now include this quantification in the main text (line 150).

5. The authors test the idea that pDMSt projection neurons are important for learning of the cued reaching task not by direct optogenetic inhibition of SPNs, but by activation of GABAergic interneurons (Nkx2.1 + ReaChR). Is it possible that in addition to inhibiting SPNs, the released GABA inhibits local dopamine terminals and might impair learning by reducing the dopamine release that occurs with cued/correct trials?

The reviewer raises an interesting question which applies to the many studies that use activation of GABAergic interneurons as a way to achieve fast, robust, and local inactivation. Luckily, we also performed the experiment using GtACR2 suppression of pDMSt activity. We see a disruption of learning updates using GtACR2 suppression of striatum, lending support to the idea that the suppression of SPNs is the mechanism of the learning disruption. We agree that this control was previously difficult to find, and we now bring it forward and discuss it in the main text (lines 263-267).

By careful reading of the Supplements and Methods I discovered that the authors also performed experiments with GtACR in SPNs (this is not mentioned at all in the main text), which would tend to address this question. However, as this experiment is not explained in the main text it's very labor-intensive as a reader to determine if it was done the same way or was up to the same standards as the experiments in the main text.

We now discuss this control in the main text (lines 263-267).

6. Did the pDMSt inhibition experiments produce any other behavioral effects (eg overall motor activity/freezing), outside of cued reaching?

No, we now include videos of the mice to demonstrate that there were no other behavioral effects (**Supplementary Videos 8-10**).

7. In line 233, the authors introduce the section with dLight photometry and extracellular electrophysiology in the pDMSt by saying that they made recordings in “beginner” and “expert” mice. Recording in both of these phases seems important given the findings with optogenetic manipulations in Figure 3. However, I could not figure out what recordings or comparisons were made in “beginner” or “expert” mice, and if there were any differences in neural activity patterns in these two phases. I think this type of comparison would be very important in light of their negative results for optical manipulations during “recall” or performance of the task, once learned.

We needed to record from mice during learning, instead of in expert mice only, because this is when we see significant behavioral updates contingent on striatum neural activity. We now describe the fraction of beginner, intermediate and expert mice (lines 286-287).

We attempted to answer the reviewer’s question by investigating if there were patterns of activity specific to the different learning stages. We saw some interesting preliminary hints. However, even with 17 mice, we did not feel that there were enough mice at each learning stage to draw strong conclusions about how the neural activity patterns change with learning. This is a fascinating question, which, we hope the reviewer agrees, is beyond the scope of the paper.

8. The authors find that once the task is learned, inhibiting the pDMSt doesn’t impair performance, and conclude that axon collaterals of visual cortex neurons that go to locations other than pDMSt may be important at this stage. However, they don’t provide any direct/positive evidence for this speculation. In the discussion they suggest it might be via pathways to other visual cortical neurons and/or the superior colliculus, but don’t provide direct evidence for this idea.

Having ruled out a function of the pathway to pDMSt during recall, we are left with the explanation that a *different* pathway underlies recall. We now provide *direct positive evidence* for this idea by inhibiting the superior colliculus using muscimol (**Supp. Fig. 7**). We show that this disrupts the cued reaching after learning. This also serves as a positive control for disrupting the behavior after learning. In striking contrast, inhibiting pDMSt had no effect after learning, although inhibiting pDMSt prevented learning. Therefore, striatum and superior colliculus have complementary functions in the task. We hope this directly satisfies the reviewer’s concern.

Minor comments:

1. The authors describe the optogenetic cued trials, stating on line 117 that “the pellet was unavailable before cue onset” and then that the pellet “became available shortly before cue onset” (0.22 seconds). These statements seem to be contradictory. I suspect the latter statement is more accurate and this sentence should just state that the pellet became available 0.22 seconds before each cue onset.

You are correct. We have now fixed this (lines 121-123).

2. Is it meaningful that the performance of controls in the ReaChR experiments (Figure 3), in terms of reaches/sec after the cue, or *d prime*, was lower than in animals shown in Figure 1?

As explained in the legend, **Figure 1** shows the learning curves for mice that ultimately learned. **Figure 3e** shows every mouse that we attempted to train, including animals that learned and animals that did not learn. This can be seen in the black histogram at the bottom of **Figure 3e** (some mice did not learn). There are two kinds of control mice: mice that were not run as double-blinded controls at the same time as the pDMSt silencing mice and mice that were run as double-blinded controls at the same time as the pDMSt silencing mice. Between these two groups of controls, there was no significant difference in learning. We have now included statistics in the legend (**Figure 3**) to show that there was no significant difference between these two groups of controls. Both learned at a similar rate.

Referee #2 (Remarks to the Author):

This interesting report demonstrates the selective necessity of neurons in the tail of the posterior dorsomedial striatum for learning that supports the ability to use optogenetic activation of the visual input to this structure as a discriminative cue signaling reinforcement of a reaching action.... Although there has been a long history of assessment of striatal involvement in instrumental learning and memory, this study adds information on this particular subnucleus of the striatum and its potential selective involvement in learning but not memory recall. The use of an optogenetic cue potentially isolates the pathway through which visual cue information can get to the pDMSt to support visual discriminative instrumental learning. The combination of optogenetic methods is innovative. The addition of high density recordings adds an important layer of converging evidence for the learning findings and provides useful context for how pDMSt SPNs might contribute. There are many strengths to the experimental design, including controls to validate the mice were using the optogenetic stimulation as a cue and that the optogenetic inhibition effects are not secondary to effects on kinematics. Validation of the optogenetic manipulations is a strength. The studies are very nicely done and rigorous.

Thank you for these kind words and for highlighting the novelty of our work in the context of previous studies.

The paper is well written. The results are novel and potentially important...

Thank you. We hope that the clarity is even better with this version.

...but limited in 3 ways.

First, it is clear that the pDMSt SPNs mediate learning in this task, but whether they do not mediate memory retrieval is not fully demonstrated.

Please see our full response below. In short, we now show that (1) inhibition of pDMSt SPNs was equally strong both during learning and during memory retrieval, and (2) blocks of repeated inhibition did not lead to deficits in memory retrieval. Hence, the SPNs were inactivated during memory retrieval, yet the mice were able to retrieve the cue-reach associative memory without any deficits.

Second, it is not clear if these findings apply selectively to optogenetic cue discriminative learning, or whether this region is also important for natural discriminative stimulus instrumental learning.

Please see **Point 1 in the General Comments** section above, which discusses the new experiments that we have added showing the importance of pDMSt activity for learning in a visual discrimination task.

Third, there is no information on the potentially unique or complementary contributions of the functionally and anatomically distinct subtypes of striatal projection neurons.

Please see an extensive discussion of these points below. We have added several important experiments to address the reviewer's concerns. To summarize: First, we now show that the inhibition of pDMSt is equally strong during vs. after learning by *in vivo* electrophysiology. Second, we develop a new behavior paradigm using an external visual cue to show that a natural visual behavior employs the same brain mechanism. This was a major undertaking, and we feel that this is an important advance. Toward the reviewer's third point, we functionally define two classes of striatal neurons in **Figure 5**. However, the assignment of these two functional classes onto molecularly labeled striatal projection neurons (e.g., direct vs. indirect pathway neurons) is beyond the scope of the current work. See many more details in-line below.

Additionally, more information is needed to understand what region of the striatum is targeted and distinguish the findings here from what is already known about the posterior DMS and tail of the striatum. I expand on these below.

See full response below. We now give an anatomical definition of pDMSt in **Supp. Fig. 2**, and we provide a thorough discussion of our findings in the context of the field in the Discussion (see more below).

1. Selective function in learning and not memory recall. There are several potential alternative explanations for the lack of an effect of pDMSt on performance during the memory-recall tasks. One simple one is differences in the amount of inhibition. Inhibition was given on all trials for learning experiments v. a random subset of trials for performance/memory recall experiments. This difference could explain why inhibition of pDMSt SPNs was effective when applied during learning, but not memory recall.

Is the putative selective effect on learning due to more inhibition or the lack of effect on recall due to an insufficient amount of inhibition?

We do not think that this is a possible explanation. We have added figures showing, with *in vivo* electrophysiology, that pDMSt inhibition is equally strong during learning and after learning (i.e., during recall) (see **Supp. Fig. 5f** and explanation in the main text line 170). Thus, we equally inhibit the striatum in both cases. Moreover, our conclusion—that pDMSt inhibition blocks learning and trial-to-trial updates but does not affect reaching or recall—is based on multiple independent experiments (**Figure 2, Figure 3, Figure 4**) and remains valid regardless of the fraction of trials in which the inhibition was applied.

A stronger and/or longer inhibition during this test phase (even with something simple like muscimol) would provide converging and stronger evidence for the null effect.

To address this possibility, we analyze blocks of consecutive trials with pDMSt inhibition on every trial during recall (just like the blocks of inhibition trials during learning). In these blocks, there is no effect on recall (**Supp. Fig. 6f**). Hence, the lack of effect on recall is independent of whether the inhibition was present on a few or many trials. We feel this is a better way to address the reviewer's concerns than a muscimol injection, as the effects of exactly the same manipulations are now shown to have effects during learning but not after.

It is well known that instrumental behaviors of the sort studied here can be controlled by multiple memory systems that rely on distinct brain circuits, centered on distinct subregions of the striatum. Action-outcome (in this case cue-action-outcome) requiring the pDMS or S-R habits requiring the DLS being a classic example of this. Another is model-free v. model-based reinforcement learning. The nature of the memory/memories used to guide behavior was not evaluated here. It is presumed to be a sensory-motor association, but it is not clear that this rather than an outcome-based memory is controlling the behavior. Most likely this behavior could be control by both of these memory systems. It would be very interesting to probe this more deeply.

The reviewer suggests that either a sensory cue-to-motor action association or a sensory cue-to-outcome association is learned. In the first case, the mouse learns to associate the cue with a reach. In the second case, the mouse learns to associate the cue with a food pellet, and then infers that a reach is necessary to obtain the food. It seems that either of these memory systems could be used to learn the association. In either case, the cue is being linked to either the reach or food, because the mice reach specifically in response to the cue. Hence, a pure action-outcome memory does not fit this behavior.

Understanding the type of learning supported in this region and how this region supports learning would be impactful. But more importantly, the possibility that this behavior could be controlled by multiple memory systems is critical for interpreting the null effects of inhibition during the performance phase. If the behavior can be controlled by 2 (or more) different forms of memory, then it could well be the case that pDMSt SPNs do mediate the recall of one type of memory (maybe the action-outcome association), but upon inhibition of these neurons perform shifts to be controlled by another (perhaps stimulus-response, model-free, or sensorimotor) memory.

We agree that some other memory system takes over during recall. This is, in fact, one of the main conclusions of our paper.

The reviewer is suggesting that, on pDMSt inhibition trials or in sessions with pDMSt inhibition after learning, the behavior strategy switches from cue-motor to cue-outcome. There is no hint in our data of a strategy switch. We do not see (1) a different reaction time on control vs. pDMSt inhibition trials, or (2) some brief period when memory recall depends on pDMSt, before the animals successfully make the strategy switch. However, *we agree that some other memory system takes over during recall*. We suggest that different brain regions enact different kinds of memory. The necessity of pDMSt for learning but not executing the association suggests that some other memory system *must take over for recall*, as pDMSt activity is no longer required. Identifying exactly *which* memory system takes over is a question that we have now attempted to address by injecting muscimol into superior colliculus. We find that superior colliculus *is* required after learning. As the reviewer suggests, it is possible, of course, that pDMSt would support memory recall in a totally different kind of behavior. To begin to address this concern, we have added an external visual discrimination behavior and show that it, too, depends on pDMSt for learning but not memory recall. Therefore, the dissociation between pDMSt's crucial function in learning but its dispensability after learning holds for two different behaviors.

However, it is also worth noting that the reviewer's proposed explanation is non-falsifiable. One can always propose that lack of effect of a manipulation is because some other system takes over to compensate for it. The

only way that we know to determine if a system is essential for a behavior is to inhibit it and see if the behavior goes away. In our case, one can conclude that pDMSt activity is not essential for recall.

Test of outcome sensitivity (e.g., with devaluation), as have been used classically and more recently to delineate striatal function in instrumental learning and memory may be useful in this regard. Such tests have previously shown the pDMS to be critical for both the learning and performance of action-outcome memories. I am tending to assume that the learned association here is instrumental, given the contingent setup. However, it is also plausible that this task can be performed in a Pavlovian manner, further leading to questions about what memory/memories are learned here and what aspects of these memories are pDMSt neurons involved in learning and/or using.

What follows is a discussion of our observations with relevance to the reviewer's questions: "*What is the type of learning?*" and "*Which association is being learned?*". We also now provide a discussion of this issue in the main manuscript (lines 400-407).

The reviewer raises an interesting set of questions. As the reviewer points out, this task requires the learning of several associations: (Type 1) The action to the outcome, i.e., reach to food. (Type 2) A cue to an action, i.e., a cue to the reach. (Type 3) The pattern of activity in the brain as a unique cue, i.e., sensory pattern to cue, or, put another way, *specific* cue to some action.

Observation A:

We found that spontaneous, uncued reaches *were* sensitive to devaluation: i.e., spontaneous reaches decreased when the mice were fully sated and not hungry. However, cued reaches were *not* sensitive to this devaluation: i.e., some expert mice, even when fully satisfied, continued to perform cued reaches to grab pellets *without consuming those pellets*. The animals simply dropped the pellets after grabbing them (**Supplementary Videos 11-12**). However, pDMSt inhibition affected neither reach type, i.e., cued and outcome-insensitive, or uncued and outcome-sensitive. This is *unlike more anterior DMS*, as the reviewer points out. We favor the idea that the pDMSt is functionally distinct from DMS (it is clearly anatomically distinct). This is consistent with the conclusions of a recent preprint (Cover, Elliott, Preuss and Krauzlis, 2024 *bioRxiv*) from another group.

Observation B:

We have added a new set of experiments using an external visual cue. Our new results, discussed in more detail above (see more about this in the General Comments section), now provide additional insights into *what type of associative learning* is affected by pDMSt silencing. We trained mice to discriminate two external visual cues that were different temporally varying patterns of light (see more about this at the top in the General Comments section labeled "External Visual Cue"). When we silenced the pDMSt, the animals learned to reach in response to *both* the cue and the distractor, but these animals could not learn to *discriminate between* the cue and distractor. Hence the animals did learn visual detection, yet the animals failed to learn to associate a *specific* cue with the reach. Note that the learning deficit during pDMSt silencing is not a visual perception or temporal integration deficit, because these animals showed no deficits in discriminating cue versus distractor *after learning* even during pDMSt inhibition (**Supp. Fig. 9d**).

Taken together, these observations suggest that pDMSt underlies the learning of a Type 3 association, in which a specific cue should be paired with the action/outcome. Interestingly, this seems consistent with the fact that pDMSt, uniquely of all striatal regions, receives sensory information directly from primary visual cortex.

We now provide a discussion of this in lines 400 to 407.

The behaviors used here cannot distinguish between model-based or model-free reinforcement learning.

An explanation for the null effect of pDMSt inhibition on memory recall/performance tests is that it is due to axon collaterals of cortico-striatal neurons triggering the cued action via another brain pathway, but this is not substantiated.

We now provide direct evidence for another pathway triggering the cued action. Injections of muscimol into the superior colliculus led to a decrease in cued reaching after learning (**Supp. Fig. 7**). Hence, we substantiate this possibility.

2. Do the findings only apply to this form learning? It seems important to know whether the proposed selective function of the pDMSt in learning, not performance of discriminative cue instrumental learning is specific to learning induced by optogenetic activation of visual input to the pDMSt or would apply to other forms of visual

cue discriminative instrumental learning or learning of discriminative cues of other modalities. The latter may be likely given prior evidence that mice can learn to respond to optogenetic stimulation of auditory inputs to the tail of the striatum. Prior work has shown that mice can perceive optogenetic manipulations to the brain and use them to guide actions (<https://www.eneuro.org/content/9/3/ENEURO.0216-22.2022>). This seems highly relevant and should be cited here. The results of this prior work bring up questions here about whether activation of the visual cortex projection to the pDMSt is critical or whether similar results would be obtained with optogenetic stimulation of other parts of the brain. It is also important to understand the role of this region in natural cue discriminative instrumental learning.

Please see **Point 1 in the General Comments** section at the top, which addresses this point.

Thank you for the citation. We now include it.

Whether the same learning mechanism applies if the optogenetic cue were transposed to another part of the brain is an interesting question beyond the scope of the current paper. Furthermore, the answer to that question does not impact the results of this study. However, we now clearly establish a role for striatum in learning both this optogenetic cue and an external visual discrimination.

3. SPN subtypes. The striatum is well known to have anatomically distinct direct and indirect projection pathways to basal ganglia output structures. Neither the recording nor manipulation components of this study distinguished between cell types, which is important given functional differences in these output pathways. Some understanding of the potentially unique, similar, or complementary functions of the direct v. indirect pathway SPNs would enhance the potential impact of this report.

We agree that this is a very interesting question. However, given the complex genetics already being used, we do not see a way to realistically carry out this experiment. We are keeping a close eye on the development of cell-type specific AAVs that exploit enhancers to restrict expression of proteins to either direct or indirect pathway neurons.

For the loss-of-function experiments, which are necessary for understanding the native function of a neuronal population, we intentionally decided to focus on the function of the striatal activity, as opposed to of the activity of specific neuron classes. Previous studies using approaches that inhibit either the direct or indirect pathway neurons have a confound of ectopic activity in the non-suppressed pathway. Our optogenetic loss-of-function method (ReaChR in interneurons), which inhibits *both* direct and indirect pathway neurons of the striatum, does not have this confound. As the primary motivation for this study is to investigate the function of pDMSt in learning or memory, we favor our simple loss-of-function approach, which we validate with *in vivo* electrophysiology. Intriguingly, we do identify two functional sub-groups of neurons in Figure 5. However, investigation of the function of direct or indirect pathway would be interesting but is beyond the scope of our current question.

4. Histological validation of viral expression and fiber placement should be in the main text figures so readers don't have to go hunting for this important information. In addition to representative examples, it would also greatly help to have schematic maps of viral expression spread and fiber placement showing the placements for all subjects. In some cases, it appears as if there might be some spread to the GPe.

We now show example histology and fiber placement in the main **Figure 2a**, as the reviewer suggests. Also, as the reviewer suggests, we have now also added a schematic of fiber placement in **Supp. Fig. 8b**. The reviewer is correct that in some mice there was leak into GPe. However, this was not consistent across mice and did not correlate with the behavioral effect on learning. This can be seen in **Supp. Fig. 8c**, where all mice had impaired learning but only 4 of 9 mice had leak into GPe.

5. Relatedly, how distinct is this subregion of the striatum from the previously studied posterior DMS and tail of the striatum? The pDMS has been considerably implicated in instrumental learning similar to that evaluated here (e.g., work from Corbit, Janak, Balleine, Yin etc.), including the performance of instrumental behavior. The tail of the striatum has also been implicated in instrumental behavior- including the performance of instrumental discriminations similar to those studied here (e.g., Znameskiy & Zador 2013). Some discussion of these findings in light of this prior work is needed to better contextualize the findings and understand what is novel here.

Thank you. We have added a discussion of how pDMSt is distinct (lines 406-407) and the literature in the main text (Ruediger and Scanziani: lines 370-373, Hikosaka: lines 410-413, Zador: lines 359-362, Neely et al.: lines 386-388). Importantly, 1. Znameskiy and Zador never test a *causal* role for the pathway through the tail of the striatum in the performance of the behavior (Znameskiy and Zador show that *cortico-striatal neurons* are involved, but Znameskiy and Zador *do not inhibit striatum*). 2. The previous work on pDMS (Corbit, Janak,

Balleine, Yin, Neely) indicates that pDMS is required for instrumental learning, consistent with our work, but these previous studies could not test A. the function of striatum in fast trial-to-trial updates, B. the *specific time window* when pDMSt is needed for learning, and C. the function of striatum for the execution of short-term memory. We now find that A. the pDMSt neural activity underlies fast trial-to-trial updates, B. it is neural activity *specifically* during the reach and post-outcome time window (not during the cue), which is required for the learning updates, and C. pDMSt is required to learn but not required to execute short-term memory after just a few tens of minutes (by the end of the training session). These results clarify the pDMSt's specific function in learning as compared to memory.

The reviewer also asks a good question: "How distinct is this subregion of striatum?" We now provide a supplementary figure (**Supp. Fig. 2c**) that defines the pDMSt, which receives V1 input, with respect to classically defined pDMS and classically defined "tail of striatum". Moreover, we cite Khibnik, Tritsch and Sabatini (2014), which previously characterized the subregion of striatum that receives the V1 input.

6. P values are provided in the figures, but full statistical reporting needed to evaluate the support for the claims appears missing from the main manuscript. It is not always clear if trials or subject is the N unit.

We added this.

7. Numbers of subjects that are male v. female is needed in the figure legends.

We added this.

Referee #3 (Remarks to the Author):

Reinhold et al. developed a unique paradigm to study how visual cortex projections to the posterior dorsomedial striatum tail (pDMSt) contribute to learning an association between a cue and a presentation of a food pellet a mouse can reach and retrieve.

Thank you for highlighting the novelty of the approach.

The physiological relevance of the experiments is uncertain and there are several major concerns about the logic of the conclusions. Given this, without further work (for example, comparing these results to inhibition of superior colliculus as mentioned in the discussion) the current conclusions seem incremental. These comments are outlined below:

We added a new set of experiments to compare the inhibition of pDMSt to the inhibition of superior colliculus, as the reviewer suggests (**Supp. Fig. 7**). We also examine the effects of pDMSt silencing in a new visual discrimination task and reach the same conclusions (**Supp. Fig. 9**). Please see the General Comments at the beginning of this document.

The authors are careful to point out that optogenetic activation of pDMSt-projecting V1 neurons also activates other regions due to axon collaterals of these neurons. However, this "off target" activation presents an important confound in their main conclusions. Specifically, the authors conclude that the specific projection from V1 to pDMSt is involved in learning the association between a cue and behavior using Nkx2.1 ReaChR in the pDMSt targeted to striatal interneurons and stimulated on every presentation of the cue for 1 second beginning 5ms before cue onset over 20 consecutive days of training. This optogenetic manipulation was shown to both activate and inhibit neurons in the pDMSt, which is consistent with direct activation of interneurons and the resulting indirect inhibition of pDMSt projection neurons. The assumption is that this inactivation is having its effect on learning by blunting the excitatory input from V1 neurons activated by the optogenetic cue. However, given that V1 activation is also activating other regions, the possibility exists that activation of these other brain regions also (or exclusively) encodes the cue and leads to learning, possibly by projecting back to the pDMSt. Thus, pDMSt inhibition is not specific to the studied pathway and could be disrupting input received from other regions influenced by the optogenetic cue.

We apologize for not being clear. Our conclusion is that neural activity in pDMSt (specifically in the 1 sec around the cued reach) is required for learning. Our experiments showing that transient pDMSt silencing eliminates trial-to-trial improvements as well as within-session and across-session learning, in support of this conclusion. There

may indeed be multiple pathways that bring information about the cue to pDMSt, but this, if true, does not impact our conclusion.

The pDMSt manipulation is not sufficient to prove that the V1 to pDMSt pathway is exclusively learning this association as the impact of the pDMSt manipulation itself may have effects on other brain regions, such as other regions of the striatum or basal ganglia output regions (e.g., could the manipulation have an aversive effect which leads to lower reaching rate?). It would be informative to record from and manipulate other V1-recipient regions, as well as to understand how the pDMSt manipulation impacts other brain areas. They also perform control experiments where they demonstrate that pDMST stimulation does not affect the frequency of uncued movements.

We do not claim that the V1 to pDMSt pathway is “exclusively learning the association”. In fact, our results suggest otherwise. Our conclusion is that although activity in pDMSt is necessary to learn the association, it is not necessary to express the learning, suggesting that the V1 to pDMSt pathway does not exclusively experience the learning-related plasticity; instead, other pathways must contribute. We now make this clearer in the Discussion.

In addition, in response to the reviewer’s concern, we added an experiment that manipulates a different V1-recipient region, the superior colliculus (SC). We find that muscimol inhibition of SC disrupts cued reaching after learning (**Supp. Fig. 7**). Thus, we demonstrate a clear dissociation between the functions of striatum and SC in this task. This dissociation places the distinct function of pDMSt in a clearer context.

The reviewer also asked if the inhibition of pDMSt might be aversive, leading to a lower reaching rate. Figure 4 indicates that this is not the case. If inhibition of pDMSt reduced reaching rate, then Figure 4b should show a decrease in the reach rate after pDMSt inhibition. However, in Figure 4b, the reach rate is *higher*, not lower, after pDMSt inhibition. Therefore, the effect of pDMSt inhibition is not simply an aversive effect. The effect of pDMSt inhibition depends on the outcome, consistent with a behavior update that underlies learning.

We wonder if the same manipulation in other projection neurons of V1 or other areas of the striatum have a same or different effect?

As a first step to address the reviewer’s question, we added an experiment that tests the function of a different region downstream of V1. We find that the superior colliculus, which is downstream of V1, has a function *after* learning (**Supp. Fig. 7**).

Previous work in the field suggests that other parts of the striatum have different functions. Moreover, other work suggests that the other projection neurons of V1 have different functions. To study all the other projection targets of V1 would take decades of work.

We wonder why wasn’t optogenetics used to test the necessity of striatum for specific epochs of the behavior (for example just during the cue period or only after the behavior was completed).

This is an excellent suggestion, and we added exactly this major experiment (**Figure 4ef**). Collecting this data was a major undertaking, as each examined time point requires a full dataset which must be collected in a sparse subset of trials.

Interestingly, we find that the neural activity in pDMSt during the cue period is not required for the learning update. It is the neural activity in pDMSt after the outcome, called the post-outcome period, that is required for the learning update. This experiment highlights the power of the optogenetic loss-of-function approach. We can target this loss-of-function to very specific time windows to discover that a specific pattern of neural activity in a specific part of the brain at a specific time is needed to drive the learning update.

Note also that our behavioral results are consistent with *in vivo* electrophysiology recordings in **Figure 5**, which show that pDMSt neural activity in the post-outcome period predicts the behavior update.

The authors should also discuss the possibility that in the post-learning phase striatum is still involved, but after learning the network is more robust to the optogenetic manipulation. Obviously this would change the narrative as evident by even the title “Striatum supports fast learning but not memory recall”. Striatum not being involved in post-learning is the most novel aspect of this study since as the authors state “a dominant theory is that the sensory cortex-to-striatum synapses are the storage site of learned cue-action associations.”

We now show *in vivo* physiology specifically in well-trained mice. The inhibition of SPNs is still >80% in well-trained mice (**Supp. Fig. 5f**) and the cued response is still disrupted (**Supp. Fig. 5f**). This strongly supports our conclusion that the striatum is not involved in post-learning recall.

To further satisfy the reviewer's concern, we now point out that (1) disrupting >80% of the striatal activity does not affect recall, but (2) there remains a possibility that the remaining 20% of the uncued activity in striatum was sufficient to fully recapitulate the entirety of the cued reaching behavior. We now mention this possibility explicitly in lines 188-191. However, note that this same manipulation prevents learning, consistent with a significant loss of striatal activity, and most optogenetic inhibitions published in the literature are far less effective than our >80% suppression.

(2) The paper is framed as an investigation into how the V1 to pDMSt pathway contributes to associating a cue and a behavior. To study this, the authors use an optogenetic activation of pDMSt projecting V1 neurons. The authors explain that this optogenetic intervention is used because visual cues themselves may activate many parallel pathways. However, this raises the question of whether this specific pathway is actually relevant to learning of cue-behavior associations in a physiological context (i.e., in the absence of the optogenetic intervention). It seems possible that the authors designed an experiment akin to a brain-machine interface study, which does not claim to mimic physiological forms of learning. This distinction is critical, as readers should be aware of whether to interpret the results as evidence of what is going on during learning of visual cue-behavior associations or as a specific study of a learning process that is imposed by the experimenter but does not mimic a physiological process. Performing the manipulation of pDMSt activity during an actual visual cue - behavior association learning would answer this. If the effect is the same, then we learn that this pathway can be used in physiological learning contexts. If the effect is different, then we should think of this experimental design akin to a brain-machine interface design which does not aim to mimic a physiological learning process.

See the **Point in the General Comments section**. We added an experiment testing an actual visual cue and recapitulate our main findings that learning but not recall (**Supp. Fig. 7**) requires pDMSt activity.

(3) The authors show that pDMSt manipulation during the second half of a daily training session does not impact cue-behavior learning within the session. While this does support their conclusion that changes in the first half a training session do not show regression when pDMSt manipulation is introduced, it is confusing when contrasted to the trial-to-trial results. Shouldn't we see a deviation during the second half of the training session, with less learning occurring compared to control? Alternatively, is it possible that pDMSt manipulation has no effect within a training session, but impacts "offline" learning? Investigating how much learning occurs "online" during training vs. "offline" between sessions, and the impact of pDMSt manipulation on these two forms of learning would be informative.

Learning still occurs in the second half of the session, because the inhibition of pDMSt is interleaved in a small subset of trials rather than present on every trial. As can be seen from the error bars, the difference the reviewer suggests cannot be observed in this dataset (insufficient statistical power). However, if we compare the learning curves over many days for all control mice versus the interleaved inhibition of pDMSt, the reviewer is exactly right that the interleaved inhibition cohort does learn more slowly than the control cohort. We added this graph to the supplementary material (**Supp. Fig. 8d**).

The author's definition of short term memory is a little questionable. They define short term memory as "Short term memory, defined here as an improvement in task performance acquired during the daily ~1 hour-long behavioral training session, might depend on pDMSt activity". Isn't trial to trial effects a version of short-term memory?

We apologize for the confusion. The trial-to-trial analysis does not address memory at all. It addresses only learning. This is because we are measuring learning from trial n to the next trial $n+1$, and the pDMSt inhibition is on trial n . There is no pDMSt inhibition on trial $n+1$. Therefore, pDMSt activity is needed to *acquire* the trial-to-trial memory, not to express that memory. However, we understand the confusion and have now clarified this in line 263.

(4) I am not convinced by the conclusions from the results in Figure 5 that pDMSt neurons can represent differences between success/failure or cued/uncued and don't find it as strong evidence supporting the author's argument that pDMSt neurons are required to learn the association between a cue and a behavior.

The results in Figure 5 seem entirely expected. Movements during success and failure trials are distinct (second half of the movement is different, e.g., mouse doesn't move hand to mouth, doesn't chew, etc.). Therefore, it

seems like these results can simply be interpreted as evidence that different movements are represented by different activity in the pDMSt, without anything to do with success or failure. An analysis of non-chewing, post-movement periods with careful behavior tracking to confirm there are no movement differences would be more convincing evidence of success/failure coding.

You are correct: the success and failure trials are different behaviorally. However, the *cued success* and *uncued success* trials *cannot* be distinguished behaviorally (**Supp. Fig. 12**). Yet the cued success and uncued success trials can be distinguished based on the neural activity. Therefore, we have a comparison of trials in which behavior is the same, yet neural activity is different. **Figure 5** shows decoding of cued vs. uncued successes using neural activity in the post-outcome period. The meaningful comparison is between cued success to uncued success, or cued failure to uncued failure, not, as the reviewer correctly points out, cued success to cued failure, nor uncued success to uncued failure. We now explain this more clearly in the main text (lines 315-319).

We do not think that it is trivial that the pDMSt neural activity can distinguish between a cued success and an uncued success many seconds after the cue has turned off.

Second, it also seems guaranteed that there is a difference in activity in pDMSt activity during cued or uncued trials, as the cue is an optogenetic activation of a population of V1 neurons that directly excites pDMSt neurons. Therefore, it seems unfair to call this “cue-related” activity as this confounds typical behavior cues and direct optogenetic activation of inputs. Therefore the logic in the paper of pDMSt encoding a cue in this context seems circular and confusing.

We apologize for the confusion. For the decoding, we are using only neural activity *after the cue* in the post-outcome period. The cue turns off more than 1 s before beginning the decoding. This is shown in **Figure 5i**. We now clarify this point (lines 330-333).

Furthermore, this paper is studying how a cue is associated with an event in the real world, which in this case is the presence of a food pellet. They state “According to reinforcement learning (RL) theory, reinforcement of an association between the cue and action depends on the outcome, such that only actions resulting in beneficial outcomes are reinforced”. It was not clear why analyses were conditioned based upon success or failure. I would have thought analyses conditioned on whether the presence of a pellet was associated with a cue (and the temporal properties of this association) would have been more appropriate. Perhaps, an even more interesting conditioning would have been if the animals received tactile information about the presence of the pellet.

We now provide an analysis comparing drops to successes (**Supp. Fig. 10b**). In trials in which the animal drops the pellet, tactile information about the presence of the pellet is received. In trials where the animal fails to touch any pellet (called “misses”), there is no tactile information. The effect of the drop is mid-way between a success and a miss. This suggests that some animals in some cases may be able to use the drop to learn about cued reaching.

(5) The analysis of the behavior is not comprehensive enough to support the claims. The authors state: “Mice learned to use this internal, optogenetic cue to guide the timing of the reach, without changing the previously established reach kinematics” (Line 122). However, a very clear drop in the ratio of drops to successes occurs after the cue is introduced. How can this change occur if not by a change in movement kinematics? In general, the analysis of behavior is severely limited. The quantifications of behavior are success/drop/miss rate and duration of 3 reach phases. Given the availability of tools to track detailed movements (which is already done in the work with DeepLabCut), quantification of the actual kinematics seems warranted for this level of work. The authors do provide reaching trajectories, which is a powerful visual tool, but further quantification is necessary. This should include consideration of both the spatial trajectory and velocity profile, including, e.g., quantification of single trial reaching amplitude, x/y/z reaching direction amplitude, magnitude of velocity & acceleration, consistency of spatial trajectory and/or velocity profile. Additionally, the pellet reaching task involves dexterous fine movements for grasping, which is not properly considered. Changes in how the digits open/close should be examined if claims about the movement kinematics changing/non changing are made, especially because these changes in digits may be controlled separately from the control of the outward reaching movement.

We have added more movement analyses.

1. Spatial trajectories, including quantification of reaching amplitude, x/y/z/ reaching direction amplitude: In **Supp. Fig. 6e**, we now show the reaching trajectories in multiple directions. There were no differences in the raw trajectories comparing control time periods to pDMSt silencing.

2. Velocity: In **Supp. Fig. 6c**, we show that there was no change in the velocity of the movement, because the duration of each reach epoch was unchanged. Because we did not see a velocity change, we did not measure acceleration.

3. Dexterous movement: In **Supp. Fig. 6b**, we show that the animals were equally successful at grabbing pellets during pDMSt silencing and control time periods. If there were a change in dexterous movement that affects grasping, we should see a change in the success rate of grabbing the pellet. We did not. However, note that these movement analyses are only a control. We are not interested in motor learning in this paper. We aim to study associative learning, which requires only that we can detect reaches and quantify how reach initiation becomes time-locked to the cue.

(6) The main result is a deficit in learning with consistent pDMSt manipulation. However, the consistency of the results when considering individual animals is weak and the entire effect in Figure 3 seems to be driven solely by 3/7 mice (Figure 3c). The figure 3e histogram with mouse count should only include the proper control mice (7 control mice), not grouped in with other mice which seem to skew the result.

They should also discuss why some animals still don't learn even after the perturbation has ended.

This is not correct. More than 3 of the mice learned. Learning is movement toward the upper left in **Figure 3c**. Some mice did not reach very much, and so the absolute change in reach rate is small. However, what is important is the *relative* change in reaching before vs. after the cue. This is what d-prime captures.

The histogram in **Figure 3e** (top) includes only the proper control mice (i.e., run as double-blind controls), as you suggest. We have also provided a second figure, **Figure 3e** (bottom), that shows all of the control mice (n=37 controls, not just double-blind controls), because this comparison has greater statistical power. We now provide statistics showing that there was no difference in learning between the double-blind control group and the control group including all 37 controls (legend of **Figure 3**).

To address the reviewer's second point, it may be that some mice become set in their ways after successfully performing the behavior for many weeks. Recall that animals can be successful in this task without paying attention to the cue and using a random reaching strategy. Random reaching still obtains the pellet with high probability. (In fact, the behavior box is open-loop – the same number of pellets with the same timing will be delivered independent of the mouse's strategy, use of the cue, or success rate.) This may affect recovery learning. However, to address this question statistically, we compared the recovery learning to the initial learning of control mice (legend of **Figure 3**). We find that there was not a statistically significant difference between the recovery mice and controls. Hence, we cannot say whether there is only partial recovery learning or complete recovery learning. This is why we use the phrase "progressed in learning".

We would also like to point out that it is extremely rare for any study to include this kind of data. We prevented learning for >20 days and then released the silencing to follow the same cohort of mice for another 20 days. Most studies just introduce a perturbation and compare across groups without showing recovery.

(7) I am unsure about the logic behind some of the method's decisions. For example, the rationale for using a reaching task is unclear, would results be different for a different behavior, licking?

Mice performing licking tasks tend to lick very frequently and struggle to suppress licking outside of the cue period. This is why researchers use "Enforced Non-Lick" periods in licking tasks, in order to select trials in which the animal pauses licking for at least a few seconds before the cue. In a simple sensorimotor associative learning task like ours, this high baseline rate of licking would be problematic, because it would be difficult to know whether the animal licked in response to the cue or licked spontaneously in hopes of receiving a reward. This high baseline licking rate would obscure the cue-evoked response. In contrast, mice do not *reach* at high rates at baseline. This is a huge advantage in our task, because we can then cleanly observe the cue-evoked response. We now explain this in lines 108-110.

In addition, reaching for food is a hard task. It allows us to have failure trials, which permits comparison of outcome-dependent trial-to-trial changes in behavior. In our hands, failure trials typically disappear quickly in licking tasks.

(8) If this pathway is critical for cue association then I would be very interested if it was also involved in related phenomenon such as reversal learning and extinction.

This is an interesting question, but the answer is beyond the scope of our manuscript and would require several more years of work. (Developing and establishing a new behavioral paradigm can take several years, and

running mice in the learning curve experiments would take many months.) However, toward the reviewer's question, we added a new behavior that uses an external visual cue. We show that the pDMSt is critical for learning an externally cued sensorimotor association. This lends support to the conclusion that the pDMSt neural activity is involved in learning new sensorimotor associations more generally.

(9) If the 5s post outcome period is relevant as posited in Figure 5, pDMSt inhibition should be applied to this period to see if it is causal. It would be useful to determine when pDMSt inhibition impacts trial-to-trial behavior. What if applied right after reaching for 1s, what if applied 1s after reaching for 1s, etc. This would have fully taken advantage of their optogenetic method.

We agree that these are interesting questions. However, please keep in mind that acquiring statistically relevant data at each time point requires a full set of experiments. We cannot simply divide the silencing trials between two time points and be done. We have to double the number of experiments.

Nevertheless, we have undertaken a major new set of experiments to address the reviewer's question. We now show data in which we varied the timing of the optogenetic inhibition of pDMSt (**Figure 4ef**). Our result provides insight into the moment when pDMSt activity is needed for learning. Interestingly, activity in pDMSt *during the cue* is not needed. pDMSt activity *after the outcome* is needed, i.e., in the post-outcome period. We discuss this in lines 276-280. In short, this suggests that it is *specifically* activity in this post-outcome period that underlies the learning update. We agree with the reviewer that this major new experiment takes full advantage of the optogenetic striatal inhibition method.

In **Figure 5**, we show that this activity is a combined representation of the cued context and the outcome. Hence, pDMSt activity that combines the cued context and outcome is important for learning.

Moreover, we added a new analysis in **Figure 5j** that tests in which time windows we are able to decode the trial-to-trial behavior update from activity. We find that it is this post-outcome time window (only after the reach) that predicts the trial-to-trial behavior update, consistent with the newly added optogenetic inhibition experiment in **Figure 4ef**.

Minor comments:

Line 67: "To overcome the challenge imposed by the low firing rates of the SPNs (Hikosaka et al., 1989; Barnes et al., 2005; Tang et al., 2021), we recorded the activity patterns of one thousand putative SPNs in pDMSt during the behavior." I am unsure what the challenge is, is it in recording low FR neurons with in vivo electrophysiology? In understanding their function? Low firing rate neurons with bursty activity may have higher SNR in some contexts in comparison to high FR baseline neurons, so not sure what this general statement is saying.

Thank you. We changed this sentence (lines 73-75).

Line 71: "SPNs in pDMSt encoded the combination of the reach, the outcome, and the cued versus uncued context of the reach. This combination predicted the behavioral change from trial to trial during learning, consistent with a specific function of pDMSt in incremental trial-to-trial learning." I may be missing something, but I don't see the analysis of physiology measures predicting across-trial changes.

We apologize for the lack of clarity. The reviewer is correct that we do not have sufficient statistical power in a single behavioral trial to make statements about exactly where a *single* trial lies in behavior space. Yet we are able to build up a *distribution* of reach rates, and this distribution shifts in accordance with the single-trial physiology data. To remove the confusing sentence and satisfy the reviewer's concern, we have now changed the sentence (lines 77-79).

Line 273: "Moreover, the neural activity in pDMSt after a success contains lingering information about the presence or absence of the cue more than several seconds after the cue ends." This statement is based on the decoder which is able to classify trials based on the post-outcome period (5s after outstretched hand). However, there is no evidence that only the start (e.g., 200ms) of this period is used for classification. If a statement about lingering information for several seconds is made, the authors should show that using activity from smaller time bins continues to correctly classify cued/uncued trials after the cue ends (0-500ms after cue ends, 500-1000ms after cue ends, 1000-1500ms after cue ends, and so on).

We apologize for the confusion. What we meant was that the cue information is still present during the post-outcome period, which begins at least 1 second after the end of the cue. Furthermore, we added the interesting new analysis that the reviewer suggests. We use smaller time bins and vary the decoding time window. Using

small time bins, we show that the physiology is able to predict the behavior update up to 5 seconds after the cue (**Figure 5j**).

Why is the activity in Supplemental Figure 2c and Supplemental 5h so different? The frequency of the response is completely different and 5h looks more like the distractor in figure 2c. This suggests that some quantification of how reliable the neural responses are to optogenetic stimulation in V1 is critical. How consistent is the induced “cue” activity between trials or between animals? If the frequency components are different across mice, might this impact how the activity travels to striatum, as temporal integration is an important characteristic of striatal responses? Could this explain high animal-to-animal variability?

This is an interesting point. We added the variability of the visual cortex response across mice (**Supp. Fig. 2d**). This variability might result from different units being sampled in each mouse. Another possible explanation is differences in opsin expression across mice. This variability may or may not explain why some mice learn better than others.

However, we emphasize that none of the many dozens of mice that we tested showed a pDMSt dependence after learning, and all of the mice experiencing consistent pDMSt inhibition during learning showed learning deficits (**Supp. Fig. 7**). Hence the major conclusion of this paper (“Striatum supports fast learning but not memory recall”) holds.

Would like to see a plot of reaching times relative to when a pellet is randomly presented before any optogenetic cue pairing. Also would be interesting to see the same plot after pairing with optogenetic cue is learned (i.e., reaching to randomly presented pellet, no cue, but after cue is learned).

Supp. Fig. 4e shows the reaching to a randomly presented pellet before learning the pairing with the optogenetic cue (no opsin mice). The orange line in **Supp. Fig. 4b** shows the reaching to a randomly presented pellet after learning the pairing with the optogenetic cue.

A “block” of inhibition trials, with inhibition on every trial was used to show that inhibiting pDMSt disrupts learning (Fig. 3), why was a block not used to show no effect on memory recall (Fig. 2). Would a block of trials have a different effect than interleaved trials?

In response to the reviewer’s question, we now show what happens during memory recall over a block of pDMSt inhibition trials (**Supp. Fig. 6f**). There is no change in reaching compared to control trials.

Additionally, it would add more understanding of the effect if the impact of inhibiting pDMSt was examined when the task parameters change and the animal needs to update the action (if the location of the food pellet is moved).

We do not study motor learning in this manuscript (we took great effort to isolate the cue association learning phase from the motor learning phase). Therefore, we did not test what happens when the animal needs to relearn motor kinematics of the reach during pDMSt silencing. The proposed experiment is interesting but beyond the current scope. Motor learning of this type likely requires other motor areas of the striatum.

Was there a reason for choosing 250ms for the visual cortex stim? Would shorter work? Why is the pDMSt inhibition 1s long? Do the animals reach or react after the stim?

We chose 250 ms because it roughly approximates the duration of the first phase of a real visual response (**Supp. Fig. 2d**, distractor). We did not test shorter stimulus durations. The pDMSt inhibition is 1 second long, because we wanted to inhibit over a time window that covers the cue, cued reach and post-outcome period without overlapping spontaneous reaches in the inter-trial interval. Even beginner animals do not make cued reaches beyond ~1-1.5 second. Therefore this 1 second window works well.

However, note that we added a major new experiment that varies the timing of the pDMSt inhibition, using a 500 ms-long inhibition window, to test the importance of different trial epochs (**Figure 4ef**).

Naive animals do not reach or react to the pDMSt inhibition (**Supp. Fig. 5g**). However, 1 of the 9 mice experiencing consistent pDMSt inhibition throughout learning did learn to reach at a very long and unnatural delay (**Supp. Fig. 7, Example Mouse C**). This mouse may have learned to respond to the rebound from pDMSt inhibition. We show this data in **Supp. Fig. 7**. We also now include videos of the behavior (**Supplementary Videos**).

Figure 1: Is the reaction time with a peak of ~300-400ms expected? Is this similar to a visual stimulus reaction time?

We added a major new experiment with a real visual cue. We find that the reaction time to the real visual cue also peaks around ~300 ms (**Supp. Fig. 9**).

Would like to know more information and numbers about the animals/sessions that were excluded, including the% of mice that failed the controls.

Mice infrequently cheated, i.e., reached preemptively before the cue or in the absence of a cue in a non-random way. Cheating occurred in <15% of all the sessions total. 44% of the mice never cheated, 51% of the mice cheated in <=5% of their sessions, 80% of mice cheated in <15% of their sessions, and the remaining 20% of mice cheated more often. We excluded sessions with cheating. Hence, this cheating is an uncommon behavior. But importantly, it is very obvious when the mouse is cheating, because the mouse reaches before the cue. We now report the rate of cheating in the main manuscript in line 150.

Is there histology to confirm the targeting of inhibition in the pDMSt to areas that receive V1 innervation and are being stimulated? Where do the V1 collaterals go?

Our viral approach ensures that the suppressed region of pDMSt is the region downstream of the cue neurons. This logic is as follows. We inject a retro-Flp into pDMSt, which (1) retrogradely infects visual cortex neurons that project to pDMSt and (2) turns on ReaChR expression in only the striatal interneurons that receive the Flp virus. Thus, limited spatial spread of the virus ensures that the SAME region is targeted for silencing and for receipt of the cue information from the visual cortex.

We now show some histology to demonstrate this targeting of inhibition to the areas that receive V1 innervation (**Supp. Fig. 2** and **Figure 2**). We also now investigate the function of another V1 target, the superior colliculus (**Supp. Fig. 7**) and discuss targets of the cue neuron collaterals in lines 188-201. This answers the reviewer's questions.

Would MSN activation directly in the pDMSt cause the behavior?

This is an interesting question, beyond the current scope of the manuscript. It seems quite possible that activating MSNs in pDMSt would cause the behavior, as it is well known in the field that optogenetic activations of striatum can elicit movements. We now discuss this in lines 396-398.

However, such gain-of-function experiments reveal what activity in a region *can* do as opposed to what it normally does. The latter is best informed by loss-of-function experiments such as the ones we perform here.

Does the pDMSt perturbation inhibit or perturb the probability of random, spontaneous reaches?

No. This can be seen in **Figure 3b** (red lines in left panel), **Supp. Fig. 5g** (bottom), and **Supp. Fig. 7**. We see no change in the baseline rate of spontaneous reaching. We make this clear in the text (line 352).

Color schemes should be followed more faithfully

Thank you. We have made revisions.

What are the faint lines in Figure 3d?

These indicate plus and minus standard error of the mean. We explain this in the legend. The data for the individual mice are shown in **Figure 3c**.

Many of the plots in Figure 4 show some effect along the x and/or y axis - why are these other effects not discussed? How are they interpreted?

The major effects under control conditions in **Figure 4** are, first, an increase in cued reaching after a successful cued reach (movement along Y axis) and, second, an increase in uncued reaching after a successful uncued reach (movement along X axis). The other effects are small, and this is why we do not discuss them.

The major effects of pDMSt inhibition are:

(1) (**Fig. 4c**, left) to prevent the increase in cued reaching after a cued success, with a concomitant decrease in both cued and uncued reaching (however, note the relative scale of the X axis, so the decrease in uncued reaching is small, which is why we don't discuss it)

(2) (**Fig. 4c**, middle-left) to prevent any decrease in cued reaching associated with a cued failure (but the decrease in cued reaching after a cued failure is not reliable – compare **Fig. 4b**, middle-left and **Fig. 4c**, middle-left, which is why we do not discuss it)

(3) (**Fig. 4c**, middle-right) no effect on the small but not significant (according to bootstrap) decrease in cued reaching, associated with a significant increase in uncued reaching after the uncued success (we do not discuss the insignificant decrease in cued reaching, because it is not significant)

(4) (**Fig. 4c**, right) an increase in BOTH X-axis and Y-axis reaching, about equally, after pDMSt inhibition interposed between the uncued reach and the next trial. This equal increase in reaching both after and before the cue implies a non-specific increase in reaching. We do not fully understand this final effect, but one possibility is that the “disappointment” after a failure is ameliorated or disrupted when pDMSt inhibition follows the uncued reach. This is an interesting possibility, but we do not discuss it, because it is irrelevant to the main point of the figure, which is that pDMSt inhibition disrupts the clear and reliable reinforcement of cued or uncued reaching after a success.